# Robust Regression Revisited:
# Acceleration and Improved Estimation Rates

**Arun Jambulapati**
Stanford University
jmblpati@stanford.edu

**Jerry Li**
Microsoft Research
jerrl@microsoft.com

**Tselil Schramm**
Stanford University
tselil@stanford.edu

**Kevin Tian**
Stanford University
kjtian@stanford.edu

## Abstract

We study fast algorithms for statistical regression problems under the strong contamination model, where the goal is to approximately optimize a generalized linear model (GLM) given adversarially corrupted samples. Prior works in this line of research were based on the *robust gradient descent* framework of [PSBR20], a first-order method using biased gradient queries, or the *Sever* framework of [DKK+19], an iterative outlier-removal method calling a stationary point finder.

We present nearly-linear time algorithms for robust regression problems with improved runtime or estimation guarantees compared to the state-of-the-art. For the general case of smooth GLMs (e.g. logistic regression), we show that the robust gradient descent framework of [PSBR20] can be *accelerated*, and show our algorithm extends to optimizing the Moreau envelopes of Lipschitz GLMs (e.g. support vector machines), answering several open questions in the literature.

For the well-studied case of robust linear regression, we present an alternative approach obtaining improved estimation rates over prior nearly-linear time algorithms. Interestingly, our algorithm starts with an identifiability proof introduced in the context of the sum-of-squares algorithm of [BP21], which achieved optimal error rates while requiring large polynomial runtime and sample complexity. We reinterpret their proof within the Sever framework and obtain a dramatically faster and more sample-efficient algorithm under fewer distributional assumptions.

## 1   Introduction

Parameter estimation in generalized linear models (GLMs), such as linear and logistic regression problems, is among the most fundamental and well-studied statistical optimization problems. It serves as the primary workhorse in statistical studies arising from a variety of disciplines, ranging from economics [Smi12], biology [VGSM05], and the social sciences [Gor10]. Formally, given a *link function* $\gamma : \mathbb{R}^2 \to \mathbb{R}$ and a dataset of covariates and labels $\{(X_i, y_i)\}_{i \in [n]} \subset \mathbb{R}^d \times \mathbb{R}$ drawn from an underlying distribution $\mathcal{D}_{Xy}$, the problem of statistical (generalized linear) regression asks to

$$\text{estimate } \theta^\star := \text{argmin}_{\theta \in \mathbb{R}^d} \left\{ \mathop{\mathbb{E}}_{(X,y) \sim \mathcal{D}_{Xy}} \left[ \gamma(\langle \theta, X \rangle, y) \right] \right\}. \tag{1}$$

For example, when $\gamma(v, y) = \frac{1}{2}(v-y)^2$, (1) corresponds to (statistical) linear regression. The problem (1) also has an interpretation as computing a maximum likelihood estimate for a parameterized distributional model for data generation, and indeed is only tractable under certain distributional

assumptions, since we only have access to samples from $\mathcal{D}_{Xy}$ rather than the underlying distribution itself (see e.g. [BP21] for tractability results in the linear regression setting).

However, in many modern settings these strong distributional assumptions fail to hold. In practically relevant settings, regression is often performed on massive datasets, where the data come from a poorly-understood distribution and have not been thoroughly vetted or cleaned of outliers. This has prompted the study of robust regression, i.e. regression under weak distributional or corruption assumptions. In this work, we study (1) in the *strong contamination model*. In this model, we assume the data points we receive are independently drawn from $\mathcal{D}_{Xy}$, but that an arbitrary $\epsilon$-fraction of the samples are then adversarially *contaminated* or replaced. The strong contamination model has garnered significant recent interest in the algorithmic statistics and learning communities for several reasons. Firstly, it is a *flexible* model of corruption and can be used to study both truly adversarial data poisoning attacks (where e.g. part of the dataset is sourced from malicious respondents), as well as model misspecification, where the generative $\mathcal{D}_{Xy}$ does not exactly satisfy our distributional assumptions, but is close in total variation to a distribution that does. Furthermore, a line of work building upon [DKK+16, LRV16] (discussed in our survey of prior work in Section 1.2) has achieved remarkable positive results for mean estimation and related problems under strong contamination, with statistical guarantees scaling independently of the dimension $d$. This dimension-free error promise is important in modern high-dimensional settings.

## 1.1 Our results

We give multiple *nearly-linear time algorithms*[1] for problem (1) under the strong contamination model, with improved statistical or runtime guarantees compared to the state-of-the-art. Prior algorithms for (1) under the strong contamination model in the literature typically followed one of two frameworks. The first, which we refer to as *robust gradient descent*, was pioneered by [PSBR20], and is based on reframing (1) as a problem where we have noisy gradient access to an unknown function we wish to optimize, coupled with the design of a noisy gradient oracle based on a robust mean estimation primitive. The second, which we refer to as *Sever*, originated in work of [DKK+19], and uses the guarantees of stationary point finders such as stochastic gradient descent to repeatedly perform outlier removal. Interestingly, we show that both approaches can be dramatically sped up, and give two complementary types of algorithms within these frameworks.

**Robust acceleration.** We demonstrate that under the noisy gradient estimation framework for solving well-conditioned regression problems (1), an *accelerated* rate of optimization can be achieved, answering an open question of [PSBR20]. We give the following result for smooth statistical regression, where we assume the uncorrupted data is drawn from $\mathcal{D}_{Xy}$ with marginals $\mathcal{D}_X$ and $\mathcal{D}_y$; throughout, $\widetilde{O}$ hides polylogarithmic factors in problem parameters and failure probabilities. Finally, we remove $\kappa$ dependences in sample complexity statements, as they are subsumed by $\epsilon$ dependences due to assumed bounds on $\epsilon\kappa$ or $\epsilon\kappa^2$.

**Theorem 1** (informal, cf. Theorem 7, supplement). *Suppose $\gamma : \mathbb{R}^2 \to \mathbb{R}$ is such that $\gamma_y(v) := \gamma(v, y)$ is convex and has (absolute) first and second derivatives at most 1 for all $y$ in the support of $\mathcal{D}_y$, and $\mathcal{D}_X$ has second moment matrix $\Sigma^\star \preceq L\mathbf{I}$. For some $\mu \geq 0$, let $\kappa := \max(1, \frac{L}{\mu})$ and let*

$$\theta^\star_{\mathrm{reg}} := \mathrm{argmin}_{\theta \in \mathbb{R}^d} \left\{ \mathop{\mathbb{E}}_{(X,y)\sim\mathcal{D}_{Xy}} \left\{ \gamma\left(\langle \theta, X \rangle, y \right) \right\} + \frac{\mu}{2} \|\theta\|_2^2 \right\}$$

*be the solution to the true regularized statistical regression problem.[2] There is an algorithm that given $n := \widetilde{O}(\frac{d}{\epsilon})$ $\epsilon$-corrupted samples from $\mathcal{D}_{Xy}$, for $\epsilon\kappa^2$ at most an absolute constant, obtains $\theta$ with $\|\theta - \theta^\star_{\mathrm{reg}}\|_2 = O\left(\sqrt{\frac{\kappa\epsilon}{\mu}}\right)$ with high probability in time $\widetilde{O}(nd\sqrt{\kappa})$.*

A canonical example of a link function $\gamma$ satisfying assumptions of Theorem 1 is the logit function $\gamma(v, y) = \log(1 + \exp(-vy))$, when the labels $y$ are $\pm 1$. To contextualize Theorem 1, [PSBR20]

---

[1]Throughout, we reserve the description "nearly-linear" for runtimes scaling linearly in the dataset size $nd$, and polynomially in $\epsilon^{-1}$ and the condition number, up to a polylogarithmic overhead in problem parameters.

[2]To simplify bounds and avoid estimation error for non-strongly convex statistical regression problems scaling with the initial search radius (which may be dimension-dependent), we focus on regularized problems. There is a substantial line of work on reductions between rates for strongly convex and convex smooth optimization in the non-robust setting, see e.g. [ZH16], and we defer an analogous exploration in the robust setting to future work.

obtains a similar statistical guarantee in its setting, using $\widetilde{O}(\kappa)$ calls to a *noisy gradient oracle*, implemented via a subroutine based on robust mean estimation. Since then, [CAT$^+$20] showed that for the case of linear regression (see Theorem 3 for the formal setup, as the linear regression link function is not Lipschitz), the framework was amenable to nearly-linear time mean estimation techniques of [CDG19], and gave an algorithm with runtime $\widetilde{O}(nd\kappa\epsilon^{-6})$. Theorem 1 represents an improvement to these results on two fronts: we apply tools from [DHL19] to remove the poly$(\epsilon^{-1})$ runtime dependence for a general class of regression problems, and we achieve an iteration count of $\widetilde{O}(\sqrt{\kappa})$, matching the accelerated runtime of [Nes83] for (non-robust) smooth optimization. We remark that the application of tools inspired by [DHL19] is fairly straightforward, and not a primary contribution of our work compared to the accelerated dependence on $\kappa$.

We demonstrate the generality of our acceleration framework by applying it to optimizing the *Moreau envelope* for Lipschitz, but possibly non-smooth, link functions $\gamma$; a canonical example is the hinge loss $\gamma(v, y) = \max(0, 1 - vy)$ with $\pm 1$ labels, used in training support vector machines. The Moreau envelope is an extremely well-studied smooth approximation, and is an everywhere additive approximation if the original function is Lipschitz (see e.g. [Sho97]). In the non-robust setting many state-of-the-art rates for Lipschitz optimization are attainable by accelerated optimization of an appropriate Moreau envelope [TJNO20]. We show that even without explicit access to the Moreau envelope, we can approximately minimize it with our robust acceleration framework.

**Theorem 2** (informal, cf. Theorem 8, supplement)**.** *Suppose $\gamma : \mathbb{R}^2 \to \mathbb{R}$ is such that $\gamma_y(v) := \gamma(v, y)$ is convex and has (absolute) first derivative at most 1 for all $y$ in the support of $\mathcal{D}_y$, and $\mathcal{D}_X$ has bounded second moment matrix. For some $\mu, \lambda \geq 0$, let $\kappa = \max(1, \frac{1}{\lambda\mu})$ and let*

$$\theta^\star_{\mathrm{env}} := \operatorname{argmin}_{\theta \in \mathbb{R}^d} \left\{ F^\star_\lambda(\theta) + \frac{\mu}{2} \|\theta\|_2^2 \right\}, \text{ where } F^\star_\lambda(\theta) := \inf_{\theta'} \left\{ F^\star(\theta') + \frac{1}{2\lambda} \|\theta - \theta'\|_2^2 \right\}$$

$$\text{is the Moreau envelope of } F^\star(\theta) := \mathbb{E}_{(X, y) \sim \mathcal{D}_{Xy}} \left\{ \gamma \left( \langle \theta, X \rangle, y \right) \right\}.$$

*There is an algorithm that given $n := \widetilde{O}(\frac{d}{\epsilon})$ $\epsilon$-corrupted samples from $\mathcal{D}_{Xy}$, for $\epsilon\kappa^2$ at most an absolute constant, obtains $\theta$ with $\left\| \theta - \theta^\star_{\mathrm{reg}} \right\|_2 = O\left( \sqrt{\frac{\kappa\epsilon}{\mu}} \right)$ with high probability in time $\widetilde{O}(\frac{nd\sqrt{\kappa}}{\epsilon})$.*

To obtain this result, we give a nearly-linear time construction of a noisy gradient oracle for the Moreau envelope, which may be of independent interest; we note similar gradient oracle constructions in different settings have been developed in the optimization literature (see e.g. [CJJS21]).

**Robust linear regression.** Perhaps the most ubiquitous example of statistical regression, the specific problem of robust linear regression has received substantial attention in the literature (cf. Section 1.2). Amongst the algorithms developed for the variant of this problem that we study, the only nearly-linear time algorithm is the recent work of [CAT$^+$20].[3] For a robust linear regression problem with noise variance bounded by $\sigma^2$ and covariate second moment matrix $\mathbf{\Sigma}^\star := \mathbb{E}_{X \sim \mathcal{D}_X}[XX^\top]$, [DKK$^+$19, PSBR20, CAT$^+$20] attain distance to the true regression minimizer $\theta^\star$ scaling as $\sigma\kappa\sqrt{\epsilon}$ in the $\mathbf{\Sigma}^\star$ norm (the "Mahalanobis distance")[4] under a bounded 4$^{\mathrm{th}}$ moment assumption. We give one result (Theorem 3) which improves the runtime of [DKK$^+$19, PSBR20, CAT$^+$20] under the noisy gradient descent framework, and one result (Theorem 4) which improves its estimation rate, under the Sever framework.

We first demonstrate that applying our robust acceleration framework leads to a similar estimation guarantee as [DKK$^+$19, PSBR20, CAT$^+$20] under the same assumptions, but with improved runtime.

**Theorem 3** (informal, cf. Theorem 6, supplement)**.** *Suppose $\mathcal{D}_X$ is 2-to-4 hypercontractive with second moment matrix $\mathbf{\Sigma}^\star = \mathbb{E}_{X \sim \mathcal{D}_X}[XX^\top]$ satisfying $\mu\mathbf{I} \preceq \mathbf{\Sigma}^\star \preceq L\mathbf{I}$, and $y \sim \mathcal{D}_y$ is generated as $\langle \theta^\star, X \rangle + \delta$, for $\delta \sim \mathcal{D}_\delta$ with variance at most $\sigma^2$ independent of $X$. Let $\kappa := \frac{L}{\mu}$. There is an algorithm that given $n := \widetilde{O}(\frac{d}{\epsilon})$ $\epsilon$-corrupted samples from $\mathcal{D}_{Xy}$, for $\epsilon\kappa^2$ at most an absolute constant, obtains $\theta$ with $\|\theta - \theta^\star\|_{\mathbf{\Sigma}^\star} = O(\sigma\kappa\sqrt{\epsilon})$ with high probability in time $\widetilde{O}(nd\sqrt{\kappa})$.*

---

[3]Another algorithm was recently given by [] under different assumptions, most notably a known second moment matrix.

[4]We measure error in the $\mathbf{\Sigma}^\star$ norm as it is scale invariant and the natural norm to measure the underlying (quadratic) statistical regression problem under; some prior works gave $\ell_2$ norm guarantees, which we convert.

We give a formal definition of 2-to-4 hypercontractivity in Section 2. We remark that attaining estimation rates for robust linear regression scaling polynomially in $\epsilon$ is known to be impossible under only bounded second moments ([BP21]); the $4^{\text{th}}$ moment bound we require is the minimal assumption known in the literature under which such robust estimation is possible.[5] Theorem 3 matches the distribution assumptions and error of [CAT+20], while obtaining an accelerated runtime.

Interestingly, under the $4^{\text{th}}$ moment bound used in Theorem 3, [BP21] showed that the information-theoretically optimal rate of estimation in the $\mathbf{\Sigma}^\star$ norm is *independent* of $\kappa$, and presented a matching upper bound under an analogous, but more stringent, distributional assumption.[6] However, thus far robust linear regression algorithms have broadly fallen under two categories. The first family (e.g. [KKM18, ZJS20, BP21]), based on the sum-of-squares paradigm for algorithm design, sacrifices practicality to obtain improved error rates by paying a large runtime and sample complexity overhead (as well as requiring stronger distributional assumptions). The second (e.g. [DKK+19, PSBR20, CAT+20]), which opts for more practical approaches to algorithm design, has been bottlenecked at Mahalanobis distance $O(\sigma\kappa\sqrt{\epsilon})$ and the requirement that $\epsilon\kappa^2 = O(1)$.

We present a nearly-linear time method for robust linear regression overcoming this bottleneck for the first time amongst non-sum-of-squares algorithms, attaining improved statistical rates compared to Theorem 3 while only requiring $\epsilon\kappa = O(1)$.

**Theorem 4** (informal, cf. Theorem 5, supplement). *Suppose $\mathcal{D}_X$ is 2-to-4 hypercontractive with second moment matrix $\mathbf{\Sigma}^\star = \mathbb{E}_{X\sim\mathcal{D}_X}[XX^\top]$ satisfying $\mu\mathbf{I} \preceq \mathbf{\Sigma}^\star \preceq L\mathbf{I}$, and $y \sim \mathcal{D}_y$ is generated as $\langle\theta^\star, X\rangle + \delta$, for $\delta \sim \mathcal{D}_\delta$, a 2-to-4 hypercontractive distribution with variance at most $\sigma^2$ independent of X. Let $\kappa := \frac{L}{\mu}$. There is an algorithm that given $n := \widetilde{O}((\frac{d^2}{\epsilon} + \frac{d}{\epsilon^4}))$ samples from $\mathcal{D}_{Xy}$, for $\epsilon\kappa$ at most an absolute constant, uses $\widetilde{O}(\frac{1}{\epsilon})$ calls to an empirical risk minimization algorithm[7] and $\widetilde{O}(\frac{nd}{\epsilon})$ additional runtime, and obtains $\theta$ with $\|\theta - \theta^\star\|_{\mathbf{\Sigma}^\star} = O(\sigma\sqrt{\kappa\epsilon})$ with probability at least $\frac{9}{10}$.*

This second algorithm does require more resources than that of Theorem 3: the sample complexity scales quadratically in $d$, and the runtime is never faster. Further, we make the slightly stronger assumption of hypercontractive noise for the uncorrupted samples. On the other hand, the improved dependence on the condition number in the error can be significant for distributions in practice, which may be far from isotropic. All told, Theorem 4 presents an intermediate tradeoff inheriting some statistical gains of the sum-of-squares approach (albeit still depending on $\kappa$) without sacrificing a nearly-linear runtime. Interestingly, we obtain Theorem 4 by reinterpreting an identifiability proof used in the algorithm of [BP21], and combining it with tools inspired by the Sever robust estimation framework. We note that our sample complexity dramatically improves that of [DKK+17]'s original linear regression algorithm in the Sever framework in terms of the dependence on $d$ (but not $\epsilon$), which requires $\widetilde{O}(\frac{d^5}{\epsilon^2})$ samples (in addition to beating their weaker error guarantee). We elaborate on these points further in Section 4; we believe it is an interesting open problem to understand if the worse sample complexity of Theorem 4 is truly necessary for the improved statistical guarantees.

## 1.2 Prior work

We give a general overview contextualizing our work in this section, and defer the comparison of technical components we develop in this work to the relevant sections. Specifically, in Section 3 we discuss prior work on acceleration with noisy gradients, and how it compares to our results.

The study of learning in the presence of adversarial noise is known as *robust statistics*, with a long history dating back over 60 years [Ans60, Tuk60, Hub64, Tuk75, Hub04]. Despite this, the first efficient algorithms with near-optimal error for many fundamental high dimensional robust statistics problems were only recently developed [DKK+16, LRV16, DKK+17]. Since these works, efficient robust estimators have been developed for a variety of more complex problems; a comprehensive overview may be found in [DK19, Li18, Ste18].

---

[5] This is true not only for efficient algorithms but even information-theoretically in the case of $\mathcal{D}_X$.

[6] The algorithm of [BP21] requires $\mathcal{D}_X$ to be *certifiably hypercontractive*, an algebraic condition frequently required by the sum-of-squares algorithmic paradigm to apply to robust statistical estimation problems.

[7] The empirical risk minimization algorithm used is up to the practitioner; its runtime will never scale worse than $\widetilde{O}(nd\sqrt{\kappa})$ by applying (non-robust) accelerated gradient descent, but can be substantially better if recent advances in stochastic gradient methods are used, e.g. [Zhu17].

Table 1: Robust linear regression results in the $\kappa \gg 1$ regime. All listed results assume 2-to-4 hypercontractivity and independent noise (although some also give rates for non-independent noise). Error guarantees are in Mahalanobis distance. We omit polylogarithmic factors for simplicity.

| Reference | Runtime | Error guarantee | Range of $\epsilon$ | Comments |
|-----------|---------|-----------------|---------------------|----------|
| [KKM18] | $\mathrm{poly}(d)$ | $\sigma\epsilon^{\frac{1}{4}}$ | N/A | Certifiable hypercontractivity |
| [ZJS20] | $\mathrm{poly}(d)$ | $\sigma\epsilon^{\frac{1}{2}}$ | N/A | Certifiable hypercontractivity |
| [BP21] | $\mathrm{poly}(d)$ | $\sigma\epsilon^{\frac{3}{4}}$ | N/A | Certifiable hypercontractivity |
| [DKK$^+$19] | $\mathrm{poly}(d)$ | $\sigma\kappa\epsilon^{\frac{1}{2}}$ | $O(\kappa^{-2})$ | Linear sample size |
| [CAT$^+$20] | $nd\kappa \cdot \mathrm{poly}(\epsilon^{-1})$ | $\sigma\kappa\epsilon^{\frac{1}{2}}$ | $O(\kappa^{-2})$ | Linear sample size |
| Theorem 3 | $nd\sqrt{\kappa}$ | $\sigma\kappa\epsilon^{\frac{1}{2}}$ | $O(\kappa^{-2})$ | Linear sample size |
| Theorem 4 | $nd \cdot \mathrm{poly}(\epsilon^{-1})$ | $\sigma(\kappa\epsilon)^{\frac{1}{2}}$ | $O(\kappa^{-1})$ | Quadratic sample size |

Our results sit within the line of work in this field on robust stochastic optimization. The first works which achieved dimension-independent error rates with efficient algorithms for the problems we consider in this paper are the aforementioned works of [PSBR20, DKK$^+$19]. Similar problems were previously considered in [CSV17a, BDLS17]. In [CSV17a], the authors consider a setting where a majority of the data is corrupted, and the goal is to output a short list of hypotheses so that at least one is close to the true regressor. However, because most of their data is corrupted, they achieve weaker statistical rates; in particular, their techniques do not achieve vanishing error as the fraction of error goes to zero. In [BDLS17], the authors consider a somewhat different model with stronger assumptions on the structure of the functions. In particular, they assume that the uncorrupted covariates are Gaussian with identity covariance, and are primarily concerned with the case where the regressors are sparse. Their main goal is to achieve sublinear sample complexities by leveraging sparsity. We also remark that the algorithms in [CSV17a, BDLS17] are also much more cumbersome, requiring heavy-duty machinery such as black-box SDP solvers and cutting plane methods, and as a result are more computationally intensive than those considered in [PSBR20, DKK$^+$17].

There has been a large body of subsequent work on the special case of robust linear regression [KKM18, KKK19, DKS19, ZJS20, CAT$^+$20, BP21]; however, the majority of this line of work focuses on achieving improved error rates under additional distributional assumptions by using the sum-of-squares hierachy. As a result, their algorithms are likely impractical in high dimensions, and require large (albeit polynomial) sample complexity and runtime. Of particular interest to us is [CAT$^+$20], who combine the framework of [PSBR20] with the robust mean estimation algorithm of [CDG19] to achieve nearly-linear runtimes in the problem dimension and the number of samples. Our Theorem 3 can be thought of as the natural accelerated version of [CAT$^+$20], with an additional $\epsilon^{-6}$ runtime overhead removed using more sophisticated mean estimation techniques.

After the initial submission of this paper, we were made aware of another recent work [PJL20] preventing a different approach to robust regression based on covariate filtering. Their work focuses on the highly well-conditioned setting, where the true second moment matrix has constant condition number, whereas our work primarily aims to improve various rates in terms of dependence on the condition number $\kappa$ in the ill-conditioned regime (i.e. $\kappa \gg 1$).

## 2  Preliminaries

**Notation.** For $d \in \mathbb{N}$ we let $[d] := \{j \mid j \in \mathbb{N}, 1 \le j \le d\}$. The $\ell_p$ norm of a vector is $\|\cdot\|_p$. The all-ones vector (of appropriate dimension from context) is $\mathbb{1}$, and the identity matrix is $\mathbf{I}$. The (solid) probability simplex is $\Delta^n := \{w \in \mathbb{R}^n_{\ge 0}, \|w\|_1 \le 1\}$. For $v \in \mathbb{R}^n$ and $S \subseteq [n]$, we let $v_S \in \mathbb{R}^n$ zero out coordinates $[n] \setminus S$. We call $S_1, S_2$ a *bipartition* of $S$ if $S_1 \cap S_2 = \emptyset$ and $S_1 \cup S_2 = S$. For symmetric $d \times d$ $\mathbf{A}, \mathbf{B}$ we write $\mathbf{A} \preceq \mathbf{B}$ to mean $\mathbf{B} - \mathbf{A}$ is positive semidefinite. For positive definite $\mathbf{M}$, we define the induced norm $\|v\|_{\mathbf{M}} := \sqrt{v^\top \mathbf{M} v}$. We use $\|\cdot\|_{\mathrm{op}}$ to mean the $\ell_2$-$\ell_2$ operator norm.

**Functions.** We say differentiable $f : \mathbb{R}^d \to \mathbb{R}$ is $\lambda$-Lipschitz in $\|\cdot\|_{\mathbf{M}}$ if $\|\nabla f(\theta)\|_{\mathbf{M}^{-1}} \le \lambda$ for all $\theta \in \mathbb{R}^d$, and that twice-differentiable $f : \mathbb{R}^d \to \mathbb{R}$ is $L$-smooth and $\mu$-strongly convex in $\|\cdot\|_{\mathbf{M}}$ if $\mu\mathbf{M} \preceq \nabla^2 f(\theta) \preceq L\mathbf{M}$, for all $\theta \in \mathbb{R}^d$. When $\mathbf{M}$ is not specified, we assume $\mathbf{M} = \mathbf{I}$.

**Distributions.** For $w \in \Delta^n$ and a set of vectors $\mathbf{X} := \{X_i\}_{i \in [n]}$, the empirical second moment matrix is denoted $\mathrm{Cov}_w(\mathbf{X}) := \sum_{i \in [n]} \frac{w_i}{\|w\|_1} X_i X_i^\top$. We say distribution $\mathcal{D}$ supported on $\mathbb{R}^d$ is 2-to-4 hypercontractive if for all $v \in \mathbb{R}^d$, $\mathbb{E}_{X \sim \mathcal{D}}[\langle X, v \rangle^4] \le O(1) \mathbb{E}_{X \sim \mathcal{D}}[\langle X, v \rangle^2]^2$.

**Corruption model.** We provide provable guarantees for statistical problems captured by the following standard statistical model (the "strong contamination model"): given link function $\gamma : \mathbb{R}^2 \to \mathbb{R}$ and a "true" underlying distribution $\mathcal{D}_{Xy}$ on $\mathbb{R}^d \times \mathbb{R}$ with marginals $\mathcal{D}_X, \mathcal{D}_y$, a dataset $\{(\tilde{X}_i, \tilde{y}_i)\}_{i \in [n]}$ is independently sampled from $\mathcal{D}_{Xy}$, and an arbitrary (unknown) subset $B \subset [n]$ is replaced by arbitrary points in $\mathrm{supp}(\mathcal{D}_{Xy})$. For $i \in B$, we observe the corrupted $(X_i, y_i)$, and otherwise we observe $(X_i, y_i) \leftarrow (\tilde{X}_i, \tilde{y}_i)$. When $|B| = \epsilon n$, we say the dataset is $\epsilon$-corrupted. We define $f_i(\theta) := \gamma(\langle X_i, \theta \rangle, y_i)$, $g_i(\theta) := \nabla f_i(\theta)$, and for $w \in \Delta^n$, $F_w(\theta) := \sum_{i \in [n]} w_i f_i(\theta)$.

**Filtering.** Our algorithms make frequent use of an algorithmic primitive we call "filtering" [ABL14, DKK$^+$17, SCV18]. Here, we have an index set $[n]$ with fixed unknown bipartition $G \cup B$; intuitively, $G$ is a "good" set of indices we wish to keep, and $B$ is a "bad" set we would like to remove. Naively, one would expect that we would choose $G$ to be the set of remaining uncorrupted points. For technical reasons, we will often have to choose (large) subsets of the set of remaining uncorrupted points.

The filtering algorithm maintains weights $w \in \Delta^n$ it iteratively downweights using "scores" $\tau \in \mathbb{R}_{\ge 0}^n$, with the goal of producing weights close to the uniform distribution on $G$. We call $w \in \Delta^n$ $c$-saturated (or *saturated* for short if $c = 0$) if $w \le \frac{1}{n} \mathbb{1}$ entrywise and $\left\| [\frac{1}{n} \mathbb{1} - w]_G \right\|_1 \le \left\| [\frac{1}{n} \mathbb{1} - w]_B \right\|_1 + c$. The following lemma (implicit in [DKK$^+$17, CSV17b, Li18, Ste18]) is representative of this technique.

**Lemma 1.** *Suppose $G \cup B$ is a bipartition of $[n]$, and $\langle w_G, \tau \rangle \le \langle w_B, \tau \rangle$ for $c$-saturated $w \in \Delta^n$, $\tau \in \mathbb{R}_{\ge 0}^n$. Then if $w_i' \leftarrow (1 - \frac{\tau_i}{\tau_{\max}}) w_i$ $\forall i \in [n]$ and $\tau_{\max} := \max_{i \in [n] | w_i \ne 0} \tau_i$, $w'$ is also $c$-saturated.*

In other words, we can keep weights saturated (retaining most of the weight on $G$) by repeatedly identifying scores whose empirical average is large due to $B$. We use a variety of different scores; one subroutine which follows as a consequence of this paradigm, which is implicit in [DHL19], is a way to rapidly decrease the operator norm of an empirical second moment matrix whose restriction to a "good" majority set is bounded. Many of our algorithms use this subroutine, stated here.

**Lemma 2** (cf. Proposition 5, supplement). *There is an algorithm, FastCovFilter, taking inputs $\mathbf{V} := \{v_i\}_{i \in [n]} \in \mathbb{R}^{n \times d}$, saturated $w \in \Delta^n$ with respect to $[n] = G \cup B$ with $|B| = \epsilon n$, and $R \ge 0$ with the promise that $\|\frac{1}{|G|} \sum_{i \in G} v_i v_i^\top\|_{\mathrm{op}} \le R$. Then, FastCovFilter returns saturated $w' \in \Delta^n$ such that $\|\sum_{i \in [n]} w_i' v_i v_i^\top\|_{\mathrm{op}} = O(R)$ with high probability in time $\widetilde{O}(nd)$.*

**Organization.** In Section 3, we describe a general framework for attaining accelerated optimization rates under a "noisy gradient oracle" model, which we use to obtain Theorems 1, 2, and 3 by efficiently constructing appropriate oracles. In Section 4, we overview techniques used in designing the nearly-linear time linear regression algorithm of Theorem 4. Due to space constraints, we defer proofs, a precise statement of our statistical models and assumptions, and an extended exposition of our techniques to the unabridged version of this paper in the supplementary material.

## 3 Acceleration under noisy gradient oracle access

Our robust acceleration framework, which we use to prove Theorems 1, 2, and 3, addresses the following problem formulation: there is an unknown function $F^\star$ with minimizer $\theta^\star$ which is $L$-smooth and $\mu$-strongly convex, and we wish to estimate $\theta^\star$, but our only mode of accessing $F^\star$ is through a *noisy gradient oracle* $\mathcal{O}_{\mathrm{ng}}$. Namely, for some $\sigma, \epsilon$, we can query $\mathcal{O}_{\mathrm{ng}}$ with some $\theta \in \mathbb{R}^d$ and an upper bound $R \ge \|\theta - \theta^\star\|_2$ and receive an estimate $G(\theta)$ such that

$$\|G(\theta) - \nabla F^\star(\theta)\|_2 = O\left(\sqrt{L\epsilon}\sigma + L\sqrt{\epsilon}R\right). \tag{2}$$

In other words, we receive gradients perturbed by both fixed additive noise, and multiplicative noise depending on (a known upper bound on) distance to $\theta^\star$. Prior works observed that by using tools from robust mean estimation, appropriate noisy gradient oracles could be constructed for the "mean" functions $\mathbb{E}_{(X,y) \sim \mathcal{D}_{Xy}}[\gamma(\langle X, \theta \rangle, y)]$ arising from the distributional assumptions in Theorems 1 and 3.

**Efficient $\mathcal{O}_{\mathrm{ng}}$ construction.** Our first contribution is speeding up the implementation of $\mathcal{O}_{\mathrm{ng}}$ to run in nearly-linear time $\widetilde{O}(nd)$, leveraging recent advances by [DHL19] for robust mean estimation. This

improves a similar construction by [CAT+20], running in time $\widetilde{O}(nd \cdot \epsilon^{-6})$ in the linear regression setting. To obtain this construction, we use the following lemma implicit in [DHL19].

**Lemma 3** (Lemma 11, supplement). *Let $w \in \Delta^n$ be saturated with respect to bipartition $[n] = G \cup B$, and let $w_G^\star := \frac{1}{|G|} \mathbb{1}_G$, $\tilde{w} := \frac{w}{\|w\|_1}$. Then,*

$$\left\| \nabla F_{\tilde{w}}(\theta) - \nabla F_{w_G^\star}(\theta) \right\|_2 = O\left(\sqrt{\epsilon}\right) \left( \left\| \mathrm{Cov}_{w_G^\star}\{g_i(\theta)\}_{i \in [n]} \right\|_{\mathrm{op}}^{\frac{1}{2}} + \left\| \mathrm{Cov}_{\tilde{w}}\{g_i(\theta)\}_{i \in [n]} \right\|_{\mathrm{op}}^{\frac{1}{2}} \right).$$

By demonstrating that our "uncorrupted" dataset has function gradients which are appropriately bounded under our distributional assumptions, we can apply FastCovFilter (Lemma 2) in conjunction with the above lemma to produce an weighting $\tilde{w} \in \Delta^n$ whose empirical gradient satisfies (2), in time $\widetilde{O}(nd)$. We give a $\mathcal{O}_{\mathrm{ng}}$ construction for linear regression in the distributional model of Theorem 3 and a *radiusless* $\mathcal{O}_{\mathrm{ng}}$ construction for smooth GLMs (satisfying (2) with no dependence on $R$ and $\sigma = 1$) in the distributional model of Theorem 1, in Corollaries 1 and 2 of the supplement. Finally, in the setting of Theorem 2 (Lipschitz GLMs), we give a reduction from constructing a noisy gradient oracle for Moreau envelopes to $\widetilde{O}(\epsilon^{-1})$ queries of a noisy gradient oracle (cf. Corollary 3, supplement) using a characterization of the Moreau gradient as a solution to a regularized objective.

**Noisy proximal oracle construction.** For the remainder of this section, we assume access to $\mathcal{O}_{\mathrm{ng}}$ satisfying (2). Our second, and much more technically involved, contribution is demonstrating that acceleration is achievable under the noise model (2). Designing accelerated algorithms under noisy gradient access is extremely well-studied, and there are both strong positive results [d'A08, MS13, DG16, CDO18, MRJ19, BJL+19] as well as negative results [DGN14] showing under certain noise assumptions, accelerated gradient descent may be outperformed by unaccelerated methods. Motivated by these negative results, [PSBR20] posed the question of whether acceleration was possible under (2). We demonstrate the *accelerated proximal point framework* of [MS13] is amenable to such noise.

In the noiseless setting, accelerated proximal point algorithms (introduced by [Gül92]) are a reduction from optimization of a function $F^\star$ to iteratively solving *proximal subproblems* of the form

$$\theta_{\bar{\theta}}^\star \leftarrow \mathrm{argmin}_{\theta \in \mathbb{R}^d} \left\{ F^\star(\theta) + \frac{1}{2\lambda} \left\| \theta - \bar{\theta} \right\|_2^2 \right\}, \tag{3}$$

for some $\bar{\theta} \in \mathbb{R}^d$. By tuning $\lambda$ we can trade off how efficiently we can solve the subproblem with how many times we need to solve them. In our accelerated algorithm, we will always set $\lambda = \frac{1}{L}$ in proximal subproblems. We first use the (unaccelerated) robust gradient descent analysis of [PSBR20, CAT+20] to show that given a current upper bound $R$ on $\left\| \bar{\theta} - \theta^\star \right\|_2$, we can solve the proximal subproblems (3) to appropriate accuracy, yielding a "noisy proximal oracle."

**Lemma 4** (Definition 5, Proposition 7, supplement). *There is an algorithm, NoisyProximalOracle, which given $\bar{\theta} \in \mathbb{R}^d$, $R \geq \left\| \bar{\theta} - \theta^\star \right\|_2$, and $\mathcal{O}_{\mathrm{ng}}$ giving noisy gradient estimates satisfying (2), returns $\hat{\theta}$ satisfying $\|\hat{\theta} - \theta_{\bar{\theta}}^\star\| = O(\sigma\sqrt{\frac{\epsilon}{L}} + \sqrt{\epsilon}R)$, where $\theta_{\bar{\theta}}^\star$ is the minimizer to (3). The complexity of the algorithm is dominated by $\widetilde{O}(1)$ calls to a noisy gradient oracle $\mathcal{O}_{\mathrm{ng}}$.*

In other words, NoisyProximalOracle runs in $\widetilde{O}(nd)$ time (as expected, since (3) with $\lambda = L^{-1}$ has condition number $\leq 2$). We now discuss how to use NoisyProximalOracle in our main subroutine.

**Halving the radius with** NoisyProximalOracle**.** Our accelerated algorithm runs in logarithmically many phases, each halving an upper bound on the distance to the optimizer (while above a certain noise floor, $\Omega(\sigma(\frac{\kappa\epsilon}{\mu})^{\frac{1}{2}})$, due to the additive error in (2)). We state one phase of our accelerated algorithm below as HalfRadiusAccel, assuming access to oracle $\mathcal{O}_{\mathrm{np}}$ with the guarantees of Lemma 4.[8]

In Line 5, scalars $\{a_t, A_t\}$ are parameters used by acceleration potential analyses, defined by recursions $A_0 = 0$, $A_t = La_t^2$, $A_{t+1} = A_t + a_{t+1}$. We remark HalfRadiusAccel is simply the proximal framework of [MS13], where we call NoisyProximalOracle in place of a (classical) proximal oracle in Line 6, and constrain Line 7. Our main technical innovation is to show the analysis of [MS13] is robust to guarantees of NoisyProximalOracle, by carefully balancing accumulated errors.

---

[8]Though our algorithm is randomized, we omit discussion of failure here for brevity. Our algorithm succeeds with high probability (with sample sizes and runtimes depending polylogarithmically on failure probabilities).

---

**Algorithm 1** HalfRadiusAccel($\bar{\theta}, R, \mathcal{O}_{\text{np}}$)

---

1: **Input:** $\bar{\theta} \in \mathbb{R}^d$, $R \geq \left\|\bar{\theta} - \theta^\star\right\|_2$ with $R = \Omega(\sigma(\frac{\kappa\epsilon}{\mu})^{\frac{1}{2}})$, $\mathcal{O}_{\text{np}}$ satisfying guarantees of Lemma 4
2: **Output:** $\hat{\theta} \in \mathbb{R}^d$ with $\|\hat{\theta} - \theta^\star\|_2 \leq \frac{1}{2}R$
3: $T \leftarrow O(\sqrt{\kappa})$ for a sufficiently large constant, $t \leftarrow 0$, $\theta_0 \leftarrow \bar{\theta}$, $v_0 \leftarrow \bar{\theta}$
4: **while** $t < T$ **do**
5: $\quad y_t \leftarrow \frac{A_t}{A_{t+1}}\theta_t + \frac{a_{t+1}}{A_{t+1}}v_t$
6: $\quad \theta_{t+1} \leftarrow \mathcal{O}_{\text{np}}(y_t, 3R)$
7: $\quad v_{t+1} \leftarrow \operatorname{argmin}_{v \in \mathbb{B}}\{a_{t+1}\langle y_t - \theta_{t+1}, v\rangle + \frac{1}{2L}\|v - v_t\|_2^2\}$, where $\mathbb{B} := \{v \mid \left\|v - \bar{\theta}\right\|_2 \leq R\}$
8: $\quad t \leftarrow t+1$
9: **end while**
10: **return** $\theta_t$

---

To make the analysis more modular, we first observe via characterizing proximal minimizers $\theta^\star_{y_t}$ (following (3)) that though iterates of Algorithm 1 may drift more than $R$ away from $\theta^\star$ (due to error in $\mathcal{O}_{\text{np}}$), they never drift too far. Formally, Lemma 13 of the supplement shows all $y_t$ satisfy $\|y_t - \theta^\star\|_2 \leq 3R$, validating calls to $\mathcal{O}_{\text{np}}$. We then give our main noisy potential bound.

**Lemma 5** (main potential bound, Lemma 14, supplement). *For all $0 \leq t \leq T$, define $E_t := F^\star(\theta_t) - F^\star(\theta^\star)$, $D_t := \frac{1}{2}\|v_t - \theta^\star\|_2^2$, and $\Phi_t := A_t E_t + D_t$. Then for all $0 \leq t < T$,*

$$\Phi_{t+1} - \Phi_t \leq O\left(t\sigma\sqrt{\frac{\epsilon}{L}}R + t\sqrt{\epsilon}R^2 + t^2\sigma^2\frac{\epsilon}{L} + t^2\epsilon R^2\right).$$

The potential $\Phi_t$ defined in Lemma 5 is typical in acceleration analyses, and indeed the (non-robust) analysis of [MS13] shows $\Phi_{t+1} \leq \Phi_t$ under an exact proximal oracle. Since the coefficient of function error in the potential, $A_t$, grows as $\Omega(\frac{t^2}{L})$, it is straightforward (via strong convexity) to show HalfRadiusAccel halves the radius in $O(\sqrt{\kappa})$ iterations classically. We demonstrate the analysis still goes through under the potential increase of Lemma 5 in the same number of iterations (up to constant factors), concluding analysis of HalfRadiusAccel. Our final algorithm iterates HalfRadiusAccel until reaching the noise floor, requiring $\widetilde{O}(\sqrt{\kappa})$ iterations dominated by calls to $\mathcal{O}_{\text{ng}}$.

**Discussion.** Our result builds upon recent studies of the noise tolerance of proximal point methods, e.g. [BJL+19], who used a model with fixed additive gradient error, and [CJJS21], who used noisy solutions to (3) with fixed additive error. Our model crucially tolerates both additive and multiplicative guarantees for both gradient and proximal solution estimates. An interesting open question is whether a similar result is obtainable without a proximal point abstraction, e.g. by extending [CDO18]. We view our result as a proof-of-concept that acceleration is possible under (2). We believe a unified study of acceleration under models encompassing (2) warrants further exploration, and defer it to interesting future work.

## 4 Improved statistical rates for robust linear regression

We now overview the algorithm that achieves Theorem 4, and describe its correctness proof. We define the following, strong deterministic condition on the uncorrupted points.

**Assumption 1** (deterministic regularity for linear regression). Let $\epsilon$ be sufficiently small, and let $r \in (0, \epsilon^2)$. Assume $\{(X_i, y_i)\}_{i \in [n]} \subset \mathbb{R}^d \times \mathbb{R}$ is $(\epsilon, r)$-*good for linear regression* (or $(\epsilon, r)$-good if context is clear), which means there is a partition $[n] = G \cup B$ with $|G| \geq (1 - \epsilon)n$ which satisfies:

1. For any $w \in \Delta^n$ with $\|w_G\|_1 \geq 1 - 4\epsilon$, $\frac{1}{2}\Sigma^\star \preceq \text{Cov}_{w_G}(\mathbf{X}) \preceq \frac{3}{2}\Sigma^\star$.

2. There is a constant $C_{\text{est}}$ such that for all $\theta \in \mathbb{R}^d$ and all $\epsilon$-saturated $w \in \Delta^n$, there exists a $G' \subseteq G$ satisfying $|G'| \geq (1 - r)|G|$ so that if we let $\tilde{w} := \frac{w_{G'}}{\|w_{G'}\|_1}$,

$$\|\nabla F_{\tilde{w}}(\theta) - \nabla F^\star(\theta)\|_2 \leq C_{\text{est}}\sqrt{L}\epsilon\left(\sigma + \|\theta - \theta^\star\|_{\Sigma^\star}\right), \tag{4}$$

$$\left\|\text{Cov}_{\tilde{w}}\left(\{g_i(\theta)\}_{i \in G}\right)\right\|_{\text{op}} \leq C_{\text{est}}L\left(\|\theta - \theta^\star\|_{\Sigma^\star}^2 + \sigma^2\right). \tag{5}$$

3. There is a constant $C_{\mathrm{ub}}$ such that $\sum_{i \in G} \frac{1}{|G|} f_i(\theta^\star) = \frac{1}{2|G|} \sum_{i \in G} (\langle X_i, \theta^\star \rangle - y_i)^2 \leq C_{\mathrm{ub}} \sigma^2$.

The following claim establishes that Assumption 1 (up to constants in definitions) holds with good probability under the statistical model in Theorem 4. We give the proof in the supplement.

**Proposition 1** (Proposition 2, supplement). *Let $\alpha \geq 1$ and $\epsilon > 0$ be sufficiently small. Let $\{(X_i, y_i)\}_{i \in [n]} \subset \mathbb{R}^d \times \mathbb{R}$ be $\epsilon$-corrupted samples from $\mathcal{D}_{Xy}$ as in Theorem 4. Then, if*

$$n = O\left( \frac{d\alpha^2 \log d}{\epsilon^4} + \frac{d^2 \alpha \log(d/\epsilon)}{\epsilon^2} \right) ,$$

*the set $\{(X_i, y_i)\}_{i \in [n]}$ is $(2\epsilon, \frac{\epsilon^2}{\alpha})$-good for linear regression with probability at least $\frac{9}{10}$.*

Through this section, we will assume that our set of points is $(2\epsilon, \frac{\epsilon^2}{\alpha})$-good, where we choose $\alpha = O\left(\max\left(1, \epsilon\kappa \log \frac{R_0}{\sigma}\right)\right)$. Here, $R_0$ is an initial distance bound on $\|\theta_0 - \theta^\star\|_{\mathbf{\Sigma}^\star}$. This assumption inflates the sample complexity of Proposition 1 by a factor depending on $\alpha$.

**Filtering under goodness.** Enforcing Assumption 1, and in particular, Assumption 1.2 holds inductively introduces some additional technical difficulties for our algorithm. However, our choice of $\alpha$ ensures that we never remove more than $3\epsilon$ mass from $G$, and that our weights are always $\epsilon$-saturated with respect to $G \cup B$, allowing for inductive application of Assumption 1. This holds because we show that our algorithm uses at most $\frac{\alpha}{2\epsilon}$ distinct subsets of $G$ for filtering. Each distinct set satisfies Lemma 1 up to a $\frac{2\epsilon^2}{\alpha}$ additive discrepancy, whose accumulation loses an additive $\epsilon$.

**Algorithms through identifiability.** We now describe our algorithm, assuming Assumption 1 holds throughout. We begin with the following (slight) reinterpretation of an identifiability proof in [BP21].

**Proposition 2** (Proposition 6, supplement). *Let $w \in \Delta^n$ be $\epsilon$-saturated with respect to bipartition $[n] = G \cup B$, and let $\theta \in \mathbb{R}^d$. Assuming $\epsilon\kappa$ is sufficiently small,*

$$\|\theta - \theta^\star\|_{\mathbf{\Sigma}^\star} = O\left( \sqrt{\epsilon} \left( \sigma\sqrt{\kappa} + \sqrt{\left\| \mathrm{Cov}_w \left( \{g_i(\theta)\}_{i \in [n]} \right) \right\|_{\mathrm{op}} \cdot \mu^{-1}} \right) + \|\nabla F_w(\theta)\|_{(\mathbf{\Sigma}^\star)^{-1}} \right).$$

This shows if $\|\nabla F_w(\theta)\|_{(\mathbf{\Sigma}^\star)^{-1}}$ and $\|\mathrm{Cov}_w(\{g_i(\theta)\}_{i \in [n]})\|_{\mathrm{op}}$ are *simultaneously* bounded, then we obtain a distance bound to $\theta^\star$. We can bound the former at *fixed $w$* by setting $\theta$ to minimize $F_w$, and the latter at *fixed $\theta$* by filtering $w$ via FastCovFilter. This introduces a chicken-and-egg problem of obtaining both bounds at once; we accomplish this via an alternating approach, and argue fast termination using a *third* potential monotone under both subroutines, the function value $F_w(\theta)$.

**Halving the radius.** We design a procedure, HalfRadiusLinReg, with the following guarantee. Suppose $\bar{w} \in \Delta^n$ is $\epsilon$-saturated and $\bar{\theta} \in \mathbb{R}^d$, and we know $R = \Omega(\sigma)$ with the promise $\|\bar{\theta} - \theta^\star\|_{\mathbf{\Sigma}^\star} \leq R$. HalfRadiusLinReg then returns a new $\theta$ with $\|\theta - \theta^\star\|_{\mathbf{\Sigma}^\star} \leq \frac{1}{2}R$. We do so by using saturated weights to guide a potential analysis via Proposition 2. Before stating HalfRadiusLinReg, we give subroutines achieving our earlier stated goals. The first is an approximate optimizer.

**Definition 1.** We call $\mathcal{O}_{\mathrm{ERM}}$ a $\gamma$-approximate ERM oracle if on input $F : \mathbb{R}^d \to \mathbb{R}$ it returns a point $\hat{\theta}$ such that $F(\hat{\theta}) - F(\theta_F^\star) \leq \gamma$, for $\theta_F^\star := \mathrm{argmin}_{\theta \in \mathbb{R}^d} F(\theta)$.

The second controls the initial error, which we use to yield distance bounds via strong convexity.

**Lemma 6** (Lemma 6, supplement). FunctionFilter *takes as input $\epsilon$-saturated $w \in \Delta^n$, $\theta \in \mathbb{R}^d$, and $R \geq \|\theta - \theta^\star\|_{\mathbf{\Sigma}^\star}$, and produces $\epsilon$-saturated $w' \in \Delta^n$ with $F_{w'}(\theta) \leq 2C_{\mathrm{ub}}(\sigma^2 + R^2)$, in time $\widetilde{O}(nd)$.*

FunctionFilter is a straightforward implementation of the filtering paradigm using Assumption 1.3, and we formally define it in the supplementary material. We can now state (a simplified version of) HalfRadiusLinReg, which iteratively filters weights initially set to uniform. It calls FunctionFilter to preprocess, and then alternates calls to $\mathcal{O}_{\mathrm{ERM}}$ and FastCovFilter, which respectively decrease the gradient norm and operator norm in Proposition 1. It finally terminates when the function value decrease between consecutive iterations is small, whence we conclude an $\mathcal{O}_{\mathrm{ERM}}$ call was not actually necessary for Proposition 2 to imply progress.

**Lemma 7** (Lemma 7, supplement). HalfRadiusLinReg *correctly returns $(w, \theta)$ such that $w \in \Delta^n$ is $\epsilon$-saturated and $\|\theta - \theta^\star\|_{\mathbf{\Sigma}^\star} \leq \frac{1}{2}R$ in $\kappa$ calls to $\mathcal{O}_{\mathrm{ERM}}$ and $\widetilde{O}(nd\kappa)$ additional time.*

---

**Algorithm 2** HalfRadiusLinReg($\mathbf{X}, y, \bar{w}, \bar{\theta}, R, \mathcal{O}_{\mathrm{ERM}}$)

---

1: **Input:** Dataset $\mathbf{X} = \{X_i\}_{i \in [n]} \in \mathbb{R}^{n \times d}$, $y = \{y_i\}_{i \in [n]} \in \mathbb{R}^n$, satisfying Assumption 1, $\epsilon$-saturated $\bar{w} \in \Delta^n$ with respect to bipartition $[n] = G \cup B$ with $\mathbf{X}^\top \mathbf{diag}(\bar{w}) \mathbf{X} \preceq 8L\mathbf{I}$, $\bar{\theta} \in \mathbb{R}^d$ with $\|\bar{\theta} - \theta^\star\|_{\mathbf{\Sigma}^\star} \leq R$ for $R = \Omega(\sigma)$, $O(\frac{\sigma^2}{\kappa})$-approximate ERM oracle $\mathcal{O}_{\mathrm{ERM}}$.
2: **Output:** Saturated $w$ with respect to $G \cup B$, and $\theta$ with $\|\theta - \theta^\star\|_{\mathbf{\Sigma}^\star} \leq \frac{1}{2}R$.
3: $t \leftarrow 0$, $w^{(0)} \leftarrow \mathsf{FunctionFilter}(w, \bar{\theta}, R)$, $\Delta_0 \leftarrow \infty$
4: **while** $\Delta_t = \Omega(\frac{R^2}{\kappa})$ **do**
5: $\quad \theta^{(t)} \leftarrow \mathcal{O}_{\mathrm{ERM}}(F_{w^{(t)}})$
6: $\quad w^{(t+1)} \leftarrow \mathsf{FastCovFilter}(\{g_i(\theta^{(t)})\}_{i \in [n]}, w^{(t)}, O(LR^2))$
7: $\quad \Delta_{t+1} \leftarrow F_{w^{(t+1)}}(\theta^{(t)}) - F_{w^{(t+1)}}(\theta^{(t+1)})$
8: $\quad t \leftarrow t + 1$
9: **end while**
10: **return** $(w^{(t)}, \theta^{(t-1)})$

---

We prove Lemma 7 by using the proof strategy outlined earlier: first, strong convexity, our initial function error bound, and monotonicity of function error ensure that $\|\theta^{(t)} - \theta^\star\|_{\mathbf{\Sigma}^\star} = O(R)$ throughout. Next, if $\Delta_t$ is ever too small, the gradient norm in Proposition 2 was small even before we ran the loop, so we could have terminated with the prior $\theta^{(t)}$. Finally, the whole algorithm runs $O(\kappa)$ times, since each loop decreases function error (initially at $O(R^2)$) substantially.

**Putting it all together.** We iteratively apply Lemma 7 $O(\log \frac{R_0}{\sigma})$ times, at which point we are within distance $O(\sigma)$ in the $\mathbf{\Sigma}^\star$ norm. To achieve distance $O(\sigma\sqrt{\kappa\epsilon})$, the natural bottleneck of Proposition 2, we define LastPhase, a variant of Algorithm 2 with a more stringent termination condition in Line 4, and use a similar argument to show it iterates $O(\frac{1}{\epsilon})$ times before concluding. Finally, we verify that throughout we only filtered against at most $O(\kappa \log \frac{R_0}{\sigma} + \frac{1}{\epsilon})$ distinct sets, where the summands are due to calls to HalfRadiusLinReg and LastPhase respectively. This fulfills the premise of using $\frac{\alpha}{2\epsilon}$ distinct sets for the $\alpha$ defined at the beginning of the section, and concludes our analysis.

### Acknowledgments

KT is supported by NSF Grant CCF-1955039 and the Alfred P. Sloan Foundation.

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
