# Robust Regression Revisited:
# Acceleration and Improved Estimation Rates

Arun Jambulapati[*]     Jerry Li[†]     Tselil Schramm[‡]     Kevin Tian[§]

## Abstract

We study fast algorithms for statistical regression problems under the strong contamination model, where the goal is to approximately optimize a generalized linear model (GLM) given adversarially corrupted samples. Prior works in this line of research were based on the *robust gradient descent* framework of [PSBR20], a first-order method using biased gradient queries, or the *Sever* framework of [DKK+19], an iterative outlier-removal method calling a stationary point finder.

We present nearly-linear time algorithms for robust regression problems with improved runtime or estimation guarantees compared to the state-of-the-art. For the general case of smooth GLMs (e.g. logistic regression), we show that the robust gradient descent framework of [PSBR20] can be *accelerated*, and show our algorithm extends to optimizing the Moreau envelopes of Lipschitz GLMs (e.g. support vector machines), answering several open questions in the literature.

For the well-studied case of robust linear regression, we present an alternative approach obtaining improved estimation rates over prior nearly-linear time algorithms. Interestingly, our method starts with an identifiability proof introduced in the context of the sum-of-squares algorithm of [BP21], which achieved optimal error rates while requiring large polynomial runtime and sample complexity. We reinterpret their proof within the Sever framework and obtain a dramatically faster and more sample-efficient algorithm under fewer distributional assumptions.

---

[*]Stanford University, `jmblpati@stanford.edu`
[†]Microsoft Research, `jerrl@microsoft.com`
[‡]Stanford University, `tselil@stanford.edu`
[§]Stanford University, `kjtian@stanford.edu`

# 1 Introduction

Parameter estimation in generalized linear models, such as linear and logistic regression problems, is among the most fundamental and well-studied statistical optimization problems. It serves as the primary workhorse in statistical studies arising from a variety of disciplines, ranging from economics [Smi12], biology [VGSM05], and the social sciences [Gor10]. Formally, given a *link function* $\gamma : \mathbb{R}^2 \to \mathbb{R}$ and a dataset of covariates and labels $\{(X_i, y_i)\}_{i \in [n]} \subset \mathbb{R}^d \times \mathbb{R}$ drawn from an underlying distribution $\mathcal{D}_{Xy}$, the problem of statistical (generalized linear) regression asks to

$$\text{estimate } \theta^\star := \operatorname{argmin}_{\theta \in \mathbb{R}^d} \left\{ \mathop{\mathbb{E}}_{(X,y) \sim \mathcal{D}_{Xy}} [\gamma(\langle \theta, X \rangle, y)] \right\}. \tag{1}$$

For example, when $\gamma(v, y) = \frac{1}{2}(v - y)^2$, (1) corresponds to (statistical) linear regression. The problem (1) also has an interpretation as computing a maximum likelihood estimate for a parameterized distributional model for data generation, and indeed is only tractable under certain distributional assumptions, since we only have access to samples from $\mathcal{D}_{Xy}$ rather than the underlying distribution itself (see e.g. [BP21] for tractability results in the linear regression setting).

However, in modern settings, these strong distributional assumptions may fail to hold. In practically relevant settings, regression is often performed on massive datasets, where the data comes from a poorly-understood distribution and has not been thoroughly vetted or cleaned of outliers. This has prompted the study of robust regression, i.e. regression under weak distributional or corruption assumptions. In this work, we study the problem of regression (1) in the *strong contamination model*. In this model, we assume the data points we receive are independently drawn from $\mathcal{D}_{Xy}$, but that an arbitrary $\epsilon$-fraction of the samples are then adversarially *contaminated* or replaced. The strong contamination model has recently drawn interest in the algorithmic statistics and learning communities for several reasons. Firstly, it is a *flexible* model of corruption and can be used to study both truly adversarial data poisoning attacks (where e.g. part of the dataset is sourced from malicious respondents), as well as model misspecification, where the generative $\mathcal{D}_{Xy}$ does not exactly satisfy our distributional assumptions, but is close in total variation to a distribution that does. Furthermore, a line of work building upon [DKK+16, LRV16] (discussed in our survey of prior work in Section 1.2) has achieved remarkable positive results for mean estimation and related problems under strong contamination, with statistical guarantees scaling independently of the dimension $d$. This dimension-free error promise is important in modern high-dimensional settings.

## 1.1 Our results

We give multiple *nearly-linear time algorithms*[1] for problem (1) under the strong contamination model, with improved statistical or runtime guarantees compared to the state-of-the-art. Prior algorithms for (1) under the strong contamination model typically followed one of two frameworks. The first, which we refer to as *robust gradient descent*, was pioneered by [PSBR20], and is based on reframing (1) as a problem where we have noisy gradient access to an unknown function we wish to optimize, coupled with the design of a noisy gradient oracle based on a robust mean estimation primitive. The second, which we refer to as *Sever*, originated in work of [DKK+19], and uses the guarantees of stationary point finders such as stochastic gradient descent to repeatedly perform outlier removal. In this work, we show that both approaches can be sped up dramatically, and give

---

[1]Throughout, we reserve the description "nearly-linear" for runtimes scaling linearly in the dataset size $nd$, and polynomially in $\epsilon^{-1}$ and the condition number, up to a polylogarithmic overhead in problem parameters.

two complementary types of algorithms within these frameworks.

**Robust acceleration.** Our first contribution is to demonstrate that within the noisy gradient estimation framework for minimizing well-conditioned regression problems of the form (1), an *accelerated* rate of optimization can be achieved, answering an open question asked by [PSBR20]. We demonstrate the following result for smooth statistical regression problems, where we assume the uncorrupted data is drawn from $\mathcal{D}_{Xy}$ with marginals $\mathcal{D}_X$ and $\mathcal{D}_y$, $\mathcal{D}_X$ has support in $\mathbb{R}^d$, and $\widetilde{O}$ hides polylogarithmic factors in problem parameters (cf. Section 2.1 for technical definitions). Finally, we remove $\kappa$ dependences in sample complexity statements, as they are subsumed by $\epsilon$ dependences due to assumed bounds on $\epsilon\kappa$ or $\epsilon\kappa^2$.

**Theorem 1** (informal, see Theorem 7). *Suppose $\gamma : \mathbb{R}^2 \to \mathbb{R}$ is such that $\gamma_y(v) := \gamma(v, y)$ is convex and has (absolute) first and second derivatives at most $1$ for all $y$ in the support of $\mathcal{D}_y$, and $\mathcal{D}_X$ has second moment matrix $\mathbf{\Sigma}^\star \preceq L \cdot \mathbf{I}$. For some $\mu \geq 0$, let $\kappa = \max(1, \frac{L}{\mu})$ and let*

$$\theta_{\text{reg}}^\star := \operatorname{argmin}_{\theta \in \mathbb{R}^d} \left\{ \mathop{\mathbb{E}}_{(X,y) \sim \mathcal{D}_{Xy}} \left\{ \gamma\left(\langle \theta, X \rangle, y\right) \right\} + \frac{\mu}{2} \|\theta\|_2^2 \right\}$$

*be the solution to the true regularized statistical regression problem.[2] There is an algorithm that given $n := \widetilde{O}(\frac{d}{\epsilon})$ $\epsilon$-corrupted samples from $\mathcal{D}_{Xy}$, for $\epsilon\kappa^2$ at most an absolute constant, runs in time $\widetilde{O}(nd\sqrt{\kappa})$ and obtains $\theta$ with $\|\theta - \theta_{\text{reg}}^\star\|_2 = O\left(\sqrt{\frac{\kappa\epsilon}{\mu}}\right)$ with probability at least $1 - \delta$.*

A canonical example of a link function $\gamma$ satisfying the assumptions of Theorem 1 is the logit function $\gamma(v, y) = \log(1 + \exp(-vy))$, when the labels $y$ are $\pm 1$. To contextualize Theorem 1, the earlier work [PSBR20] obtains a similar statistical guarantee in its setting, using $\widetilde{O}(\kappa)$ calls to a *noisy gradient oracle*, which they implement via a subroutine inspired by works on robust mean estimation. At the time of its initial dissemination, nearly-linear time robust mean estimation algorithms were not known; since then, [CAT+20] showed that for the case of linear regression (see Theorem 3 for the formal setup, as the linear regression link function is not Lipschitz), the framework was amenable to mean estimation techniques of [CDG19], and gave an algorithm running in time $\widetilde{O}(nd\kappa\epsilon^{-6})$. Theorem 1 represents an improvement to these results on two fronts: we apply tools from the mean-estimation algorithm of [DHL19] to remove the poly($\epsilon^{-1}$) runtime dependence for a general class of regression problems, and we achieve an iteration count of $\widetilde{O}(\sqrt{\kappa})$, matching the accelerated gradient descent runtime of [Nes83] for smooth optimization in the non-robust setting. We remark that the application of tools inspired by [DHL19] is fairly straightforward, and not a primary contribution of our work compared to the accelerated dependence on $\kappa$.

We demonstrate the generality of our acceleration framework by demonstrating that it applies to optimizing the *Moreau envelope* for Lipschitz, but possibly non-smooth, link functions $\gamma$; a canonical example of such a function is the hinge loss $\gamma(v, y) = \max(0, 1 - vy)$ with $\pm 1$ labels, used in training support vector machines. The Moreau envelope is a well-studied smooth approximation to a non-smooth function which everywhere additively approximates the original function if it is Lipschitz (see e.g. [Sho97]), and in the non-robust setting many state-of-the-art rates for Lipschitz optimization are known to be attained by accelerated optimization of an appropriate Moreau envelope [TJNO20]. We show that even without explicit access to the Moreau envelope, we can obtain

---

[2]To simplify our bounds and avoid estimation error for non-strongly convex statistical regression problems scaling with the initial search radius (which may be dimension-dependent), we focus on regularized regression problems. There is a substantial line of work on reductions between rates for strongly convex and convex smooth optimization in the non-robust setting, see e.g. [ZH16], and we defer an analogous exploration in the robust setting to future work.

approximate minimizers to it through our robust acceleration framework.

**Theorem 2** (informal, see Theorem 8). *Suppose $\gamma : \mathbb{R}^2 \to \mathbb{R}$ is such that $\gamma_y(v) := \gamma(v, y)$ is convex and has (absolute) first derivative at most 1 for all $y$ in the support of $\mathcal{D}_y$, and $\mathcal{D}_X$ has bounded second moment matrix. For some $\mu, \lambda \geq 0$, let $\kappa = \max(1, \frac{1}{\lambda\mu})$ and let*

$$\theta^\star_{\mathrm{env}} := \mathrm{argmin}_{\theta \in \mathbb{R}^d} \left\{ F^\star_\lambda(\theta) + \frac{\mu}{2} \|\theta\|^2_2 \right\}, \text{ where } F^\star_\lambda(\theta) := \inf_{\theta'} \left\{ F^\star(\theta') + \frac{1}{2\lambda} \|\theta - \theta'\|^2_2 \right\}$$

$$\text{is the Moreau envelope of } F^\star(\theta) := \mathop{\mathbb{E}}_{(X,y) \sim \mathcal{D}_{Xy}} \left\{ \gamma \left( \langle \theta, X \rangle, y \right) \right\}.$$

*There is an algorithm that given $n := \widetilde{O}(\frac{d}{\epsilon})$ $\epsilon$-corrupted samples from $\mathcal{D}_{Xy}$, for $\epsilon\kappa^2$ at most an absolute constant, runs in time $\widetilde{O}(\frac{nd\sqrt{\kappa}}{\epsilon})$ and obtains $\theta$ with $\|\theta - \theta^\star_{\mathrm{reg}}\|_2 = O\left(\sqrt{\frac{\kappa\epsilon}{\mu}}\right)$ with probability at least $1 - \delta$.*

To obtain this result, we give a nearly-linear time construction of a noisy gradient oracle for the Moreau envelope, which may be of independent interest; we note similar gradient oracle constructions in different settings have been developed in the optimization literature (see e.g. [CJJS21]).

**Robust linear regression.** The specific problem of robust linear regression is perhaps the most ubiquitous example of statistical regression [KKM18, KKK19, DKS19, ZJS20, CAT+20, BP21]. Amongst the algorithms developed for this problem, the only nearly-linear time algorithm is the recent work of [CAT+20]. For an instance of robust linear regression with noise variance bounded by $\sigma^2$ and covariate second moment matrix $\mathbf{\Sigma}^\star := \mathbb{E}_{X \sim \mathcal{D}_X}[XX^\top]$, the algorithms of [DKK+19, PSBR20, CAT+20] attain distance to the true regression minimizer $\theta^\star$ scaling as $\sigma\kappa\sqrt{\epsilon}$ in the $\mathbf{\Sigma}^\star$ norm (the "Mahalanobis distance") under a bounded 4th moment assumption. We measure error in the $\mathbf{\Sigma}^\star$ norm as it is scale invariant and the natural norm in which to measure the underlying (quadratic) statistical regression error.[3] We give one result (Theorem 3) which improves the runtime of [DKK+19, PSBR20, CAT+20] under the noisy gradient descent framework, and one result (Theorem 4) which improves its estimation rate, under the Sever framework.

We first demonstrate that directly applying our robust acceleration framework leads to a similar estimation guarantee as [DKK+19, PSBR20, CAT+20] under the same assumptions.

**Theorem 3** (informal, see Theorem 6). *Suppose $\mathcal{D}_X$ is a 2-to-4 hypercontractive distribution with second moment matrix $\mathbf{\Sigma}^\star = \mathbb{E}_{X \sim \mathcal{D}_X}[XX^\top]$ satisfying $\mu \cdot \mathbf{I} \preceq \mathbf{\Sigma}^\star \preceq L \cdot \mathbf{I}$, and $y \sim \mathcal{D}_y$ is generated as $\langle \theta^\star, X \rangle + \delta$, for $\delta \sim \mathcal{D}_\delta$ with variance at most $\sigma^2$ independent of $X$. Let $\kappa := \frac{L}{\mu}$. There is an algorithm, $\mathsf{RobustAccel}$, that given $n := \widetilde{O}(\frac{d}{\epsilon})$ $\epsilon$-corrupted samples from $\mathcal{D}_{Xy}$, for $\epsilon\kappa^2$ at most an absolute constant, runs in time $\widetilde{O}(nd\sqrt{\kappa})$ and obtains $\theta$ with $\|\theta - \theta^\star\|_{\mathbf{\Sigma}^\star} = O(\sigma\kappa\sqrt{\epsilon})$ with probability $1 - \delta$.*

We give a formal definition of 2-to-4 hypercontractivity in Section 2.1; as a lower bound of [BP21] shows, attaining estimation rates for robust linear regression scaling polynomially in $\epsilon$ is impossible under only bounded second moments, and such a 4th moment bound is the minimal assumption under which robust estimation is known to be possible. Theorem 6 matches the distribution assumptions and error of [CAT+20], while obtaining an accelerated runtime.

Interestingly, under the 4th moment bound used in Theorem 6, [BP21] showed that the information-theoretically optimal rate of estimation in the $\mathbf{\Sigma}^\star$ norm is *independent* of $\kappa$, and presented a match-

---

[3]Some prior works gave $\ell_2$ norm guarantees, which we have translated for comparison.

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

## 1.3 Techniques

We now describe the techniques we use to obtain the accelerated rates of Theorems 1, 2, and 3 as well as the robust linear regression algorithm of Theorem 4.

**Robust acceleration.** Our robust acceleration framework is based on the following abstract formulation of an optimization problem: there is an unknown function $F^\star : \mathbb{R}^d \to \mathbb{R}$ with minimizer $\theta^\star$ which is $L$-smooth and $\mu$-strongly convex, and we wish to estimate $\theta^\star$, but our only mode of accessing $F^\star$ is through a *noisy gradient oracle* $\mathcal{O}_{\mathrm{ng}}$. Namely, for some $\sigma$, $\epsilon$, we can query $\mathcal{O}_{\mathrm{ng}}$ at

any point $\theta \in \mathbb{R}^d$ with an upper bound $R \geq \|\theta - \theta^\star\|_2$ and receive an estimate $G(\theta)$ such that

$$\|G(\theta) - \nabla F^\star(\theta)\|_2 = O\left(\sqrt{L\epsilon}\sigma + L\sqrt{\epsilon}R\right). \tag{2}$$

In other words, we receive gradients perturbed by both fixed additive noise, and multiplicative noise depending on the distance to $\theta^\star$. The prior works [DKK⁺19, PSBR20, CAT⁺20] observed that by using tools from robust mean estimation, appropriate noisy gradient oracles could be constructed for the functions $F^\star(\theta) = \mathbb{E}_{(X,y)\sim\mathcal{D}_{Xy}}[\sigma(\langle\theta, X\rangle - y)]$ arising from the distributional assumptions in Theorems 1, 2, and 3. Our first contribution is speeding up the implementation of $\mathcal{O}_{\mathrm{ng}}$ to run in nearly-linear time $\widetilde{O}(nd)$, leveraging recent advances by [DHL19] for robust mean estimation.

Our second, and more technically involved, contribution is demonstrating that accelerated runtimes are achievable under the noisy gradient oracle access model of (2). Designing accelerated algorithms under noisy gradient access is an extremely well-studied problem, and there are both strong positive results [d'A08, MS13, DG16, CDO18, MRJ19, BJL⁺19] as well as negative results [DGN14] showing that under certain noise models, accelerated gradient descent may be outperformed by unaccelerated methods. Indeed, it was asked (motivated by these negative results) as an open question in [PSBR20] whether an accelerated rate was possible under the noise model (2).

Our accelerated algorithm runs in logarithmically many phases, where we halve the distance to the optimizer (while it is above a certain noise floor depending on the additive error in (2)) in each phase. The subroutine we design for implementing each phase is a robust accelerated "outer loop" tolerant to noisy gradient access in the manner provided by our oracle $\mathcal{O}_{\mathrm{ng}}$. By carefully balancing the accuracy of subproblem solutions, the multiplicative error in our gradient estimates within the accelerated outer loop, and the drift of the phase's iterates (which may venture further from $\theta^\star$ than our initial iterate upper bound), we show that above the noise floor we can halve the distance to $\theta^\star$ in $\widetilde{O}(\sqrt{\kappa})$ queries to $\mathcal{O}_{\mathrm{ng}}$; recursing on this guarantee yields our complete algorithm.

To obtain Theorem 2, we demonstrate that for Lipschitz functions $F^\star$ admitting a *radiusless* noisy gradient oracle, i.e. one which satisfies (2) with no dependence on $R$, we can further efficiently construct a noisy gradient oracle for the Moreau envelope of $F^\star$ using projected subgradient descent. This construction enables applying our robust accelerated method to Lipschitz regression problems.

Our acceleration framework crucially tolerates both additive and multiplicative guarantees for gradient estimation. While it is possible that arguments of other noisy acceleration frameworks e.g. [CDO18] may be extended to capture our gradient noise model, we give a self-contained derivation specialized to our specific oracle access for convenience. We view our result as a proof-of-concept that acceleration is possible under this noise model; we believe a unified study of acceleration under noise models encompassing (2) warrants further exploration, and defer it to interesting future work.

**Robust linear regression.** For the special case of linear regression, as discussed earlier, the robust optimization methods of [DKK⁺19, PSBR20, CAT⁺20] attain Mahalanobis distance scaling as $\sigma\kappa\sqrt{\epsilon}$ from the true minimizer. Directly plugging in deterministic conditions proven by [CAT⁺20] to hold under an appropriate statistical model into our robust gradient descent framework, we obtain a similar guarantee (Theorem 3) at an accelerated rate. In this technical overview, we now focus on how we obtain the improvements of Theorem 4.

At a high level, prior works lose two factors of $\sqrt{\kappa}$ in their error guarantees because of two *norm conversions* from the $\mathbf{\Sigma}^\star$ norm to the $\ell_2$ norm: one in gradient space, and one in parameter space. Because we do not have access to the true covariance $\mathbf{\Sigma}^\star$, it is natural to perform both gradient estimation and the gradient descent procedure itself in the $\ell_2$ norm. When the $\ell_2$ guarantees of

both subroutines are converted back to the $\mathbf{\Sigma}^\star$ norm, the error rate is lossy by a factor of $\kappa$.

We give a different approach to robust linear regression which bypasses this barrier in parameter space, saving a factor of $\sqrt{\kappa}$ in our error rate. In particular, we measure progress of our parameter estimates entirely in the $\mathbf{\Sigma}^\star$ norm in our analysis, which removes the need for an additional norm conversion. Our starting point is the following *identifiability proof* guarantee of [BP21], which we slightly repurpose for our needs. Let $w \in \Delta^n$ be entrywise less than $\frac{1}{n}\mathbb{1}$ such that $\|w_G\|_1 \geq 1 - 4\epsilon$ where $G$ is our "uncorrupted" data, and let $\theta \in \mathbb{R}^d$. We demonstrate in Proposition 6 that

$$\|\theta - \theta^\star\|_{\mathbf{\Sigma}^\star} = O\left(\sigma\sqrt{\kappa\epsilon} + \sqrt{\left\|\mathrm{Cov}_w\left(\{g_i(\theta)\}_{i\in[n]}\right)\right\|_{\mathrm{op}}\frac{\epsilon}{\mu}} + \sqrt{\epsilon}\left\|\nabla F_w(\theta)\right\|_{(\mathbf{\Sigma}^\star)^{-1}}\right). \qquad (3)$$

In the above display, $g_i(\theta)$ is the empirical gradient of the squared loss at our $i^{\text{th}}$ data point, $\mathrm{Cov}_w(\cdot)$ is the empirical second moment matrix of its argument under the weighting $w$, and $F_w$ is the empirical risk under $w$. The guarantee (3) suggests a natural approach to estimation: if we can *simultaneously* verify that $\theta$ is an approximate minimizer to $F_w$, and that the empirical second moment of gradients at $\theta$ (according to $w$) are small, then we have a proof that $\theta$ and $\theta^\star$ are close.

Prior work by [BP21] used this approach to obtain a polynomial-time estimator by solving a joint optimization problem in $(w, \theta)$, via an appropriate semidefinite program relaxation. However, in designing near-linear time algorithms, we cannot afford to use said relaxation. This raises a chicken-and-egg issue: for fixed $w$, it is straightforward to make $\|\nabla F_w(\theta)\|_{(\mathbf{\Sigma}^\star)^{-1}}$ small, by setting $\theta$ to the empirical risk minimizer (ERM) of $F_w$. Likewise, for fixed $\theta$, known *filtering* techniques rapidly decrease $\|\mathrm{Cov}_w(\{g_i(\theta)\})_{i\in[n]}\|_{\mathrm{op}}$ while preserving most of $w_G$, by using that the second moment restricted to uncorrupted points has a small operator norm as a certificate for outlier removal. However, performing either of these subroutines to guarantee one of our sufficient conditions passes (small operator norm or gradient norm) may adversely affect the quality of the other.

We circumvent this chicken-and-egg problem by introducing a *third* potential, namely the actual function value $F_w(\theta)$. In particular, notice that the two subroutines we described earlier (downweighting $w$ or setting $\theta$ to the ERM) both decrease this third potential. Our linear regression algorithm is an alternating procedure which iteratively filters $w$ based on the gradients at the current $\theta$ (to make the operator norm small), and sets $\theta$ to the ERM of $F_w$ (to zero out the gradient norm). We further show that if the ERM step does not make significant function progress (our third potential), then it was not strictly necessary to make progress according to (3), since the gradient norm was already small. This gives a dimension-independent bound on the number of times we could have alternated, via tracking function progress, yielding Theorem 4.

## 2 Preliminaries

We give the notation used throughout this paper in Section 2.1, and set up the statistical model we consider in Section 2.2. In Section 2.3, we give the deterministic regularity assumptions used by our regression algorithm in Section 3. In Section 2.4, we give the deterministic regularity assumptions used by our stochastic optimization algorithms in Sections 4 and 5. Finally, in Section 2.5, we state a nearly-linear time procedure for robustly decreasing the operator norm of the second moment matrix of a set of vectors. Some proofs are deferred to the appendices.

### 2.1 Notation

**General notation.** For $d \in \mathbb{N}$ we let $[d] := \{j \mid j \in \mathbb{N}, 1 \leq j \leq d\}$. The $\ell_p$ norm of a vector argument is denoted $\|\cdot\|_p$, where $\|\cdot\|_\infty$ is the element with largest absolute value; when the

argument is a symmetric matrix, we overload this to mean the Schatten-$p$ norm. The all-ones vector (of appropriate dimension from context) is denoted $\mathbb{1}$. The (solid) probability simplex is denoted $\Delta^n := \{w \in \mathbb{R}^n_{\geq 0}, \|w\|_1 \leq 1\}$. We use $\widetilde{O}$ to suppress logarithmic factors in dimensions, distance ratios, the problem condition number $\kappa$, the inverse corruption parameter $\epsilon^{-1}$, and the inverse failure probability. For $v \in \mathbb{R}^n$ and $S \subseteq [n]$, we let $v_S \in \mathbb{R}^n$ denote $v$ with coordinates in $[n] \setminus S$ zeroed out. For a set $S$, we call $S_1, S_2$ a *bipartition* of $S$ if $S_1 \cap S_2 = \emptyset$ and $S_1 \cup S_2 = S$.

**Matrices.** Matrices are denoted in boldface. We denote the zero and identity matrices (of appropriate dimension) by $\mathbf{0}$ and $\mathbf{I}$. The $d \times d$ symmetric matrices are $\mathbb{S}^d$, and the $d \times d$ positive semidefinite cone is $\mathbb{S}^d_{\geq 0}$. For $\mathbf{A}, \mathbf{B} \in \mathbb{S}^d$ we write $\mathbf{A} \preceq \mathbf{B}$ to mean $\mathbf{B} - \mathbf{A} \in \mathbb{S}^d_{\geq 0}$. The largest and smallest eigenvalue and trace of a symmetric matrix are respectively denoted $\lambda_{\max}(\cdot)$, $\lambda_{\min}(\cdot)$, and $\mathrm{Tr}(\cdot)$. The inner product on $\mathbb{S}^d$ is $\langle \mathbf{A}, \mathbf{B} \rangle := \mathrm{Tr}(\mathbf{AB})$. For positive definite $\mathbf{M}$, we define the induced norm $\|v\|_{\mathbf{M}} := \sqrt{v^\top \mathbf{M} v}$. We use $\|\cdot\|_{\mathrm{op}}$ to mean the $\ell_2$-$\ell_2$ operator norm of a matrix; when the argument is symmetric, it is synonymous with $\lambda_{\max}$, and otherwise is the largest singular value.

**Functions.** The gradient and Hessian of a twice-differentiable function are denoted $\nabla$ and $\nabla^2$. We say differentiable $f : \mathbb{R}^d \to \mathbb{R}$ is $\lambda$-Lipschitz in a quadratic norm $\|\cdot\|_{\mathbf{M}}$ if $\|\nabla f(\theta)\|_{\mathbf{M}^{-1}} \leq \lambda$ for all $\theta \in \mathbb{R}^d$. We say twice-differentiable $f : \mathbb{R}^d \to \mathbb{R}$ is $L$-smooth and $\mu$-strongly convex in $\|\cdot\|_{\mathbf{M}}$ if

$$\mu \mathbf{M} \preceq \nabla^2 f(\theta) \preceq L\mathbf{M}, \text{ for all } \theta \in \mathbb{R}^d.$$

When $\mathbf{M}$ is not specified, we assume $\mathbf{M} = \mathbf{I}$ (i.e. the norm in question is $\ell_2$). For any $\mathbf{M}$, smoothness and strong convexity imply the following bounds for all $\theta, \theta' \in \mathbb{R}^d$,

$$f(\theta) + \left\langle \nabla f(\theta), \theta' - \theta \right\rangle + \frac{\mu}{2} \left\| \theta' - \theta \right\|^2_{\mathbf{M}} \leq f(\theta') \leq f(\theta) + \left\langle \nabla f(\theta), \theta' - \theta \right\rangle + \frac{L}{2} \left\| \theta' - \theta \right\|^2_{\mathbf{M}}.$$

It is well-known that $L$-smoothness of function $f$ implies $L$-Lipschitzness of the function gradient $\nabla f$, i.e. $\|\nabla f(\theta) - \nabla f(\theta')\|_{\mathbf{M}^{-1}} \leq L \|\theta - \theta'\|_{\mathbf{M}}$ for all $\theta, \theta' \in \mathbb{R}^d$. For any $f$ which is $L$-smooth and $\mu$-strongly convex in $\|\cdot\|_{\mathbf{M}}$, with $\theta^* := \mathrm{argmin}_{\theta \in \mathbb{R}^d} f(\theta)$, it is straightforward to show

$$\frac{1}{2L} \|\nabla f(\theta)\|^2_{\mathbf{M}^{-1}} \leq f(\theta) - f(\theta^\star) \leq \frac{1}{2\mu} \|\nabla f(\theta)\|^2_{\mathbf{M}^{-1}} \text{ for all } \theta \in \mathbb{R}^d.$$

**Distributions.** The multivariate Gaussian distribution with mean $\mu$ and covariance $\mathbf{\Sigma}$ is denoted $\mathcal{N}(\mu, \mathbf{\Sigma})$. For weights $w \in \mathbb{R}^n_{\geq 0}$ and a set of vectors $\mathbf{X} := \{X_i\}_{i \in [n]}$, we let

$$\mu_w(\mathbf{X}) := \sum_{i \in [n]} \frac{w_i}{\|w\|_1} X_i, \ \mathrm{Cov}_{w, \bar{X}}(\mathbf{X}) := \sum_{i \in [n]} \frac{w_i}{\|w\|_1} \left( X_i - \bar{X} \right) \left( X_i - \bar{X} \right)^\top.$$

be the empirical mean and (centered) covariance matrix; when $\bar{X}$ is not specified, it is the zeroes vector. Draws from the uniform distribution on $\{X_i\}_{i \in [n]}$ are denoted $X \sim_{\mathrm{unif}} \mathbf{X}$. We say distribution $\mathcal{D}$ supported on $\mathbb{R}^d$ is 2-to-4 hypercontractive with parameter $C_{2 \to 4}$ if for all $v \in \mathbb{R}^d$,

$$\mathbb{E}_{X \sim \mathcal{D}} \left[ \langle X, v \rangle^4 \right] \leq C_{2 \to 4} \mathbb{E}_{X \sim \mathcal{D}} \left[ \langle X, v \rangle^2 \right]^2.$$

We will refer to this property as being $C_{2 \to 4}$-hypercontractive for short; by massaging the definition, we observe $C_{2 \to 4}$-hypercontractivity is preserved under linear transformations of the distribution.

**Filtering.** We will make much use of the following algorithmic technique, which refer to as *filtering*. In the filtering paradigm, we have an index set $[n]$, and a fixed, unknown bipartition $[n] = G \cup B$, $G \cap B = \emptyset$. The set $G$ is a "good" set of indices that we wish to keep, and the set $B$ is a set of "bad" indices which we would like to remove. The algorithm maintains a set of weights $w \in \Delta^n$ (with the goal of producing a weight vector which is close to the uniform distribution on $G$). These weights are iteratively updated according to "scores" $\tau \in \mathbb{R}^n_{\geq 0}$; the goal of filtering is to assign large scores to coordinates in $B$ and small scores to coordinates in $G$, so that the bad coordinates can be filtered out according to their scores. Concretely, we use the following definition.

**Definition 1** (saturated weights)**.** We say weights $w \in \Delta^n$ are *c-saturated* with respect to the bipartition $G \cup B = [n]$ if $w \leq \frac{1}{n} \mathbb{1}$ entrywise, and

$$\left\| \left[ \frac{1}{n} \mathbb{1} - w \right]_G \right\|_1 \leq \left\| \left[ \frac{1}{n} \mathbb{1} - w \right]_B \right\|_1 + c.$$

If $c = 0$, we refer to $w$ as simply *saturated*.

In words, $w$ is saturated if its difference from the uniform distribution has more weight on $B$ than $G$ (in the context of our algorithm, if we have started with uniform weights and produced a saturated $w$, then we have removed more mass from $B$ than $G$). We allow for a "fudge factor" of an additive $c$ to relax the above definition, which will come in handy in our linear regression applications.

**Definition 2** (safe scores)**.** Suppose $G \cup B$ is a bipartition of $[n]$, and suppose $w \in \Delta^n$ is a set of weights. We call a set of scores $\tau = \{\tau_i\}_{i \in [n]} \in \mathbb{R}^n_{\geq 0}$ *safe with respect to $w$* if it satisfies

$$\langle w_G, \tau \rangle \leq \langle w_B, \tau \rangle .$$

The following simple lemma (implicit in prior works [DKK$^+$17, CSV17b, Li18, Ste18]) is the crux of the filtering paradigm, relating these two definitions.

**Lemma 1.** *Suppose $w \in \Delta^n$ is saturated, and $\tau \in \mathbb{R}^n_{\geq 0}$ is safe with respect to $w$. Defining $w'$ by*

$$w'_i \leftarrow \left( 1 - \frac{\tau_i}{\tau_{\max}} \right) w_i \text{ for all } i \in [n], \text{ and } \tau_{\max} := \max_{i \in [n] | w_i \neq 0} \tau_i,$$

*then $w' \in \Delta^n$ is also saturated.*

*Proof.* If $G \cup B = [n]$ is the bipartition with good coordinates $G$, then by definition of safe scores,

$$\left\| [w - w']_G \right\|_1 = \frac{1}{\tau_{\max}} \langle w_G, \tau \rangle \leq \frac{1}{\tau_{\max}} \langle w_G, \tau \rangle \leq \left\| [w - w']_B \right\|_1 .$$

Now, since $w$ is saturated, and since $w' \leq w \leq \frac{1}{n} \mathbb{1}$ by definition of saturation,

$$\left\| \left[\tfrac{1}{n} \mathbb{1} - w'\right]_G \right\|_1 = \left\| \left[\tfrac{1}{n} \mathbb{1} - w\right]_G \right\|_1 + \left\| [w - w']_G \right\|_1 \leq \left\| \left[\tfrac{1}{n} \mathbb{1} - w\right]_B \right\|_1 + \left\| [w - w']_B \right\|_1 = \left\| \left[\tfrac{1}{n} \mathbb{1} - w'\right]_G \right\|_1 .$$

$\square$

We will also frequently using the following simple fact.

**Lemma 2.** *Suppose $w \in \Delta^n$ is c-saturated with respect to bipartition $[n] = G \cup B$, and suppose the bad set $|B| \leq \epsilon n$. Let $\tilde{w} = \frac{w}{\|w\|_1}$ be the distribution with probabilities proportional to $w$ and $w_G^\star = \frac{1}{|G|}\mathbb{1}_G$ be uniform over $G$. Then, $\|\tilde{w} - w_G^\star\|_1 \leq 6\epsilon + 2c$.*

*Proof.* By the definition of saturation, since there is only $\epsilon$ mass to remove from the coordinates of $B$ on $\frac{1}{n}\mathbb{1}$, clearly $\|w\|_1 \geq 1 - 2\epsilon - c$. By the triangle inequality, we have

$$\|\tilde{w} - w_G^\star\|_1 \leq \|\tilde{w} - w\|_1 + \left\|w - \tfrac{1}{n}\mathbb{1}\right\|_1 + \left\|\tfrac{1}{n}\mathbb{1} - w_G^\star\right\|_1.$$

By definition of $\tilde{w}$, $\|\tilde{w} - w\|_1 = 1 - \|w\|_1 \leq 2\epsilon + c$. By saturation, $\left\|w - \frac{1}{n}\mathbb{1}\right\|_1 \leq 2\epsilon + c$. Finally, since $|G| \geq (1 - \epsilon)n$, $\left\|\frac{1}{n}\mathbb{1} - w_G^\star\right\|_1 \leq 2\epsilon$. Combining these pieces yields the claim. $\qquad\square$

## 2.2 Our statistical models

In this paper, we provide provable guarantees for optimization problems captured by the following statistical model.

**Model 1** (stochastic optimization in the strong contamination model)**.** For $\mathcal{D}_f$ a distribution over functions $f : \mathbb{R}^d \to \mathbb{R}$, our goal is to optimize $\mathbb{E}_{f \sim \mathcal{D}_f}[f(\theta)]$. We are given access to $n$ samples $\{f_i\}_{i \in [n]}$ produced as follows:

1. Functions $\{\tilde{f}_i\}_{i \in [n]}$ are drawn independently from $\mathcal{D}_f$.
2. An arbitrary subset $B \subset [n]$ of the samples is replaced with arbitrary functions from $\mathrm{supp}(\mathcal{D}_f)$.
3. For each $i \in [n]$, if $i \in B$ we observe $f_i$ as the corrupted sample, otherwise, we observe $f_i = \tilde{f}_i$.

We call $B$ the *corrupted* samples. When $|B| = \epsilon n$, we say $\{f_i\}_{i \in [n]}$ is drawn $\epsilon$-*corrupted from* $\mathcal{D}_f$.

Throughout we use the convention that $G \cup B = [n]$ is the bipartition of sample coordinates with $B$ the corrupted samples and $G$ the "good" samples. For simplicity, we assume throughout that $\epsilon = \frac{|B|}{n}$ is smaller than some globally fixed constant. We will also frequently use the notation $g_i$ to mean $\nabla f_i$ for all $i \in [n]$. When $\mathcal{D}_f$ is clear from context, we denote the "true average function" by

$$F^\star(\theta) := \mathbb{E}_{f \sim \mathcal{D}_f}[f(\theta)].$$

For $w \in \Delta^n$ and functions $\{f_i\}_{i \in [n]}$, the (unnormalized) weighted empirical average function is

$$F_w(\theta) := \sum_{i \in [n]} w_i f_i(\theta). \qquad (4)$$

We will use $w_G^\star$ to denote the uniform distribution over $G$, $w_G^\star := \frac{1}{|G|}\mathbb{1}_G$, and we use $F_G$ as shorthand for the function $F_{w_G^\star}$. The goal of robust parameter estimation is to estimate the true optimizer, which we always denote by $\theta^\star := \mathrm{argmin}_{\theta \in \mathbb{R}^d} F^\star(\theta)$. For example, the problem estimating the mean of $\mathcal{D}$ can be expressed by choosing $f_i(\theta) = \frac{1}{2}\|\theta - X_i\|_2^2$ for $X_i \sim \mathcal{D}$. In the uncorrupted setting (i.e. $\epsilon = 0$), a typical strategy (given reasonable regularity assumptions on $\mathcal{D}_f$) is to choose the estimator $\theta_G := \mathrm{argmin}_{\theta \in \mathbb{R}^d} F_G(\theta)$. The challenge is to obtain comparable estimation performance to $\theta_G$ which is robust to an $\epsilon$-fraction of unknown corruptions.

Our focus in this paper is optimizing *generalized linear models*. In particular, throughout we will work only with $\mathcal{D}_f$ of the following form.

**Model 2** (generalized linear model). A *generalized linear model* is a distribution over functions $f_i : \mathbb{R}^d \to \mathbb{R}$ which is defined by a joint distribution $\mathcal{D}_{Xy}$ over pairs $\{X_i, y_i\} \in \mathbb{R}^d \times \mathbb{R}$ and a *link function* $\gamma : \mathbb{R}^2 \to \mathbb{R}$, so that samples $f_i \sim \mathcal{D}_f$ are generated as

$$f_i(\theta) := \gamma\left(\langle X_i, \theta \rangle, y_i\right), \quad \text{for } (X_i, y_i) \sim \mathcal{D}_{Xy}. \tag{5}$$

Note that observing $\{f_i\}_{i \in [n]}$ is equivalent to observing the dataset $\{X_i, y_i\}_{i \in [n]}$ when $\gamma$ is known.

For instance, when $\gamma(v, y) = \frac{1}{2}(v - y)^2$, this is the problem of (statistical) linear regression. Further, when $\gamma(v, y) = \log(1 + \exp(-vy))$, our problem is logistic regression, and when $\gamma(v, y) = \max(0, 1 - vy)$, it is fitting a support vector machine. We refer to the $X$ and $y$ marginals over $\mathcal{D}_{Xy}$ respectively by $\mathcal{D}_X$ and $\mathcal{D}_y$, and we denote $\boldsymbol{\Sigma}^\star := \mathbb{E}_{X \sim \mathcal{D}_X}\left[XX^\top\right]$ when $\mathcal{D}_X$ is clear from context.

### 2.3 Linear regression

In Section 3 and (part of) Section 4, we develop algorithms for the well-studied special case of the generalized linear model, Model 2 wherein $\gamma(v, y) = \frac{1}{2}(v - y)^2$, i.e. a statistical variant of linear regression. We obtain guarantees under the following model and regularity assumptions for $\mathcal{D}_{Xy}$.

**Model 3** (distributional regularity for linear regression). Given distributions $\mathcal{D}_X$ and $\mathcal{D}_\delta$ over $\mathbb{R}^d, \mathbb{R}^d$ respectively, the distribution $\mathcal{D}_{Xy}$ over $\mathbb{R}^d \times \mathbb{R}$ is sampled as follows: for an underlying vector $\theta^\star \in \mathbb{R}^d$, independently sample $X \sim \mathcal{D}_X$ and $\delta \sim \mathcal{D}_\delta$, and set $y \leftarrow \langle \theta^\star, X \rangle + \delta$. Further, $\mathcal{D}_X$ and $\mathcal{D}_\delta$ satisfy the following regularity assumptions.

1. For $\boldsymbol{\Sigma}^\star = \mathbb{E}_{\mathcal{D}_X}[XX^\top]$ and $0 < \mu < L$, we have $\mu\mathbf{I} \preceq \boldsymbol{\Sigma}^\star \preceq L\mathbf{I}$.
2. $\mathcal{D}_X$ is $C_{2 \to 4}$-hypercontractive for a constant $C_{2 \to 4}$.
3. $\mathcal{D}_\delta$ is a $C_{2 \to 4}$-hypercontractive distribution with mean zero and variance $\leq \sigma^2$.

For $\{(X_i, y_i)\}_{i \in [n]}$ (in particular, overloading to include $i \in B$), we use the notation $\delta_i := y_i - \langle X_i, \theta^\star \rangle$. We also use the following notation when discussing linear regression:

$$f_i(\theta) := \frac{1}{2}\left(\langle X_i, \theta \rangle - y_i\right)^2, \; g_i(\theta) := \nabla f_i(\theta) = X_i\left(\langle X_i, \theta \rangle - y_i\right). \tag{6}$$

We will denote the condition number of $\boldsymbol{\Sigma}^\star$ by $\kappa := \frac{L}{\mu}$ throughout. Under Model 3, it is immediate from the first-order optimality condition that for

$$F^\star(\theta) := \underset{X, y \sim \mathcal{D}_{Xy}}{\mathbb{E}}\left[\frac{1}{2}\left(\langle X, \theta \rangle - y\right)\right]^2,$$

the optimizer $\operatorname{argmin}_{\theta \in \mathbb{R}^d} F^\star(\theta)$ is exactly $\theta^\star$.

In our setting, following the description in Section 2.2 we independently draw $\{(X_i, y_i)\}_{i \in G} \sim \mathcal{D}_{Xy}$ for $|G| = (1 - \epsilon)n$ and observe $\{(X_i, y_i)\}_{i \in [n]}$ where $[n] = G \cup B$ and $\{(X_i, y_i)\}_{i \in B}$ are arbitrarily chosen. We will frequently refer to $\{X_i\}_{i \in [n]}$ as $\mathbf{X} \in \mathbb{R}^{n \times d}$. Under Model 3, recent work [BP21] obtained the following results.

**Proposition 1** ([BP21], Theorem 1.7, Theorem 1.9, Theorem 1.2). For Models 1 and 3, the minimax optimal error rate for estimators $\hat{\theta}$ is

$$\left\|\hat{\theta} - \theta^\star\right\|_{\boldsymbol{\Sigma}^\star} = O\left(\sigma\epsilon^{\frac{3}{4}}\right).$$

When the distribution $\mathcal{D}_X$ is further *certifiably hypercontractive* in the sum-of-squares proof system, there is a poly($d$)-time estimator requiring poly($d$) samples achieving this rate with high probability. Moreover, without the hypercontractivity condition in Model 3, even when $\mu, L = \Theta(1)$ it is information-theoretically impossible to attain an error rate depending polynomially on $\epsilon$.

The algorithmic result of [BP21] (and all known techniques with error rate $\ll \sqrt{\epsilon}$) crucially requires that the distributions are sum-of-squares certifiably hypercontractive, which is a stronger assumption than (standard) hypercontractivity. There is evidence that the problem of certifying $2 \to 4$ hypercontractivity is computationally intractable in general (under e.g. the small-set expansion hypothesis, see [BBH+12, BGG+19]). Even for certifiably hypercontractive distributions, known algorithms require use spectral estimators of higher-order moment matrices, and thus more samples and increased runtime complexity. Hence, error $\sqrt{\epsilon}$ is a standing barrier for fast algorithms.

In Section 3, whenever we discuss robust linear regression we operate in Models 1 and 3. These assumptions imply that the data $\{X_i, y_i\}_{i\in[n]}$ will satisfy the following deterministic conditions with probability $\frac{9}{10}$. For convenience we work with these deterministic conditions directly in our proofs.

**Assumption 1** (deterministic regularity for linear regression). Let $\epsilon$ be sufficiently small, and let $r \in (0, \epsilon^2)$. Assume $\{(X_i, y_i)\}_{i\in[n]} \subset \mathbb{R}^d \times \mathbb{R}$ is $(\epsilon, r)$-*good for linear regression* (or $(\epsilon, r)$-good if context is clear), which means there is a partition $[n] = G \cup B$ with $|G| \geq (1 - \epsilon)n$ which satisfies:

1. For any $w \in \Delta^n$ with $\|w_G\|_1 \geq 1 - 4\epsilon$, $\frac{1}{2}\mathbf{\Sigma}^\star \preceq \mathrm{Cov}_{w_G}(\mathbf{X}) \preceq \frac{3}{2}\mathbf{\Sigma}^\star$.

2. There is a constant $C_{\mathrm{est}}$ such that for all $\theta \in \mathbb{R}^d$, there exists a $G' \subseteq G$ satisfying $|G'| \geq (1 - r)|G|$ such that for all $\epsilon$-saturated $w \in \Delta^n$, if we let $\tilde{w} := \frac{w_{G'}}{\|w_{G'}\|_1}$,

$$\|\nabla F_{\tilde{w}}(\theta) - \nabla F^\star(\theta)\|_2 \leq C_{\mathrm{est}}\sqrt{L\epsilon}\left(\sigma + \|\theta - \theta^\star\|_{\mathbf{\Sigma}^\star}\right), \tag{7}$$

$$\left\|\mathrm{Cov}_{\tilde{w}}\left(\{g_i(\theta)\}_{i\in G'}\right)\right\|_{\mathrm{op}} \leq C_{\mathrm{est}}L\left(\|\theta - \theta^\star\|_{\mathbf{\Sigma}^\star}^2 + \sigma^2\right). \tag{8}$$

3. There is a constant $C_{\mathrm{ub}}$ such that

$$\sum_{i\in G}\frac{1}{|G|}f_i(\theta^\star) = \frac{1}{2|G|}\sum_{i\in G}\left(\langle X_i, \theta^\star\rangle - y_i\right)^2 \leq C_{\mathrm{ub}}\sigma^2.$$

We defer the proof of the following claim, which establishes the probabilistic validity of Assumption 1 (up to adjusting constants in definitions) under the statistical Models 1 and 3, to Appendix A.

**Proposition 2.** Let $\alpha \geq 1$ and let $\epsilon > 0$ be sufficiently small. Let $\{(X_i, y_i)\}_{i\in[n]} \subset \mathbb{R}^d \times \mathbb{R}$ be an $\epsilon$-corrupted set of samples from a distribution $\mathcal{D}_{Xy}$ as in Model 3. Then, if

$$n = O\left(\frac{d\alpha^2 \log d}{\epsilon^4} + \frac{d^2\alpha^{1.5}\log(d/\epsilon)}{\epsilon^3}\right),$$

the set $\{(X_i, y_i)\}_{i\in[n]}$ is $(2\epsilon, \frac{\epsilon^2}{\alpha})$-good for linear regression with probability at least $\frac{9}{10}$.

**Remark 1.** *We remark that the gaurantees of Proposition 2 may be recovered with $n = O(d\log d/\epsilon^4 + d^2\log d\log\frac{1}{\epsilon})$ samples when the $X_i$ are further assumed to be subgaussian.*

We observe that Assumption 1 implies the following useful bound.

**Lemma 3.** *Let $w \in \Delta^n$ be $\epsilon$-saturated with respect to bipartition $[n] = G \cup B$, let $G^\star \subseteq G$ be the subset in Assumption 1.2 corresponding to $\theta^\star$, and let $\tilde{w} = \frac{w_{G^\star}}{\|w_{G^\star}\|_1}$. Let $\theta_{\tilde{w}} = \mathrm{argmin}_{\theta \in \mathbb{R}^d} F_{\tilde{w}}(\theta)$ be the empirical minimizer of $F_{\tilde{w}}$. Then,*

$$\|\theta_{\tilde{w}} - \theta^\star\|_{\mathbf{\Sigma}^\star} \leq 4C_{\mathrm{est}}\sigma\sqrt{\kappa\epsilon}.$$

*Proof.* Applying Assumption 1.2 with $\theta = \theta^\star$, we have that

$$\|\nabla F_{\tilde{w}}(\theta^\star)\|_2 \leq C_{\mathrm{est}}\sqrt{L\epsilon}\sigma.$$

However, applying Assumption 1.1 on the weights $w_{G^\star}$ implies that $F_{\tilde{w}}$ is $\frac{1}{2}$-strongly convex in the $\mathbf{\Sigma}^\star$ norm (since $w_{G^\star}$ removes at most $\epsilon^2$ mass from $w_G$) and minimized by $\theta_{\tilde{w}}$, and hence by using consequences of strong convexity and $(\mathbf{\Sigma}^\star)^{-1} \preceq \frac{1}{\mu}\mathbf{I}$,

$$\|\nabla F_{\tilde{w}}(\theta^\star)\|_2 \geq \sqrt{\mu}\|\nabla F_{\tilde{w}}(\theta^\star)\|_{(\mathbf{\Sigma}^\star)^{-1}} \geq \frac{\sqrt{\mu}}{4}\|\theta_{\tilde{w}} - \theta^\star\|_{\mathbf{\Sigma}^\star}.$$

$\square$

Finally, in Section 4, when we develop an alternative approach to linear regression based on the robust gradient descent framework, we require a slightly weaker set of distributional assumptions and deterministic implications, which we now state.

**Model 4** (distributional regularity for linear regression, gradient descent setting)**.** Given distributions $\mathcal{D}_X$ and $\mathcal{D}_\delta$ over $\mathbb{R}^d, \mathbb{R}^d$ respectively, the distribution $\mathcal{D}_{Xy}$ over $\mathbb{R}^d \times \mathbb{R}$ is sampled as follows: for an underlying vector $\theta^\star \in \mathbb{R}^d$, independently sample $X \sim \mathcal{D}_X$ and $\delta \sim \mathcal{D}_\delta$, and set $y \leftarrow \langle \theta^\star, X \rangle + \delta$. Further, $\mathcal{D}_X$ and $\mathcal{D}_\delta$ satisfy the following regularity assumptions.

1. For $\mathbf{\Sigma}^\star = \mathbb{E}_{\mathcal{D}_X}[XX^\top]$ and $0 < \mu < L$, we have $\mu\mathbf{I} \preceq \mathbf{\Sigma}^\star \preceq L\mathbf{I}$.
2. $\mathcal{D}_X$ is $C_{2\to4}$-hypercontractive for a constant $C_{2\to4}$.
3. $\mathcal{D}_\delta$ is a distribution with mean zero and variance $\leq \sigma^2$.

The main difference between Model 3 and Model 4 is that the latter no longer requires hypercontractive noise. This corresponds to the following deterministic assumption.

**Assumption 2** (deterministic regularity for linear regression, gradient descent setting)**.** Let $\epsilon$ be sufficiently small. For every fixed $\theta \in \mathbb{R}^d$, there is a partition $[n] = G_\theta \cup B_\theta$ with $|G_\theta| \geq (1-\epsilon)n$ which satisfies: there is a constant $C_{\mathrm{est}}$ such that for $\tilde{w} := \frac{w_{G_\theta}}{\|w_{G_\theta}\|_1}$,

$$\|\nabla F_{\tilde{w}}(\theta) - \nabla F^\star(\theta)\|_2 \leq C_{\mathrm{est}}\sqrt{L\epsilon}\left(\sigma + \|\theta - \theta^\star\|_{\mathbf{\Sigma}^\star}\right), \tag{9}$$

$$\left\|\mathrm{Cov}_{\tilde{w}}\left(\{g_i(\theta)\}_{i \in G_\theta}\right)\right\|_{\mathrm{op}} \leq C_{\mathrm{est}}L\left(\|\theta - \theta^\star\|_{\mathbf{\Sigma}^\star}^2 + \sigma^2\right). \tag{10}$$

The main difference between Assumption 1 and Assumption 2 is that the latter provides gradient bounds using a different set $G_\theta$ for each $\theta$ (as opposed to the former, which uses the same set $G$ for all $\theta$). The upshot is that the corresponding required sample complexity is lower.

**Proposition 3** ([CAT⁺20], Lemma 5.5)**.** *Let $\epsilon > 0$ be sufficiently small. Let $\{(X_i, y_i)\}_{i \in [n]} \subset \mathbb{R}^d \times \mathbb{R}$ be an $O(\epsilon)$-corrupted set of samples from a distribution $\mathcal{D}_{Xy}$ as in Model 2. Then if*

$$n = O\left(\frac{d\log(d/\epsilon)}{\epsilon}\right),$$

Assumption 2 holds with probability at least $\frac{9}{10}$.

## 2.4 Regularity assumptions: Lipschitz and smooth stochastic optimization

In Section 4, we develop algorithms for the special case of Model 2 when all $\gamma_{y_i}(v) := \gamma(v, y_i)$, as viewed as a function of its first variable, satisfy

$$\left|\gamma'_{y_i}(v)\right| \leq 1,\ 0 \leq \gamma''_{y_i}(v) \leq 1 \text{ for all } v \in \mathbb{R},\ i \in [n].$$

In other words, all $\gamma_{y_i}$ are 1-smooth and 1-Lipschitz. A canonical example is when all $y_i = \pm 1$ are positive or negative labels, and $\gamma$ is the logistic loss function, $\gamma(v, y) = \log(1 + \exp(-vy))$. In this setting, we will focus on approximating the (population) regularized optimizer,

$$\theta^\star_{\text{reg}} := \operatorname{argmin}_{\theta \in \mathbb{R}^d} \left\{ F^\star(\theta) + \frac{\mu}{2} \|\theta\|_2^2 \right\},\ \text{ where } F^\star(\theta) := \mathbb{E}_{f \sim \mathcal{D}_f} [f(\theta)]. \tag{11}$$

Here, $\mu \in \mathbb{R}_{\geq 0}$ controls the amount of regularization, and is used to introduce some amount of strong convexity in the problem. Following Section 2.2, the distribution $\mathcal{D}_f$ over sampled functions is directly dependent on a dataset distribution, $\mathcal{D}_{Xy}$, through the relationship in Model 2. Concretely, we make the following regularity assumptions about the distribution $\mathcal{D}_{Xy}$ and its induced $\mathcal{D}_f$.

**Model 5** (distributional regularity for smooth GLMs). $\mathcal{D}_{Xy}$, supported on $\mathbb{R}^d \times \mathbb{R}$, its marginals $\mathcal{D}_X$, $\mathcal{D}_y$, and its induced function distribution $\mathcal{D}_f$, have the following properties.

1. Letting the second moment matrix of $\mathcal{D}_X$ be $\mathbf{\Sigma}^\star$ and $0 < L$, we have $\mathbf{\Sigma}^\star \preceq L\mathbf{I}$.
2. There is a link function $\gamma : \mathbb{R}^2 \to \mathbb{R}$, such that for all $y$ in the support of $\mathcal{D}_y$, $\gamma_y(v) := \gamma(v, y)$ satisfies $\left|\gamma'_y(v)\right| \leq 1$ and $0 \leq \gamma''_y(v) \leq 1$ for all $v \in \mathbb{R}$.
3. The distribution of $f \sim \mathcal{D}_f$ is generated as follows: for $(X, y) \sim \mathcal{D}_{Xy}$, $f(\theta) = \gamma(\langle X, \theta \rangle, y)$.

In Section 5, we further develop algorithms for the special case of Model 2 when all $\gamma_{y_i}(v) := \gamma(v, y_i)$ satisfy only a first-derivative bound,

$$\left|\gamma'_{y_i}(v)\right| \leq 1 \text{ for all } v \in \mathbb{R}, i \in [n].$$

In other words, all $\gamma_{y_i}$ are 1-Lipschitz (but possibly non-smooth). A canonical example is when all $y_i = \pm 1$ are positive or negative labels, and $\gamma$ is the support vector machine loss function (hinge loss), $\gamma(v, y) = \max(0, 1 - vy)$. In this setting, we will focus on approximating the (population) regularized optimizer of the *Moreau envelope*,

$$\theta^\star_{\text{env}} := \operatorname{argmin}_{\theta \in \mathbb{R}^d} \left\{ F^\star_\gamma(\theta) + \frac{\mu}{2} \|\theta\|_2^2 \right\},\ \text{ where } F^\star(\theta) = \mathbb{E}_{f \sim \mathcal{D}_f} [f(\theta)],$$

$$\text{and } F^\star_\lambda(\theta) := \inf_{\theta'} \left\{ F^\star(\theta') + \frac{1}{2\lambda} \|\theta - \theta'\|_2^2 \right\}.$$

The Moreau envelope is extremely well-studied [PB14], and can be viewed as a smooth approximation to a non-smooth function. We choose to focus on optimizing the Moreau envelope because it is amenable to acceleration techniques, trading off approximation for smoothness.

**Model 6** (distributional regularity for Lipschitz GLMs). $\mathcal{D}_{Xy}$, supported on $\mathbb{R}^d \times \mathbb{R}$, its marginals $\mathcal{D}_X$, $\mathcal{D}_y$, and its induced function distribution $\mathcal{D}_f$, have the following properties.

1. Letting the second moment matrix of $\mathcal{D}_X$ be $\mathbf{\Sigma}^\star$ and $0 < L$, we have $\mathbf{\Sigma}^\star \preceq L\mathbf{I}$.

2. There is a link function $\gamma : \mathbb{R}^2 \to \mathbb{R}$, such that for all $y$ in the support of $\mathcal{D}_y$, $\gamma_y(v) := \gamma(v, y)$ satisfies $\left| \gamma'_y(v) \right| \leq 1$ for all $v \in \mathbb{R}$.

3. The distribution of $f \sim \mathcal{D}_f$ is generated as follows: for $(X, y) \sim \mathcal{D}_{Xy}$, $f(\theta) = \gamma(\langle X, \theta \rangle, y)$.

In other words, Model 6 is Model 5 without the smoothness assumption. Under the weaker Model 6, [DKK$^+$19] showed that we can make the following simplifying deterministic assumptions about our observed dataset (which also extends to Model 5, as it a superset of conditions).

**Assumption 3** (deterministic regularity for Lipschitz regression)**.** The set $\{(X_i, y_i)\}_{i \in [n]} \subset \mathbb{R}^d \times \mathbb{R}$, and the link function $\gamma : \mathbb{R}^2 \to \mathbb{R}$, have the following properties, for $[n] = G \cup B$.

1. Letting $w_G^\star := \frac{1}{|G|} \mathbb{1}_G$, $\mathrm{Cov}_{w_G^\star}(\mathbf{X}) \preceq \frac{3}{2} L \mathbf{I}$.

2. There is a constant $C_{\mathrm{est}}$ such that for all $\theta \in \mathbb{R}^d$ and all saturated $w \in \Delta^n$, letting $\tilde{w} := \frac{w_G}{\|w_G\|_1}$,

$$\|\nabla F_{\tilde{w}}(\theta) - \nabla F^\star(\theta)\|_2 \leq C_{\mathrm{est}} \sqrt{L} \epsilon,$$
$$\left\| \mathrm{Cov}_{\tilde{w}} \left( \{g_i(\theta)\}_{i \in G} \right) \right\|_{\mathrm{op}} \leq C_{\mathrm{est}} L.$$

**Proposition 4** ([DKK$^+$19], Proposition C.3)**.** Let $\epsilon > 0$ be sufficiently small. Let $\{(X_i, y_i)\}_{i \in [n]} \subset \mathbb{R}^d \times \mathbb{R}$ be an $\epsilon$-corrupted set of samples from a distribution $\mathcal{D}_{Xy}$ as in Model 6. Then, if

$$n = O\left( \frac{d \log(d/\epsilon)}{\epsilon} \right),$$

Assumption 3 holds with probability at least $\frac{9}{10}$.

Finally, we make the useful observation that $F^\star$ under Model 6 is also Lipschitz.

**Lemma 4.** *Under Model 6, $F^\star = \mathbb{E}_{f \sim \mathcal{D}_f}[f]$ is $\sqrt{L}$-Lipschitz.*

*Proof.* We wish to prove $\|\nabla F^\star(\theta)\|_2 \leq \sqrt{L}$ for all $\theta \in \mathbb{R}^d$. By nonnegativity of covariance,

$$\mathbb{E}_{f \sim \mathcal{D}_f} \left[ (\nabla f(\theta)) (\nabla f(\theta))^\top \right] \succeq \left( \mathbb{E}_{f \sim \mathcal{D}_f} [\nabla f(\theta)] \right) \left( \mathbb{E}_{f \sim \mathcal{D}_f} [\nabla f(\theta)] \right)^\top = (\nabla F^\star(\theta)) (\nabla F^\star(\theta))^\top.$$

Since the right-hand side of the above display is rank-one, $\|\nabla F^\star(\theta)\|_2^2$ is at most the largest eigenvalue of the gradient second moment matrix, so it suffices to show the left-hand side is $\preceq L\mathbf{I}$: assuming $f \sim \mathcal{D}_f$ is associated with $(X, y) \sim \mathcal{D}_{Xy}$,

$$\mathbb{E}_{f \sim \mathcal{D}_f} \left[ (\nabla f(\theta)) (\nabla f(\theta))^\top \right] = \mathbb{E}_{X, y \sim \mathcal{D}_{Xy}} \left[ X \left( \gamma'(\langle X, \theta \rangle, y) \right)^2 X^\top \right] \preceq \mathbf{\Sigma}^\star \preceq L\mathbf{I}.$$

$\square$

## 2.5 Robustly decreasing the covariance operator norm

In this section, we describe a procedure, FastCovFilter, which takes as input a set of vectors $\mathbf{V} := \{v_i\}_{i \in [n]} \in \mathbb{R}^{n \times d}$ such that for an (unknown) bipartition $[n] = G \cup B$, it is promised that $\mathbb{E}_{i \sim \mathrm{unif} G} \left[ v_i v_i^\top \right]$ is bounded in operator norm by $R$. Given this promise, FastCovFilter takes as input a set of saturated weights $w \in \Delta^n$ and performs a sequence of safe weight removals to obtain a

new saturated $w' \in \Delta^n$, such that $\sum_{i \in [n]} w'_i v_i v_i^\top$ is bounded in operator norm by $O(R)$. Moreover, the procedure runs in nearly-linear time in the description size of $\mathbf{V}$. We formally describe the guarantees of FastCovFilter here as Proposition 5, and defer the proof to Appendix A.

**Proposition 5.** There is an algorithm, FastCovFilter (Algorithm 7), taking inputs $\mathbf{V} := \{v_i\}_{i \in [n]} \in \mathbb{R}^{n \times d}$, saturated weights $w \in \Delta^n$ with respect to bipartition $[n] = G \cup B$ with $|B| = \epsilon n$, $\delta \in (0, 1]$, and $R \geq 0$ with the promise that

$$\left\| \sum_{i \in G} \frac{1}{|G|} v_i v_i^\top \right\|_{\mathrm{op}} \leq R.$$

Then, with probability at least $1 - \delta$, FastCovFilter returns saturated $w' \in \Delta^n$ such that

$$\left\| \sum_{i \in [n]} w'_i v_i v_i^\top \right\|_{\mathrm{op}} \leq 5R.$$

The runtime of FastCovFilter is

$$O\left( nd \log^3(n) \log\left( \frac{n}{\delta} \right) \right).$$

An algorithm with similar guarantees to FastCovFilter is implicit in the work [DHL19], but we give a self-contained exposition in this work for completeness. Our approach in designing FastCovFilter is to use a matrix multiplicative weights-based potential function, along with the filtering approach suggested by Lemma 1, to rapidly decrease the quadratic form of the empirical second moment matrix in a number of carefully-chosen directions, and argue this quickly decreases the potential. We remark that this potential-based approach was also used in the recent work [DKK+21].

## 3 Linear regression

Throughout this section, we operate under Models 1 and 3, and Assumption 1. Namely, there is a dataset $\{X_i, y_i\}_{i \in [n]}$ and an unknown bipartition $[n] = G \cup B$, such that $\{X_i, y_i\}_{i \in G}$ were draws from a distribution $\mathcal{D}_{Xy}$, and we wish to estimate

$$\theta^\star := \operatorname{argmin}_{\theta \in \mathbb{R}^d} F^\star(\theta), \text{ where } F^\star(\theta) := \mathbb{E}_{X, y \sim \mathcal{D}_{Xy}} \left[ \frac{1}{2} (\langle X, \theta \rangle - y)^2 \right].$$

We follow notation (4), (6) in this section, and define $G^\star \subseteq G$ as the subset given by Assumption 1.2 for the true minimizer $\theta^\star$. We also define $B^\star := [n] \setminus G^\star$, so $B^\star \supseteq B$.

We begin in Section 3.1 with a preliminary on filtering under a weaker assumption than the safety condition in Definition 2; in particular, Assumption 1 is not quite compatible with Definition 2 because $G'$ can change based on $\theta$, but will not affect saturation by more than constants. In Section 3.2, we then state a general "identifiability proof" showing that controlling certain quantities such as the operator norm of the gradient covariances and near-optimality of a current estimate $\theta$, with respect to some weights $w \in \Delta^n$, suffices to bound closeness of $\theta$ and $\theta^\star$. This identifiability proof is motivated by an analogous argument in [BP21], and will guide our algorithm design. Next, we give a self-contained oracle which rapidly halves the distance to $\theta^\star$ outside of a sufficiently large radius in Section 3.3, and analyze the final phase (once this radius is reached) in Section 3.4. We put the pieces together to give our main result and full algorithm in Section 3.5.

## 3.1 Filtering under $(\epsilon, \frac{\epsilon^2}{\alpha})$-goodness

Throughout this section, we will globally fix a value (for a sufficiently large constant)

$$\alpha = O\left(\max\left(1, \epsilon\kappa \log \frac{R_0}{\sigma}\right)\right). \tag{12}$$

In this definition, $R_0$ is an initial distance bound on $\|\theta_0 - \theta^\star\|_{\boldsymbol{\Sigma}^\star}$ we will provide to our final algorithm (see the statement of Theorem 5). We operate under Assumption 1 such that our dataset is $(\epsilon, \frac{\epsilon^2}{\alpha})$-good, which inflates the sample complexity of Proposition 2 by a factor depending on $\alpha$.

At a high level, this technical complication is to ensure that throughout the algorithm we never remove more than $3\epsilon$ mass from $G$, and that our weights are always $\epsilon$-saturated with respect to $G \cup B$, allowing for inductive application of Assumption 1. Formally, we demonstrate the following (simple) generalization of Lemma 1, using safety definitions on different sets.

**Lemma 5.** *Suppose Assumption 1 holds with $r = \frac{\epsilon^2}{\alpha}$. Consider any algorithm which performs the following weight removal.*

1. *$w_0 \leftarrow \frac{1}{n}\mathbb{1}$*
2. *For $0 \le t < T$:*
   (a) *$[w_{t+1}]_i \leftarrow \left(1 - \frac{\tau_i}{\tau_{\max}}\right)[w_t]_i$ for all $i \in [n]$ and $\tau_{\max} := \max_{i \in [n]|w_i \neq 0} \tau_i$, for $\{\tau_i\}_{i \in [n]}$ safe with respect to $w_t$ and a bipartition $G'_t$, $[n] \setminus G'_t$ where $G'_t \subseteq G$, $|G'_t| \ge (1 - \frac{\epsilon^2}{\alpha})|G|$.*

*Suppose the number of distinct sets $G'_t$ throughout the algorithm is bounded by $\frac{\alpha}{2\epsilon}$. Then throughout the algorithm, $w_t$ is $\epsilon$-saturated with respect to the bipartition $G \cup B$.*

*Proof.* Consider some distinct set $G'$. The proof of Lemma 1 demonstrates that under these assumptions, in every iteration $t$ using $G'$ for weight removal, the amount of mass removed from $G'$ is less than the amount removed from $[n] \setminus G'$. Moreover, the amount of mass removed from $G$ in these iterations can be at most the amount removed from $G'$, plus the weight assigned to the *entire difference* $G \setminus G'$, which is at most a $\frac{\epsilon^2}{\alpha}$ fraction. Similarly, the amount of mass removed from $B$ is at least the amount of mass removed from $[n] \setminus G'$, minus their set difference, which is again at most $\frac{\epsilon^2}{\alpha}$. Combining over all distinct sets, this deviation is at most $\epsilon$. $\qquad\square$

It will be straightforward to verify that throughout this section, all weight removals we perform will be of the form in Lemma 5, and that we never perform weight removals with respect to more than $\frac{\alpha}{2\epsilon}$ distinct sets. Hence, we will always assume that any weights $w$ we discuss are $\epsilon$-saturated with respect to the bipartition $G \cup B$, and thus has $\|w_G\|_1 \ge 1 - 3\epsilon$, allowing for application of Assumption 1. Finally, we remark we sometimes will apply Assumption 1.1 with weight vectors $w_{G^\star}$ instead of $w_G$; since their difference is $O(\epsilon^2) < \epsilon$, this is a correct application for any $\|w_G\|_1 \ge 1 - 3\epsilon$.

## 3.2 Identifiability proof for linear regression

In this section, we give an identifiability proof similar to that appearing in [BP21] which demonstrates, for a given weight-parameter estimate pair $(w, \theta) \in \Delta^n \times \mathbb{R}^d$, verifiable technical conditions on this pair which certify a bound on $\|\theta - \theta^\star\|_{\boldsymbol{\Sigma}^\star}$. In short, Proposition 6 will show that if both of the quantities

$$\|\nabla F_w(\theta)\|_{(\boldsymbol{\Sigma}^\star)^{-1}}, \ \left\|\mathrm{Cov}_w\left(\{g_i(\theta)\}_{i \in [n]}\right)\right\|_{\mathrm{op}} \tag{13}$$

are simultaneously controlled, then we obtain a distance bound to $\theta_{G^\star}$.

**Proposition 6.** Let $w \in \Delta^n$ be $\epsilon$-saturated with respect to the bipartition $[n] = G \cup B$, and let $\theta \in \mathbb{R}^d$. Assuming $\epsilon\kappa$ is sufficiently small, there is a universal constant $C_{\mathrm{id}}$ such that

$$\|\theta - \theta^\star\|_{\mathbf{\Sigma}^\star} \leq C_{\mathrm{id}} \left( \sqrt{\epsilon} \left( \sigma\sqrt{\kappa} + \sqrt{\frac{\left\| \mathrm{Cov}_w \left( \{g_i(\theta)\}_{i \in [n]} \right) \right\|_{\mathrm{op}}}{\mu}} \right) + \|\nabla F_w(\theta)\|_{(\mathbf{\Sigma}^\star)^{-1}} \right).$$

*Proof.* Let $G'$ be the set promised by Assumption 1.2 for the point $\theta$. Throughout this proof, we define $G_\theta := G' \cap G^\star$, and we let $w_\theta^\star := \frac{1}{|G_\theta|} \mathbb{1}_{G_\theta}$ be the uniform weights on $G_\theta$. Finally, let $\hat{\theta}$ minimize $F_{w_\theta^\star}$. By applying Lemma 3 on the weights $w_\theta^\star$, we have that $\|\hat{\theta} - \theta^\star\|_{\mathbf{\Sigma}^\star} = O(\sigma\sqrt{\kappa\epsilon})$. Thus, in the remainder of the proof we focus on bounding $\|\theta - \hat{\theta}\|_{\mathbf{\Sigma}^\star}$ by the required quantity.

Let $\mathcal{C}(w_\theta^\star, \tilde{w})$ supported on $[n] \times [n]$ be an optimal *coupling* between $i \sim w_\theta^\star$ and $j \sim \tilde{w}$, where $\tilde{w} := \frac{w}{\|w\|_1}$ is the distribution proportional to $w$; we denote the coupling as $\mathcal{C}$ for short. For a pair $(i, j) \in [n] \times [n]$ sampled from $\mathcal{C}$, we let $\mathbb{1}_{i=j}$ be the indicator of the event $i = j$, and similarly define $\mathbb{1}_{i \neq j}$. From the total variation characterization of coupling, we have by Lemma 2 that

$$\mathop{\mathbb{E}}_{i,j \sim \mathcal{C}} [\mathbb{1}_{i \neq j}] = \|\tilde{w} - w_\theta^\star\|_1 \leq 9\epsilon.$$

Here we used that $\|w_\theta^\star - w_G^\star\|_1 = O(\epsilon^2)$, where $w_G^\star$ is the uniform distribution on $G$, as in Lemma 2. Now, let $v = \theta - \hat{\theta}$. We have by Assumption 1.1 that

$$\mathop{\mathbb{E}}_{i \sim w_\theta^\star} \left[ \langle v, X_i \rangle \left\langle X_i, \theta - \hat{\theta} \right\rangle \right] = \left\langle \theta - \hat{\theta}, \mathop{\mathbb{E}}_{i \sim w_\theta^\star} \left[ X_i X_i^\top \right] (\theta - \hat{\theta}) \right\rangle \geq \frac{1}{2} \left\| \theta - \hat{\theta} \right\|_{\mathbf{\Sigma}^\star}^2. \tag{14}$$

On the other hand,

$$\mathop{\mathbb{E}}_{i \sim w_\theta^\star} \left[ \langle v, X_i \rangle \left\langle X_i, \theta - \hat{\theta} \right\rangle \right] = \mathop{\mathbb{E}}_{i \sim w_\theta^\star} [\langle v, X_i \rangle (\langle X_i, \theta \rangle - y_i)] + \mathop{\mathbb{E}}_{i \sim w_\theta^\star} \left[ \langle v, X_i \rangle \left( y_i - \left\langle X_i, \hat{\theta} \right\rangle \right) \right]$$

$$= \mathop{\mathbb{E}}_{i \sim w_\theta^\star} [\langle v, g_i(\theta) \rangle] - \mathop{\mathbb{E}}_{i \sim w_\theta^\star} \left[ \left\langle v, g_i(\hat{\theta}) \right\rangle \right] = \mathop{\mathbb{E}}_{i \sim w_\theta^\star} [\langle v, g_i(\theta) \rangle].$$

Here, we used that $\mathbb{E}_{i \sim w_\theta^\star} \left[ g_i(\hat{\theta}) \right] = \nabla F_{w_\theta^\star}(\hat{\theta}) = 0$ by definition of $\hat{\theta}$. Continuing,

$$\mathop{\mathbb{E}}_{i \sim w_\theta^\star} \left[ \langle v, X_i \rangle \left\langle X_i, \theta - \hat{\theta} \right\rangle \right] = \mathop{\mathbb{E}}_{i,j \sim \mathcal{C}} [\langle v, g_i(\theta) \rangle \mathbb{1}_{i=j}] + \mathop{\mathbb{E}}_{i,j \sim \mathcal{C}} [\langle v, g_i(\theta) \rangle \mathbb{1}_{i \neq j}]$$

$$= \mathop{\mathbb{E}}_{j \sim \tilde{w}} [\langle v, g_j(\theta) \rangle] - \mathop{\mathbb{E}}_{i,j \sim \mathcal{C}} [\langle v, g_j(\theta) \rangle \mathbb{1}_{i \neq j}] + \mathop{\mathbb{E}}_{i,j \sim \mathcal{C}} [\langle v, g_i(\theta) \rangle \mathbb{1}_{i \neq j}] \tag{15}$$

$$\leq \langle v, \nabla F_{\tilde{w}}(\theta) \rangle + 3\sqrt{\epsilon} \left( \mathop{\mathbb{E}}_{j \sim \tilde{w}} \left[ \langle v, g_j(\theta) \rangle^2 \right]^{\frac{1}{2}} + \mathop{\mathbb{E}}_{i \sim w_\theta^\star} \left[ \langle v, g_i(\theta) \rangle^2 \right]^{\frac{1}{2}} \right).$$

In the last line, we used Cauchy-Schwarz and $\mathbb{E}_{i,j \sim \mathcal{C}}[\mathbb{1}_{i \neq j}^2] \leq 9\epsilon$ to deal with the second and third terms. To bound the term corresponding to $i \sim w_\theta^\star$,

$$\mathop{\mathbb{E}}_{i \sim w_\theta^\star} \left[ \langle v, g_i(\theta) \rangle^2 \right] \leq \|v\|_2^2 \left\| \mathrm{Cov}_{w_\theta^\star} \left( \{g_i(\theta)\}_{i \in [n]} \right) \right\|_{\mathrm{op}} \leq 1.1 C_{\mathrm{est}} \kappa \|v\|_{\mathbf{\Sigma}^\star}^2 \left( \|\theta - \theta^\star\|_{\mathbf{\Sigma}^\star}^2 + \sigma^2 \right).$$

The first inequality is by definition of $\|\cdot\|_{\mathrm{op}}$, and the second used $\|v\|_2^2 \geq \frac{1}{\mu} \|v\|_{\mathbf{\Sigma}^\star}^2$ and Assump-

tion 1.2, since $G_\theta$ is a subset of the set $G'$ promised by Assumption 1.2 (where we adjusted by a constant for normalization). We similarly arrive at the bound

$$\mathbb{E}_{j\sim\tilde{w}}\left[\langle v, g_j(\theta)\rangle^2\right] \le \frac{\|v\|_{\boldsymbol{\Sigma}^\star}^2}{\mu}\left\|\mathrm{Cov}_w\left(\{g_i(\theta)\}_{i\in[n]}\right)\right\|_{\mathrm{op}}.$$

Now plugging the above displays into (15) and combining with (14), as well as using $\|\theta - \theta^\star\|_{\boldsymbol{\Sigma}^\star}^2 + \sigma^2 = O(\|\theta - \hat{\theta}\|_{\boldsymbol{\Sigma}^\star}^2 + \sigma^2)$ by the triangle inequality and our earlier bound on $\|\theta^\star - \hat{\theta}\|_{\boldsymbol{\Sigma}^\star}$,

$$\left\|\theta - \hat{\theta}\right\|_{\boldsymbol{\Sigma}^\star}^2 \le \left\|\theta - \hat{\theta}\right\|_{\boldsymbol{\Sigma}^\star}\left\|\nabla F_{\tilde{w}}(\theta)\right\|_{(\boldsymbol{\Sigma}^\star)^{-1}} + O\left(\sqrt{\epsilon\kappa}\right)\left(\sigma\left\|\theta - \hat{\theta}\right\|_{\boldsymbol{\Sigma}^\star} + \left\|\theta - \hat{\theta}\right\|_{\boldsymbol{\Sigma}^\star}^2\right)$$

$$+ O\left(\sqrt{\epsilon}\right)\left(\left\|\theta - \hat{\theta}\right\|_{\boldsymbol{\Sigma}^\star}\sqrt{\frac{\left\|\mathrm{Cov}_w\left(\{g_i(\theta)\}_{i\in[n]}\right)\right\|_{\mathrm{op}}}{\mu}}\right).$$

In the first line, we used Cauchy-Schwarz to bound the term $\langle v, \nabla F_{\tilde{w}}(\theta)\rangle$. Dividing through by $\|\theta - \hat{\theta}\|_{\boldsymbol{\Sigma}^\star}$, and using that $\epsilon\kappa$ is sufficiently small, yields the conclusion. $\qquad\square$

Proposition 6 suggests a natural approach. On the one hand, if $\theta$ is an (approximate) minimizer to $F_w$, the first quantity in (13) will be small. On the other hand, for a fixed $\theta \in \mathbb{R}^d$, we can filter on $w$ using our subroutine FastCovFilter until the second quantity in (13) is small. The main challenge is accomplishing both bounds simultaneously. To this end, we show that the number of times we have to repeat this process of filtering and then computing an approximate minimizer is bounded, using $F_w$ as a potential function; by preprocessing $F_w$ so that is smooth at all points, if $\theta$ is an approximate minimizer for the $F_w$ attained *after* filtering on gradients at $\theta$, then we can exit the subroutine. Otherwise, we argue we make substantial function progress by calling an empirical risk minimizer to terminate quickly. We make this strategy formal in the following sections.

### 3.3 Halving the distance to $\theta^\star$

In this section, we design a procedure, HalfRadiusLinReg, with the following guarantee. Suppose that we have $\epsilon$-saturated $\bar{w} \in \Delta^n$ and a parameter $\bar{\theta} \in \mathbb{R}^d$, as well as scalar $R$ with the promise

$$\left\|\bar{\theta} - \theta^\star\right\|_{\boldsymbol{\Sigma}^\star} \le R.$$

The goal of HalfRadiusLinReg is to return a new $\theta$ with $\|\theta - \theta^\star\|_{\boldsymbol{\Sigma}^\star} \le \frac{1}{2}R$; we require that $R = \Omega(\sigma)$ for a sufficiently large constant in this section. We do so by using saturated weights to guide a potential analysis, crucially using Proposition 6. Before stating HalfRadiusLinReg, we require two additional helper tools. The first is an approximate optimization procedure.

**Definition 3.** We call $\mathcal{O}_{\mathrm{ERM}}$ a $\gamma$-approximate ERM oracle if on input $F : \mathbb{R}^d \to \mathbb{R}$ it returns a point $\hat{\theta}$ such that $F(\hat{\theta}) - F(\theta_F^\star) \le \gamma$, for $\theta_F^\star := \mathrm{argmin}_{\theta\in\mathbb{R}^d}F(\theta)$.

The second controls the initial error, which we use to yield distance bounds via strong convexity.

**Lemma 6.** *There is an algorithm,* FunctionFilter, *which takes as input $\epsilon$-saturated $w \in \Delta^n$, $\theta \in \mathbb{R}^d$, and $R \ge \|\theta - \theta^\star\|_{\boldsymbol{\Sigma}^\star}$, and produces $\epsilon$-saturated $w' \in \Delta^n$ such that $F_{w'}(\theta) \le 2C_{\mathrm{ub}}(\sigma^2 + R^2)$, in time $O(nd + n\log\frac{D}{R^2})$, where $D$ is a bound on the largest $\frac{1}{2}(\langle X_i, \theta\rangle - y_i)^2$ for any nonzero $w_i$.*

*Proof.* Define $\tau_i := f_i(\theta) = \frac{1}{2}(\langle X_i, \theta \rangle - y_i)^2$. Assumption 1.1 shows that $F_{w_{G^\star}}$ is $\frac{3}{2}$-smooth in the $\boldsymbol{\Sigma}^\star$ norm, since its Hessian is exactly $\|w_{G^\star}\|_1 \operatorname{Cov}_{w_{G^\star}}(\mathbf{X}) \preceq \frac{3}{2}\boldsymbol{\Sigma}^\star$. Moreover, letting $\hat{\theta}$ minimize $F_{w_{G^\star}}$, by Lemma 3, the triangle inequality and the assumed bound $R$,

$$\left\| \theta - \hat{\theta} \right\|_{\boldsymbol{\Sigma}^\star} \le R + \left\| \theta^\star - \hat{\theta} \right\|_{\boldsymbol{\Sigma}^\star} \le R + \sigma.$$

Hence, assuming $C_{\mathrm{ub}}$ is large enough and $R \ge \sigma$, smoothness demonstrates

$$\sum_{i \in G^\star} w_i \tau_i = F_{w_{G^\star}}(\theta) \le F_{w_{G^\star}}(\hat{\theta}) + \frac{3}{4}\left\| \theta - \hat{\theta} \right\|_{\boldsymbol{\Sigma}^\star}^2 \le C_{\mathrm{ub}}(\sigma^2 + R^2),$$

Next, letting $\tau_{\max} := \max_{i \in [n] | w_i \ne 0} \tau_i$ and $K \in \mathbb{N} \cup \{0\}$ be smallest such that

$$\sum_{i \in [n]} \left( 1 - \frac{\tau_i}{\tau_{\max}} \right)^K w_i \tau_i \le 2C_{\mathrm{ub}}(\sigma^2 + R^2),$$

each of the first $K$ weight removals according to the scores $\tau$ are safe with respect to $G^\star \cup B^\star$, and Lemma 5 implies we can output $\epsilon$-saturated $w_i' = \left( 1 - \frac{\tau_i}{\tau_{\max}} \right)^K w_i$ entrywise. It remains to binary search for $K$; given access to the scores $\tau$, checking if a given $K$ passes the above display takes $O(n)$ time. We can upper bound $K$ by the following inequality:

$$\sum_{i \in [n]} \left( 1 - \frac{\tau_i}{\tau_{\max}} \right)^K w_i \tau_i \le \sum_{i \in [n]} \exp\left( -\frac{K \tau_i}{\tau_{\max}} \right) w_i \tau_i \le \frac{1}{eK} \sum_{i \in [n]} w_i \tau_{\max} \le \frac{\tau_{\max}}{K}.$$

Here the second inequality used $x \exp(-Cx) \le \frac{1}{eC}$ for all nonnegative $x, C$, where we chose $C = \frac{K}{\tau_{\max}}$ and $x = \tau_i$. Hence, $K = O(\frac{D}{R^2})$. The runtime follows as computing scores takes time $O(nd)$. $\square$

We remark that every time we use FunctionFilter is a weight removal of the form in Lemma 5, which is safe with respect to the bipartition $G^\star \cup B^\star$. This accounts for one distinct set throughout.

**Lemma 7.** HalfRadiusLinReg *is correct, i.e. if its preconditions are met, it successfully returns* $(w, \theta)$ *such that* $w \in \Delta^n$ *is $\epsilon$-saturated and* $\|\theta - \theta^\star\|_{\boldsymbol{\Sigma}^\star} \le \frac{1}{2}R$. *It runs in* $O(\kappa)$ *calls to* $\mathcal{O}_{\mathrm{ERM}}$, *plus*

$$O\left( n \log\left( \frac{D}{R^2} \right) + nd\kappa \log^3(n) \log\left( \frac{n\kappa}{\delta} \right) \right) \quad \text{additional time.}$$

*Proof.* We discuss correctness and runtime separately.

*Correctness.* The first step of our correctness proof is to show that throughout the algorithm,

$$\left\| \theta^{(t)} - \theta^\star \right\|_{\boldsymbol{\Sigma}^\star} \le 6\sqrt{C_{\mathrm{ub}}}R. \tag{17}$$

To see this, consider iteration $t$ and suppose $w^{(t)}$ is $\epsilon$-saturated. Let $G' \subseteq G$ be the set promised by Assumption 1.2 for the pair $(w^{(t)}, \theta^{(t)})$, and let $G_t = G' \cap G^\star$. At the beginning of the algorithm we applied FunctionFilter (see Lemma 6), and the minimum value of $F_w$ is monotone nonincreasing as $w$ is decreasing, and $\mathcal{O}_{\mathrm{ERM}}$ only decreases function value (else Line 4 would fail), so since $R \ge \sigma$,

$$F_{w_{G_t}^{(t)}}\left( \theta^{(t)} \right) \le F_{w^{(t)}}\left( \theta^{(t)} \right) \le 4C_{\mathrm{ub}}R^2.$$

---

**Algorithm 1** HalfRadiusLinReg($\mathbf{X}, y, \bar{w}, \bar{\theta}, R, D, \mathcal{O}_{\mathrm{ERM}}, \delta$)

---

1: **Input:** Dataset $\mathbf{X} = \{X_i\}_{i \in [n]} \in \mathbb{R}^{n \times d}$, $y = \{y_i\}_{i \in [n]} \in \mathbb{R}^n$, satisfying Assumption 1, $\epsilon$-saturated $\bar{w} \in \Delta^n$ with respect to bipartition $[n] = G \cup B$ such that $\mathbf{X}^\top \mathbf{diag}(\bar{w}) \mathbf{X} \preceq 8L\mathbf{I}$, $\bar{\theta} \in \mathbb{R}^d$ with $\left\| \bar{\theta} - \theta^\star \right\|_{\mathbf{\Sigma}^\star} \leq R$, $\delta \in (0, 1)$, $O(\frac{\sigma^2}{\kappa})$-approximate ERM oracle $\mathcal{O}_{\mathrm{ERM}}$,

$$D \geq \max_{i \in [n]} \frac{1}{2} (\langle X_i, \bar{\theta} \rangle - y_i)^2. \tag{16}$$

2: **Output:** With probability $\geq 1 - \delta$, saturated $w$ with respect to $G \cup B$, and $\theta$ with

$$\|\theta - \theta^\star\|_{\mathbf{\Sigma}^\star} \leq \frac{1}{2} R.$$

3: $t \leftarrow 0$, $w^{(0)} \leftarrow \mathsf{FunctionFilter}(w, \bar{\theta}, R, D)$, $\Delta_0 \leftarrow \infty$
4: **while** $\Delta_t > \frac{R^2}{512 \kappa C_{\mathrm{id}}^2}$ **do**
5: $\quad \theta^{(t)} \leftarrow \mathcal{O}_{\mathrm{ERM}}(F_{w^{(t)}})$
6: $\quad w^{(t+1)} \leftarrow \mathsf{FastCovFilter}\left( \{g_i(\theta^{(t)})\}_{i \in [n]}, w^{(t)}, \frac{\delta}{O(\kappa)}, 40 C_{\mathrm{est}} C_{\mathrm{ub}} L R^2 \right)$
7: $\quad \Delta_{t+1} \leftarrow F_{w^{(t+1)}}(\theta^{(t)}) - F_{w^{(t+1)}}(\theta^{(t+1)})$
8: $\quad t \leftarrow t + 1$
9: **end while**
10: **return** $\left( w^{(t)}, \theta^{(t-1)} \right)$

---

On the other hand, $F_{w_{G_t}^{(t)}}$ is $\frac{1}{3}$-strongly convex in the $\mathbf{\Sigma}^\star$ norm by Assumption 1.1 (adjusting for the normalization factor), and hence letting $\theta_t^\star$ be the minimizer of $F_{w_{G_t}^{(t)}}$, we have

$$4 C_{\mathrm{ub}} R^2 \geq F_{w_{G_t}^{(t)}}\left( \theta^{(t)} \right) \geq F_{w_{G_t}^{(t)}}\left( \theta^{(t)} \right) - F_{w_{G_t}^{(t)}}(\theta_t^\star) \geq \frac{1}{6} \left\| \theta^{(t)} - \theta_t^\star \right\|_{\mathbf{\Sigma}^\star}^2 \implies \left\| \theta^{(t)} - \theta_t^\star \right\|_{\mathbf{\Sigma}^\star} \leq 5 \sqrt{C_{\mathrm{ub}}} R.$$

Finally, by using Lemma 3 on $\theta_t^\star$, we have $\|\theta_t^\star - \theta^\star\|_{\mathbf{\Sigma}^\star} \leq 4 C_{\mathrm{est}} \sigma \sqrt{\kappa \epsilon} \leq R$. From this and the above display, the triangle inequality yields (17). Now, (17) with Assumption 1.2 shows that for all $t$,

$$\left\| \mathrm{Cov}_{\frac{1}{|G_t|} \mathbb{1}_{G_t}} \left( \{g_i(\theta^{(t)})\}_{i \in G_t} \right) \right\|_{\mathrm{op}} \leq 40 C_{\mathrm{est}} C_{\mathrm{ub}} L R^2.$$

We argue later in this proof that there are at most $O(\kappa)$ loops throughout the algorithm. This shows all calls to $\mathsf{FastCovFilter}$ are safe with respect to the bipartition $G_t, [n] \setminus G_t$ (adjusting the definition of $\epsilon$ by a constant in Proposition 5), and thus taking a union bound the algorithm succeeds with probability at least $1 - \delta$. Condition on this for the remainder of the proof.

The success of all calls to $\mathsf{FastCovFilter}$ implies that in every iteration $t$ (following Proposition 5),

$$\left\| \mathrm{Cov}_{w^{(t+1)}} \left( \{g_i(\theta^{(t)})\}_{i \in [n]} \right) \right\|_{\mathrm{op}} = O\left( L R^2 \right). \tag{18}$$

Next, in any iteration where $\Delta_{t+1} \le \frac{R^2}{512\kappa C_{\mathrm{id}}^2}$, we claim that

$$\left\| \nabla F_{w^{(t+1)}} \left( \theta^{(t)} \right) \right\|_{(\mathbf{\Sigma}^\star)^{-1}} \le \frac{R}{4C_{\mathrm{id}}}. \tag{19}$$

This is because $F_{w^{(t+1)}}$ is $8\kappa$-smooth in the $\mathbf{\Sigma}^\star$ norm, since $\nabla^2 F_{w^{(t+1)}} = \mathbf{X}^\top \mathbf{diag}\left( w^{(t+1)} \right) \mathbf{X} \preceq 8L\mathbf{I}$ by assumption, and $8\kappa \mathbf{\Sigma}^\star \succeq 8L\mathbf{I}$ by assumption, so by the guarantees of $\mathcal{O}_{\mathrm{ERM}}$,

$$\frac{1}{16\kappa} \left\| \nabla F_{w^{(t+1)}} \left( \theta^{(t)} \right) \right\|_{(\mathbf{\Sigma}^\star)^{-1}}^2 \le F_{w^{(t+1)}} \left( \theta^{(t)} \right) - \min_{\theta \in \mathbb{R}^d} F_{w^{(t+1)}} (\theta)$$
$$\le \Delta_{t+1} + O\left( \frac{\sigma^2}{\kappa} \right) \le \frac{R^2}{256\kappa C_{\mathrm{id}}^2}.$$

Rearranging indeed yields (19). Now by combining (18) and (19) in Proposition 6, we see that if $\Delta_{t+1} \le \frac{R^2}{512\kappa C_{\mathrm{id}}^2}$ in an iteration, we obtain the desired $\left\| \theta^{(t)} - \theta^\star \right\|_{\mathbf{\Sigma}^\star} \le \frac{1}{2}R$.

*Runtime.* We first observe that the loop in Lines 4-9 of Algorithm 1 can only run $O(\kappa)$ times. This is because FunctionFilter decreases the initial function value until it is $O(R^2)$, and every loop decreases the function value by $\Omega(\frac{R^2}{\kappa})$. Hence, the cost of the whole algorithm is one call to FunctionFilter, and $O(\kappa)$ calls to $\mathcal{O}_{\mathrm{ERM}}$, FastCovFilter, and two function value computations, which fit into the allotted runtime budget by Lemma 6 and Proposition 5.

$\square$

Finally, we remark that throughout Algorithm 1, we only filtered weight based on FunctionFilter and FastCovFilter, with respect to at most $O(\kappa)$ distinct sets (in the manner described by Lemma 5): the sets $\{G_t\}$ used in correctness calls to FastCovFilter, and the set $G^\star$ for the one call to FunctionFilter.

### 3.4 Last phase analysis

In this section, we give a slight variant of Algorithm 1 which applies when $R \le C_{\mathrm{lp}}\sigma$ for a universal constant $C_{\mathrm{lp}}$. It will have a somewhat more stringent termination condition, because we require the gradient term in Proposition 6 to be $O(\sigma\sqrt{\epsilon\kappa})$, but otherwise is identical.

**Lemma 8.** LastPhase *is correct, i.e. if its preconditions are met, it successfully returns $(w, \theta)$ such that $w \in \Delta^n$ is $\epsilon$-saturated and $\|\theta - \theta^\star\|_{\mathbf{\Sigma}^\star} = O\left( \sigma\sqrt{\kappa\epsilon} \right)$. It runs in $O(\frac{1}{\epsilon})$ calls to $\mathcal{O}_{\mathrm{ERM}}$, plus*

$$O\left( n \log\left( \frac{D}{R^2} \right) + \frac{nd}{\epsilon} \log^3(n) \log\left( \frac{n}{\delta\epsilon} \right) \right) \text{ additional time.}$$

*Proof.* On the correctness side, the analysis is nearly identical to Lemma 7; the same logic applies to yield an analogous bound to (17), which shows that all iterates are within $O(\sigma)$ from the minimizer, so all calls to FastCovFilter are correct. This implies that (analogous to (18)) the gradient operator norm is always bounded by $O(L\sigma^2)$. Similarly, since the threshold for termination is when the function decrease is $\sigma^2\epsilon$, we have that on the terminating iteration,

$$\frac{1}{16\kappa} \left\| \nabla F_{w^{(t+1)}} \left( \theta^{(t)} \right) \right\|_{(\mathbf{\Sigma}^\star)^{-1}}^2 = O(\sigma^2\epsilon),$$

and combining this with the operator norm bound in Proposition 6 yields the conclusion. On the runtime side, the analysis is the same as Lemma 7, except that there are now $O(\frac{1}{\epsilon})$ iterations. $\square$

---

**Algorithm 2** LastPhase($\mathbf{X}, y, \bar{w}, \bar{\theta}, D, \mathcal{O}_{\mathrm{ERM}}, \delta$)

---

1: **Input:** Dataset $\mathbf{X} = \{X_i\}_{i \in [n]} \in \mathbb{R}^{n \times d}$, $y = \{y_i\}_{i \in [n]} \in \mathbb{R}^n$, satisfying Assumption 1, $\epsilon$-saturated $\bar{w} \in \Delta^n$ with respect to bipartition $[n] = G \cup B$ such that $\mathbf{X}^\top \mathbf{diag}(\bar{w}) \mathbf{X} \preceq 8L\mathbf{I}$, $\bar{\theta} \in \mathbb{R}^d$ with $\left\| \bar{\theta} - \theta^\star \right\|_{\boldsymbol{\Sigma}^\star} \leq C_{\mathrm{lp}} \sigma$, $\delta \in (0, 1)$, $O(\sigma^2 \epsilon)$-approximate ERM oracle $\mathcal{O}_{\mathrm{ERM}}$,

$$D \geq \max_{i \in [n]} \frac{1}{2} (\langle X_i, \bar{\theta} \rangle - y_i)^2. \tag{20}$$

2: **Output:** With probability $\geq 1 - \delta$, saturated $w$ with respect to $G \cup B$, and $\theta$ with

$$\|\theta - \theta^\star\|_{\boldsymbol{\Sigma}^\star} = O\left(\sigma \sqrt{\kappa \epsilon}\right).$$

3:    $t \leftarrow 0$, $w^{(0)} \leftarrow \mathsf{FunctionFilter}(w, \bar{\theta}, C_{\mathrm{lp}} \sigma, D)$, $\Delta_0 \leftarrow \infty$
4: **while** $\Delta_t > \sigma^2 \epsilon$ **do**
5:      $\theta^{(t)} \leftarrow \mathcal{O}_{\mathrm{ERM}}(F_{w^{(t)}})$
6:      $w^{(t+1)} \leftarrow \mathsf{FastCovFilter}\left(\left\{g_i\left(\theta^{(t)}\right)\right\}_{i \in [n]}, w^{(t)}, O(\delta \epsilon), 40 C_{\mathrm{est}} C_{\mathrm{ub}} C_{\mathrm{lp}}^2 L \sigma^2\right)$
7:      $\Delta_{t+1} \leftarrow F_{w^{(t+1)}}\left(\theta^{(t)}\right) - F_{w^{(t+1)}}\left(\theta^{(t+1)}\right)$
8:      $t \leftarrow t + 1$
9: **end while**
10: **return** $\left(w^{(t)}, \theta^{(t-1)}\right)$

---

Again, we remark here that throughout Algorithm 2, we filtered weight with respect to at most $O(\epsilon^{-1})$ distinct sets (in the manner described by Lemma 5).

### 3.5 Full algorithm

Before we give our full algorithm, we require some preliminary pruning procedures on the dataset.

**Lemma 9.** *Under Assumption 1, for all $i \in G$, $\|X_i\|_2 \leq \sqrt{2Ln}$.*

*Proof.* Suppose otherwise for some $i \in G$. Then, since $\mathrm{Cov}_{w_G}(\mathbf{X}) \succeq \frac{1}{|G|} X_i X_i^\top \succeq \frac{1}{n} X_i X_i^\top$, $\mathrm{Cov}_{w_G}(\mathbf{X})$ has an eigenvalue larger than $\frac{3}{2} L$ (certified by $\frac{X_i}{\|X_i\|_2}$), contradicting Assumption 1.1. $\square$

We next give a bound on $D$ required by Algorithms 1 and 2, assuming a bound on $\left\| \bar{\theta} - \theta^\star \right\|_{\boldsymbol{\Sigma}^\star}$.

**Lemma 10.** *Suppose all $\{X_i\}_{i \in [n]}$ satisfy $\|X_i\|_2 \leq \sqrt{2Ln}$, and we have a bound $\left\| \bar{\theta} - \theta^\star \right\|_{\boldsymbol{\Sigma}^\star} \leq R$. Under Assumption 1, it suffices to set $D$ in $\mathsf{HalfRadiusLinReg}$ or $\mathsf{LastPhase}$ to*

$$D = 2n C_{\mathrm{ub}} \sigma^2 + 2\kappa n R^2. \tag{21}$$

*Proof.* First, under Assumption 1 we have that for all $i \in G$,

$$|\langle X_i, \theta^\star \rangle - y_i| \leq \sqrt{2n C_{\mathrm{ub}} \sigma^2}.$$

Applying Cauchy-Schwarz, we have that for all $i \in G$, since $\left\| \bar{\theta} - \theta^\star \right\|_2 \leq \frac{1}{\sqrt{\mu}} \left\| \bar{\theta} - \theta^\star \right\|_{\boldsymbol{\Sigma}^\star} \leq \frac{R}{\sqrt{\mu}}$,

$$\left| \langle X_i, \bar{\theta} \rangle - y_i \right| \leq \sqrt{2n C_{\mathrm{ub}} \sigma^2} + \|X_i\|_2 \left\| \bar{\theta} - \theta^\star \right\|_2 \leq \sqrt{2n C_{\mathrm{ub}} \sigma^2} + \sqrt{2\kappa n} R.$$

$\square$

Finally, we give our full algorithm for regression, FastRegression, below.

---

**Algorithm 3** FastRegression($\mathbf{X}, y, \theta_0, R_0, \mathcal{O}_{\mathrm{ERM}}, \delta$)

---

1: **Input:** Dataset $\mathbf{X} = \{X_i\}_{i \in [n]} \in \mathbb{R}^{n \times d}$, $y = \{y_i\}_{i \in [n]} \in \mathbb{R}^n$, satisfying Models 1 and 3, and satisfying Assumptions 1, $\theta_0 \in \mathbb{R}^d$ with $\|\theta_0 - \theta^\star\|_{\Sigma^\star} \le R_0$, $\delta \in (0, 1)$, $O(\sigma^2 \epsilon)$-approximate ERM oracle $\mathcal{O}_{\mathrm{ERM}}$.

2: **Output:** With probability $\ge 1 - \delta$, $\theta$ with

$$\|\theta - \theta^\star\|_{\Sigma^\star} = O\left(\sigma\sqrt{\kappa}\epsilon\right).$$

3: Remove all $(X_i, y_i)$ with $\|X_i\|_2 > \sqrt{2nL}$, $n \leftarrow$ new dataset size
4: $w \leftarrow \mathsf{FastCovFilter}(\mathbf{X}, \frac{1}{n}\mathbb{1}, \frac{\delta}{3}, \frac{3}{2}L, 2nL)$, $R \leftarrow R_0 + 4C_{\mathrm{est}}\sigma\sqrt{\kappa}\epsilon$, $\theta \leftarrow \theta_0$
5: $T \leftarrow O(\log \frac{R_0}{\sigma})$ for a sufficiently large constant
6: **while** $R > C_{\mathrm{lp}}\sigma$ **do**
7: $\quad D \leftarrow$ value in (21) with current setting of $R$
8: $\quad$ Remove all $(X_i, y_i)$ not satisfying bound (16), $n \leftarrow$ new dataset size
9: $\quad (w, \theta) \leftarrow \mathsf{HalfRadiusLinReg}(\mathbf{X}, y, w, \theta, R, D, \mathcal{O}_{\mathrm{ERM}}, \frac{\delta}{3T})$
10: $\quad R \leftarrow \frac{1}{2}R$
11: **end while**
12: $D \leftarrow$ value in (21) with current setting of $R$
13: Remove all $(X_i, y_i)$ not satisfying bound (16), $n \leftarrow$ new dataset size
14: $(w, \theta) \leftarrow \mathsf{HalfRadiusLinReg}(\mathbf{X}, y, w, \theta, D, \mathcal{O}_{\mathrm{ERM}}, \frac{\delta}{3})$
15: **return** $\theta$

---

**Theorem 5.** *In Models 1 and 3, under Assumption 1 with $r = \frac{\epsilon^2}{\alpha}$ for $\alpha$ as in (12), supposing $\epsilon\kappa$ is sufficiently small, given $\theta_0 \in \mathbb{R}^d$ and $\|\theta_0 - \theta^\star\|_{\Sigma^\star} \le R_0$, FastRegression returns $\theta$ with $\|\theta - \theta^\star\|_{\Sigma^\star} = O(\sigma\sqrt{\kappa}\epsilon)$ with probability at least $1 - \delta$. The algorithm runs in $O\left(\frac{1}{\epsilon} + \kappa \log \frac{R_0}{\sigma}\right)$ calls to a $O(\sigma^2\epsilon)$-approximate ERM oracle, and*

$$O\left(\left(nd\log^3(n)\log\left(\frac{nR_0}{\sigma\delta\epsilon}\right)\right)\left(\frac{1}{\epsilon} + \kappa\log\frac{R_0}{\sigma}\right)\right) \text{ additional time.}$$

*Proof.* First, correctness of Lines 3, 8, and 13 of the algorithm follow from Lemmas 9 and 10. Also, Line 4 ensures that throughout the algorithm we have $\mathbf{X}^\top \mathbf{diag}(w)\mathbf{X} \preceq 8L\mathbf{I}$, so the preconditions of HalfRadiusLinReg and LastPhase are met. Finally, the initial setting of $R$ is correct by Lemma 3 and the assumed bound $R_0$. The correctness and runtime then follow from applying Lemma 7 $T$ times and Lemma 8 once. The failure probability comes from union bounding over the one call to FastCovFilter, the $T$ calls to HalfRadiusLinReg, and the one call to LastPhase.

Finally, for completeness we check that the promise of Section 3.1 is kept by Algorithm 3. There are at most $\log(\frac{R_0}{\sigma})$ calls to HalfRadiusLinReg, and one call to LastPhase. Combined, this accounts for at most $O(\frac{1}{\epsilon} + \kappa \log \frac{R_0}{\sigma})$ distinct sets we filtered with respect to, in the manner described by Lemma 5. For a sufficiently large $\alpha$ in (12), this is indeed at most $\frac{\alpha}{2\epsilon}$ distinct sets. $\square$

We conclude this section by noting that the guarantees of Proposition 2 imply the sample complexity required for Algorithm 3 to succeed with probability at least $\frac{9}{10} - \delta$ is $n = \widetilde{O}(\frac{d}{\epsilon^4} + \frac{d^2}{\epsilon^3})$.

# 4 Robust acceleration

In this section, we give a general-purpose algorithm for solving statistical optimization problems with a finite condition number $\kappa$ under the strong contamination model. We study the following abstract problem: we wish to minimize a function $F : \mathbb{R}^d \to \mathbb{R}$ which is $L$-smooth and $\mu$-strongly convex with minimizer $\theta_F^\star$, but we only have black-box access to $F$ through a *noisy gradient oracle* $\mathcal{O}_{\mathrm{ng}}$. In particular, we can query $\mathcal{O}_{\mathrm{ng}}$ at any point $\theta \in \mathbb{R}^d$ with a parameter $R \geq \|\theta - \theta_F^\star\|_2$ and receive $G(\theta)$ such that for a universal constant $C_{\mathrm{ng}}$,

$$\|G(\theta) - \nabla F(\theta)\|_2 \leq C_{\mathrm{ng}} \left( \sqrt{L\epsilon}\sigma + L\sqrt{\epsilon}R \right).$$

Notably, our algorithm is *accelerated*, running in a number of iterations depending on $\sqrt{\kappa}$ rather than $\kappa$. It applies to both the regression setting of Section 2.3 and the smooth stochastic optimization setting of Section 2.4. We demonstrate in Section 4.1 how to build a noisy gradient oracle for regression and smooth stochastic optimization settings. In Section 4.2 we then give our main subroutine, an accelerated procedure for halving the radius to the optimizer using noisy gradient oracles. We give our complete algorithm and its applications in Section 4.3. Throughout we assume $\epsilon\kappa^2 < 1$ is sufficiently small.

## 4.1 Noisy gradient oracle

In this section, we build noisy gradient oracles for the problems in Sections 2.3 and 2.4. We now give a formal definition below; the remaining sections will access $F$ through this abstraction.

**Definition 4** (Noisy gradient oracle). We call $\mathcal{O}_{\mathrm{ng}}(\theta, R)$ a $(L, \sigma, \delta)$-*noisy gradient oracle* for $F : \mathbb{R}^d \to \mathbb{R}$ with minimizer $\theta_F^\star$ if on query $\theta \in \mathbb{R}^d$ and given $R \geq \|\theta - \theta_F^\star\|_2$, with probability $\geq 1 - \delta$ it returns $G(\theta)$ satisfying for a universal constant $C_{\mathrm{ng}}$,

$$\|G(\theta) - \nabla F(\theta)\|_2 \leq C_{\mathrm{ng}} \left( \sqrt{L\epsilon}\sigma + L\sqrt{\epsilon}R \right).$$

If the returned $G(\theta)$ always satisfies the stronger bound $\|G(\theta) - \nabla F(\theta)\|_2 \leq C_{\mathrm{ng}}\sqrt{L\epsilon}\sigma$, we call $\mathcal{O}_{\mathrm{ng}}$ a $(L, \sigma, \delta)$-*radiusless noisy gradient oracle*.

Before developing our implementations, we state a useful identifiability result relating gradient estimation to controlling operator norms of gradient second moments for finite sum functions.

**Lemma 11.** *Suppose* $F_G(\theta) = \frac{1}{|G|} \sum_{i \in G} f_i(\theta)$ *for some functions* $\{f_i\}_{i \in G}$, *and let* $w \in \Delta^n$ *be saturated with respect to bipartition* $[n] = G \cup B$. *For* $\tilde{w} := \frac{w}{\|w\|_1}$ *and* $w_G^\star = \frac{1}{|G|}\mathbb{1}_G$, *we have*

$$\left\| \mathbb{E}_{i \sim w_G^\star}[\nabla f_i(\theta)] - \mathbb{E}_{j \sim \tilde{w}}[\nabla f_j(\theta)] \right\|_2 \leq \sqrt{24\epsilon} \left( \left\| \mathrm{Cov}_{w_G^\star}\left( \{\nabla f_i(\theta)\}_{i \in [n]} \right) \right\|_{\mathrm{op}}^{\frac{1}{2}} + \left\| \mathrm{Cov}_{\tilde{w}}\left( \{\nabla f_i(\theta)\}_{i \in [n]} \right) \right\|_{\mathrm{op}}^{\frac{1}{2}} \right).$$

*Proof.* Throughout this proof, we let $\mathcal{C}$ supported on $[n] \times [n]$ be an optimal coupling between

$i \sim w_G^\star$ and $j \sim \tilde{w}$, and follow notation from Proposition 6. For some unit vector $v \in \mathbb{R}^d$, we have

$$\left\langle v, \underset{i \sim w_G^\star}{\mathbb{E}}[\nabla f_i(\theta)] - \underset{j \sim \tilde{w}}{\mathbb{E}}[\nabla f_j(\theta)] \right\rangle = \underset{i,j \sim \mathcal{C}}{\mathbb{E}}[\langle v, \nabla f_i(\theta) - \nabla f_j(\theta)\rangle]$$

$$= \underset{i,j \sim \mathcal{C}}{\mathbb{E}}[\langle v, \nabla f_i(\theta) - \nabla f_j(\theta)\rangle \mathbb{1}_{i \neq j}]$$

$$\leq \sqrt{6\epsilon} \underset{i,j \sim \mathcal{C}}{\mathbb{E}}\left[\langle v, \nabla f_i(\theta) - \nabla f_j(\theta)\rangle^2\right]^{\frac{1}{2}}$$

$$\leq \sqrt{24\epsilon}\left(\underset{i \sim w_G^\star}{\mathbb{E}}\left[\langle v, \nabla f_i(\theta)\rangle^2\right]^{\frac{1}{2}} + \underset{j \sim \tilde{w}}{\mathbb{E}}\left[\langle v, \nabla f_j(\theta)\rangle^2\right]^{\frac{1}{2}}\right).$$

The conclusion follows from choosing $v$ to be in the direction of $\mathbb{E}_{i \sim w_G^\star}[\nabla f_i(\theta)] - \mathbb{E}_{j \sim \tilde{w}}[\nabla f_j(\theta)]$, and using the definition of the operator norm. $\qquad\square$

Lemma 11 implies that for approximating gradients of functions $F$ which are "closely approximated" by an (unknown) finite sum function $F_G$, it suffices to find a weighting $\tilde{w}$ such that the operator norm of $\mathrm{Cov}_{\tilde{w}}$ applied to gradients is bounded. We now demonstrate applications of this strategy to linear regression and smooth stochastic optimization.

**Corollary 1.** *Consider a robust linear regression instance where we have sample access to datasets* $\mathbf{X} = \{X_i\}_{i \in [n]} \in \mathbb{R}^{n \times d}$ *and* $y = \{y_i\}_{i \in [n]} \in \mathbb{R}^n$ *under Models 1, 4, with sample size $n$ corresponding to Proposition 3. For*

$$F(\theta) = F^\star(\theta) = \underset{X,y \sim \mathcal{D}_{Xy}}{\mathbb{E}}\left[\frac{1}{2}(\langle X_i, \theta\rangle - y_i)^2\right],$$

*we can construct a $(L, \sigma, \delta)$-noisy gradient oracle for $F$ in $O\left(nd \log^3(n) \log^2\left(\frac{n}{\delta}\right)\right)$ time, using $O(\log \frac{1}{\delta})$ queries of samples from Proposition 3.*

*Proof.* We first demonstrate how to construct a noisy gradient oracle with success probability $\geq \frac{8}{10}$. The algorithm is as follows: first, sample a dataset under Models 1, 4, according to Proposition 3. Then, at point $\theta \in \mathbb{R}^d$, with probability $\frac{9}{10}$ Assumption 2 gives us a set $G := G_\theta$ (where we drop the subscript for simplicity, as this proof only discusses a single $\theta$) with $|G| = (1 - \epsilon)$ such that (10) holds. If we have the promise $\|\theta - \theta^*\|_2 \leq R$, let $w \in \Delta^n$ be the output of

$$\mathsf{FastCovFilter}\left(\{g_i(\theta)\}_{i \in [n]}, \frac{1}{n}\mathbb{1}, \frac{9}{10}, C_{\mathrm{est}}\left(L\sigma^2 + LR^2\right)\right), \text{ where } g_i(\theta) := X_i\left(\langle X_i, \theta\rangle - y_i\right).$$

where $\mathsf{FastCovFilter}$ (Algorithm 7) is the algorithm of Proposition 5. We then output $\mathbb{E}_{j \sim \tilde{w}}[g_j(\theta)]$, where $\tilde{w} = \frac{w}{\|w\|_1}$. The runtime is $O(nd \log^4 n)$ from the bottleneck operation of running $\mathsf{FastCovFilter}$. The assumptions of $\mathsf{FastCovFilter}$, namely a bound on $\mathrm{Cov}_{w_G^\star}\left(\{g_i(\theta)\}_{i \in [n]}\right)$, are satisfied by Assumption 2.2. Guarantees of $\mathsf{FastCovFilter}$ and Lemma 11 then imply

$$\left\|\underset{i \sim w_G^\star}{\mathbb{E}}[g_i(\theta)] - \underset{j \sim \tilde{w}}{\mathbb{E}}[g_j(\theta)]\right\|_2 = O\left(\sqrt{L\epsilon}\sigma + L\sqrt{\epsilon}R\right).$$

The conclusion follows from Assumption 2.2 which bounds $\left\|\mathbb{E}_{i \sim w_G^\star}[g_i(\theta)] - \nabla F(\theta)\right\|_2$.

We now describe how to boost the success probability, by calling our sample access $T = O(\log \frac{1}{\delta})$ times. Let $G^\star := \nabla F(\theta)$ be the true gradient, and run the procedure described above $T$ times, producing $\{G_t\}_{t \in [T]}$, such that each $G_t$ satisfies $\|G_t - G^\star\|_2 \leq C(\sqrt{L\epsilon}\sigma + L\sqrt{\epsilon}R)$ with probability

at least $\frac{4}{5}$, for some constant $C$. By standard binomial concentration, with probability at least $1 - \delta$, at least $\frac{3}{5}$ of the $\{G_t\}_{t \in [T]}$ will satisfy this bound; call such a satisfying $G_t$ "good." We return any $G_t$ which is within distance $2C(\sqrt{L\epsilon}\sigma + L\sqrt{\epsilon}R)$ from at least $\frac{3}{5}$ of the gradient estimates. Note this will never return any $G_t$ with $\|G_t - G^\star\|_2 > 4C(\sqrt{L\epsilon}\sigma + L\sqrt{\epsilon}R)$, since the triangle inequality implies this $G_t$ will miss all the good estimates, a contradiction since there is at most a $\frac{2}{5}$ fraction which is not good. Thus, this procedure satisfies the requirements with $C_{\mathrm{ng}} = 4C$; the additional runtime overhead is $O(\log^2(\frac{1}{\delta}))$ distance comparisons between our gradient estimates. $\qquad\square$

**Corollary 2.** *Consider a robust Lipschitz (not necessarily smooth) stochastic optimization instance where we have sample access to datasets* $\mathbf{X} = \{X_i\}_{i \in [n]} \in \mathbb{R}^{n \times d}$ *and* $y = \{y_i\}_{i \in [n]} \in \mathbb{R}^n$ *under Models 1, 2, 6 with sample size $n$ corresponding to Proposition 4. For*

$$F(\theta) = F^\star(\theta) + \frac{\mu}{2} \|\theta\|_2^2 = \underset{f \sim \mathcal{D}_f}{\mathbb{E}} [f(\theta)] + \frac{\mu}{2} \|\theta\|_2^2,$$

*we can construct a $(L, 1, \delta)$-radiusless noisy gradient oracle for $F$ in $O\left(nd \log^3(n) \log^2\left(\frac{n}{\delta}\right)\right)$ time, using $O(\log \frac{1}{\delta})$ queries of samples from Proposition 4.*

*Proof.* The proof follows identically to that of Corollary 1, where we use Assumption 3.2 in place of Assumption 2.2. $\qquad\square$

In most of Sections 4.2 and 4.3, we will no longer discuss any specifics of the unknown function $F : \mathbb{R}^d \to \mathbb{R}$ we wish to optimize, except that it is $L$-smooth, $\mu$-strongly convex, has minimizer $\theta_F^\star$, and supports a noisy gradient oracle $\mathcal{O}_{\mathrm{ng}}$. We will apply Corollaries 1 and 2 to derive concrete rates and sample complexities for specific applications at the conclusion of this section.

## 4.2 Halving the distance to $\theta_F^\star$

In this section, we give a subroutine used in our full algorithm, which halves the distance to $\theta_F^\star$, the minimizer of $F$, outside of a sufficiently large radius. Suppose that we have an initial point $\bar{\theta} \in \mathbb{R}^d$, as well as a sufficiently large scalar $R \geq 0$ with the promise that (for a universal constant $C_{\mathrm{lp}}$)

$$\left\|\bar{\theta} - \theta_F^\star\right\|_2 \leq R, \text{ where } R \geq C_{\mathrm{lp}} \sigma \sqrt{\frac{\kappa \epsilon}{\mu}}. \tag{22}$$

We begin by stating a standard lemma from convex analysis, following from first-order optimality.

**Lemma 12** (Proximal three-point inequality). *Let $f$ be a convex function, and let $\mathcal{S}$ be a convex set. For any point $y \in \mathcal{S}$, define $\mathrm{Prox}(y) := \mathrm{argmin}_{x \in \mathcal{S}} \left\{ f(x) + \|x - y\|_2^2 \right\}$. Then if $x = \mathrm{Prox}(y)$,*

$$\langle \nabla f(x), x - u \rangle \leq \frac{1}{2} \left( \|y - u\|_2^2 - \|x - u\|_2^2 - \|x - y\|_2^2 \right), \text{ for all } u \in \mathcal{S}.$$

We now state a procedure, HalfRadiusAccel, which returns a new $\theta$ with $\|\theta - \theta_F^\star\|_2 \leq \frac{1}{2}R$. In its statement, we define a sequence of scalars $\{a_t, A_t\}_{0 \leq t < T}$ given by the recursions

$$A_0 = 0, \ A_t = 3a_t^2, \ A_{t+1} = A_t + a_{t+1}. \tag{23}$$

The following fact is well-known (see for instance Chapter 2.2 of [Nes03]).

**Fact 1.** *For all $0 \leq t < T$, $A_t = \Theta(t^2)$ and $a_t = \Theta(t)$.*

---

**Algorithm 4** HalfRadiusAccel($\bar{\theta}, R, \mathcal{O}_{\mathrm{ng}}, \delta$)

---

1: **Input:** $\mathcal{O}_{\mathrm{ng}}$, a $(L, \sigma, \frac{\delta}{T})$-noisy gradient oracle for $L$-smooth, $\mu$-strongly convex $F : \mathbb{R}^d \to \mathbb{R}$ with minimizer $\theta_F^\star$, $\bar{\theta} \in \mathbb{R}^d$, $R \in \mathbb{R}_{\geq 0}$ satisfying (22), and $T = O(\sqrt{\kappa})$, $\delta \in (0, 1)$
2: **Output:** With probability $\geq 1 - \delta$, $\theta \in \mathbb{R}^d$ with $\|\theta - \theta_F^\star\|_2 \leq \frac{1}{2}R$
3: $T \leftarrow O\left(\sqrt{\kappa}\right)$ for a sufficiently large constant, $t \leftarrow 0$, $\theta_0 \leftarrow \bar{\theta}$, $v_0 \leftarrow \bar{\theta}$
4: **while** $t < T$ **do**
5: $\quad y_t \leftarrow \frac{A_t}{A_{t+1}}\theta_t + \frac{a_{t+1}}{A_{t+1}}v_t$
6: $\quad g_t \leftarrow \mathcal{O}_{\mathrm{ng}}(y_t, 2R)$
7: $\quad \theta_{t+1} \leftarrow \operatorname{argmin}_{\theta \in \mathbb{B}}\left\{\frac{1}{3L}\langle g_t, \theta\rangle + \frac{1}{2}\|\theta - y_t\|_2^2\right\}$, where $\mathbb{B} := \{\theta \mid \|\theta - \bar{\theta}\|_2 \leq R\}$
8: $\quad v_{t+1} \leftarrow \operatorname{argmin}_{v \in \mathbb{B}}\left\{\frac{a_{t+1}}{L}\langle g_t, v\rangle + \frac{1}{2}\|v - v_t\|_2^2\right\}$
9: $\quad t \leftarrow t + 1$
10: **end while**
11: **return** $\theta_t$

---

We remark throughout we assume that Lines 7 and 8 are implemented exactly for simplicity; it is straightforward to verify from the proof of Lemma 13 that it suffices to implement these steps to inverse-polynomial precision in problem parameters. This can be done by a standard binary search (see e.g. Proposition 8 of [CJJ+20]), and is not the bottleneck operation compared to calling $\mathcal{O}_{\mathrm{ng}}$.

We give the main technical lemma of this section, which shows a potential bound on iterates of HalfRadiusAccel. Our proof is based on a standard analysis of accelerated methods by [AO17].

**Lemma 13.** *In every iteration $0 \leq t \leq T$, define*

$$E_t := F(\theta_t) - F(\theta_F^\star), \; D_t := \frac{1}{2}\|v_t - \theta_F^\star\|_2^2, \; \Phi_t := \frac{A_t}{L}E_t + D_t.$$

*Then, for all $0 \leq t < T$, for a universal constant $C_{\mathrm{pot}}$,*

$$\Phi_{t+1} - \Phi_t \leq C_{\mathrm{pot}}\left(t\sigma\sqrt{\frac{\epsilon}{L}}R + t\sqrt{\epsilon}R^2 + t^2\sigma^2\frac{\epsilon}{L} + t^2\epsilon R^2\right).$$

*Proof.* Throughout this proof Lemma 12 will be applied to the set $\mathcal{S} = \mathbb{B}$. Fix an iteration $t$. We observe that $\theta_t$ and $v_t$ lie in $\mathbb{B}$ by the constraints on Lines 7 and 8: therefore $y_t \in \mathbb{B}$. As $\|\bar{\theta} - \theta_F^\star\|_2 \leq R$, the triangle inequality yields $\|y_t - \theta_F^\star\|_2 \leq \|\bar{\theta} - \theta_F^\star\|_2 + \|\bar{\theta} - y_t\|_2 \leq 2R$: thus by the guarantee of $\mathcal{O}_{\mathrm{ng}}$ we have, for some $C_{\mathrm{ng}} = O(1)$ (adjusting the definition of $C_{\mathrm{ng}}$ by a constant)

$$\left\|g_t - \nabla F\left(\theta_{t+1}^\star\right)\right\|_2 \leq C_{\mathrm{ng}}\left(\sqrt{L\epsilon}\sigma + L\sqrt{\epsilon}R\right).$$

For convenience, we define $\rho = C_{\mathrm{ng}}\left(\sqrt{L\epsilon}\sigma + L\sqrt{\epsilon}R\right)$. We define the helper function

$$\mathsf{Prog}(y; g) = \min_{\theta \in \mathbb{B}}\left\{\langle g, \theta - y\rangle + \frac{3L}{2}\|\theta - y\|_2^2\right\},$$

and observe

$$
\begin{aligned}
\mathsf{Prog}(y_t; g_t) &= \min_{\theta \in \mathbb{B}} \left\{ \langle g_t, \theta - y_t \rangle + \frac{3L}{2} \|\theta - y_t\|_2^2 \right\} \\
&\overset{(i)}{=} \frac{3L}{2} \|\theta_{t+1} - y_t\|^2 + \langle g_t, \theta_{t+1} - y_t \rangle \\
&= \left( \frac{L}{2} \|\theta_{t+1} - y_t\|^2 + \langle \nabla F(y_t), \theta_{t+1} - y_t \rangle \right) + L \|\theta_{t+1} - y_t\|_2^2 - \langle \nabla F(y_t) - g_t, \theta_{t+1} - y_t \rangle \\
&\overset{(ii)}{\geq} F(\theta_{t+1}) - F(y_t) + L \|\theta_{t+1} - y_t\|_2^2 - \langle \nabla F(y_t) - g_t, \theta_{t+1} - y_t \rangle \\
&\overset{(iii)}{\geq} F(\theta_{t+1}) - F(y_t) - \frac{1}{4L} \|\nabla F(y_t) - g_t\|_2^2 \geq F(\theta_{t+1}) - F(y_t) - \frac{\rho^2}{4L}.
\end{aligned}
\tag{24}
$$

Here $(i)$ is by the optimality of $\theta_{t+1}$, $(ii)$ holds via the $L$-smoothness of $F$, and $(iii)$ follows from Young's inequality $\langle a, b \rangle - \frac{1}{2} \|a\|_2^2 \leq \frac{1}{2} \|b\|_2^2$ and our bound on $\|\nabla F(y_t) - g_t\|_2$. Next, we note

$$
\frac{a_{t+1}}{L} \langle g_t, v_{t+1} - \theta_F^\star \rangle \leq \frac{1}{2} \left( \|v_t - \theta_F^\star\|_2^2 - \|v_{t+1} - \theta_F^\star\|_2^2 - \|v_t - v_{t+1}\|_2^2 \right)
\tag{25}
$$

by the proximal three-point inequality Lemma 12 on Line 8. Define

$$
\widetilde{y}_t = \frac{A_t}{A_{t+1}} \theta_t + \frac{a_{t+1}}{A_{t+1}} v_{t+1},
$$

and note that $y_t - \widetilde{y}_t = \frac{a_{t+1}}{A_{t+1}} (v_t - v_{t+1})$ and $\widetilde{y}_t \in \mathbb{B}$ by convexity. Consequently,

$$
\begin{aligned}
\frac{a_{t+1}}{L} \langle g_t, v_t - \theta_F^\star \rangle &= \frac{a_{t+1}}{L} \langle g_t, v_t - v_{t+1} \rangle + \frac{a_{t+1}}{L} \langle g_t, v_{t+1} - \theta_F^\star \rangle \\
&\overset{(i)}{\leq} \frac{a_{t+1}}{L} \langle g_t, v_t - v_{t+1} \rangle + \frac{1}{2} \left( \|v_t - \theta_F^\star\|_2^2 - \|v_{t+1} - \theta_F^\star\|_2^2 - \|v_t - v_{t+1}\|_2^2 \right) \\
&\overset{(ii)}{=} \frac{A_{t+1}}{L} \langle g_t, y_t - \widetilde{y}_t \rangle - \frac{A_{t+1}^2}{2a_{t+1}^2} \|y_t - \widetilde{y}_t\|_2^2 + D_t - D_{t+1} \\
&\overset{(iii)}{=} \frac{A_{t+1}}{L} \left( \langle g_t, y_t - \widetilde{y}_t \rangle - \frac{3L}{2} \|y_t - \widetilde{y}_t\|_2^2 \right) + D_t - D_{t+1} \\
&\overset{(iv)}{\leq} -\frac{A_{t+1}}{L} \mathsf{Prog}(y_t; g_t) + D_t - D_{t+1} \\
&\overset{(v)}{\leq} -\frac{A_{t+1}}{L} \left( F(\theta_{t+1}) - F(y_t) - \frac{\rho^2}{4L} \right) + D_t - D_{t+1}.
\end{aligned}
$$

Here $(i)$ uses (25), $(ii)$ uses the definition of $\widetilde{y}_t$, $(iii)$ uses that $A_{t+1} = 3a_{t+1}^2$, $(iv)$ uses the definition

of Prog, and $(v)$ uses our lower bound on $\mathsf{Prog}(y_t; g_t)$ (24). Finally, by convexity of $F$ we have

$$
\begin{aligned}
\frac{a_{t+1}}{L} \left( F(y_t) - F(\theta_F^\star) \right) &\leq \frac{a_{t+1}}{L} \left\langle \nabla F(y_t), y_t - \theta_F^\star \right\rangle \\
&= \frac{a_{t+1}}{L} \left\langle \nabla F(y_t), y_t - v_t \right\rangle + \frac{a_{t+1}}{L} \left\langle \nabla F(y_t), v_t - \theta_F^\star \right\rangle \\
&\overset{(i)}{=} \frac{A_t}{L} \left\langle \nabla F(y_t), \theta_t - y_t \right\rangle + \frac{a_{t+1}}{L} \left\langle \nabla F(y_t), v_t - \theta_F^\star \right\rangle \\
&\leq \frac{A_t}{L} \left( F(\theta_t) - F(y_t) \right) + \frac{a_{t+1}}{L} \left\langle g_t, v_t - \theta_F^\star \right\rangle + \frac{a_{t+1}}{L} \left\langle \nabla F(y_t) - g_t, v_t - \theta_F^\star \right\rangle \\
&\overset{(ii)}{\leq} \frac{A_t}{L} \left( F(\theta_t) - F(y_t) \right) + \frac{a_{t+1}}{L} \left\langle g_t, v_t - \theta_F^\star \right\rangle + \frac{a_{t+1}}{L} \left\| \nabla F(y_t) - g_t \right\|_2 \left\| v_t - \theta_F^\star \right\|_2 \\
&\overset{(iii)}{\leq} \frac{A_t}{L} \left( F(\theta_t) - F(y_t) \right) + \frac{a_{t+1}}{L} \left\langle g_t, v_t - \theta_F^\star \right\rangle + \frac{2a_{t+1}\rho R}{L}
\end{aligned}
$$

where $(i)$ used $y_t - v_t = \frac{A_t}{a_{t+1}} (\theta_t - y_t)$, $(ii)$ used the Cauchy-Schwarz inequality, and $(iii)$ used that $\left\| v_t - \theta_F^\star \right\|_2 \leq \left\| v_t - \bar\theta \right\|_2 + \left\| \bar\theta - \theta_F^\star \right\|_2 \leq 2R$. Combining the above two equations and rearranging,

$$
\begin{aligned}
\frac{a_{t+1}}{L} \left( F(y_t) - F(\theta_F^\star) \right) + \frac{A_{t+1}}{L} \left( F(\theta_{t+1}) - F(y_t) \right) &\leq \frac{A_t}{L} \left( F(\theta_t) - F(y_t) \right) \\
&\quad + D_t - D_{t+1} + \frac{2a_{t+1}\rho R}{L} + \frac{A_{t+1}\rho^2}{4L^2}.
\end{aligned}
$$

Adding $\frac{A_t}{L} \left( F(y_t) - F(\theta_F^\star) \right)$ to both sides, we obtain

$$
\Phi_{t+1} - \Phi_t \leq \frac{2a_{t+1}\rho R}{L} + \frac{A_{t+1}\rho^2}{4L^2}.
$$

Finally, applying Fact 1, we see that this potential increase is indeed bounded as

$$
\frac{2a_{t+1}\rho R}{L} + \frac{A_{t+1}\rho^2}{4L^2} = O\left( \frac{t\rho R}{L} + \frac{t^2\rho^2}{L^2} \right) = O\left( t\sigma\sqrt{\frac{\epsilon}{L}} R + t\sqrt{\epsilon}R^2 + t^2\sigma^2\frac{\epsilon}{L} + t^2\epsilon R^2 \right).
$$

$\square$

Finally, we are ready to analyze the output of HalfRadiusAccel.

**Lemma 14.** *With probability at least $1 - \delta$, the output $\theta_T$ of* HalfRadiusAccel *satisfies*

$$
\left\| \theta_T - \theta_F^\star \right\|_2 \leq \frac{1}{2}R.
$$

*Proof.* The failure probability comes from union bounding over the failures of calls to $\mathcal{O}_{\mathrm{ng}}$, so we discuss correctness assuming all calls succeed. By telescoping Lemma 13 over $T$ iterations, we have

$$
\Phi_T \leq \Phi_0 + C_{\mathrm{pot}} \sum_{t \in [T]} \left( t\sigma\sqrt{\frac{\epsilon}{L}} R + t\sqrt{\epsilon}R^2 + t^2\sigma^2\frac{\epsilon}{L} + t^2\epsilon R^2 \right).
$$

It is clear from definition that $\Phi_0 \leq \frac{1}{2}R^2$, so it remains to bound all other terms. By examination,

$$\sum_{t \in [T]} t\sigma\sqrt{\frac{\epsilon}{L}}R = O\left(\sigma\kappa\sqrt{\frac{\epsilon}{L}}R\right) = O\left(R^2\right),$$

$$\sum_{t \in [T]} t\sqrt{\epsilon}R^2 = O\left(\kappa\sqrt{\epsilon}R^2\right) = O\left(R^2\right),$$

$$\sum_{t \in [T]} t^2\sigma^2\frac{\epsilon}{L} = O\left(\kappa^{1.5}\sigma^2\frac{\epsilon}{L}\right) = O\left(R^2\right),$$

$$\sum_{t \in [T]} t^2\epsilon R^2 = O\left(\kappa^{1.5}\epsilon R^2\right) = O\left(R^2\right).$$

Each of the above lines follows from $\epsilon\kappa^2$ being sufficiently small, and the lower bound in (22). Thus for a large enough value of $C_{\mathrm{lp}}$ in (22), we have that $\Phi_T \leq R^2$. Since $\Phi_T = A_T E_T + D_T \geq A_T E_T$, choosing a sufficiently large value of $T = \sqrt{\kappa}$ combined with Fact 1 yields

$$E_T \leq \frac{R^2}{A_T} \leq \frac{LR^2}{8\kappa} = \frac{\mu R^2}{8}.$$

The conclusion follows from strong convexity of $F$, which implies $E_T \geq \frac{\mu}{2}\|\theta_T - \theta_F^\star\|_2^2$.

*A note on constants.* To check there are no conflict of interests hidden in the constants of this proof, note first that conditional on the bound at time $T$ being at most $R^2$, the number of iterations $T$ can be chosen solely as a function of the constants in Fact 1. From this point, the constant $C_{\mathrm{lp}}$ in (22) can be chosen to ensure that the potential bound is indeed $R^2$. $\square$

### 4.3 Full accelerated algorithm

We conclude this section with a statement of a complete accelerated algorithm, and its applications.

---
**Algorithm 5** RobustAccel($\theta_0, R_0, \mathcal{O}_{\mathrm{ng}}, \delta$)

---
1: **Input:** $\mathcal{O}_{\mathrm{ng}}$, a $(L, \sigma, \frac{\delta}{N})$-noisy gradient oracle for $L$-smooth, $\mu$-strongly convex $F : \mathbb{R}^d \to \mathbb{R}$ with minimizer $\theta_F^\star$ for $N = O\left(\sqrt{\kappa}\log\left(\frac{R_0\sqrt{L}}{\sigma\sqrt{\epsilon}}\right)\right)$, $\theta_0 \in \mathbb{R}^d$ with $\|\theta_0 - \theta_F^\star\|_2 \leq R_0$, $\delta \in (0, 1)$
2: **Output:** With probability $\geq 1 - \delta$, $\theta \in \mathbb{R}^d$ with

$$\|\theta - \theta_F^\star\|_2 = O\left(\sigma\sqrt{\frac{\kappa\epsilon}{\mu}}\right). \tag{26}$$

3: $T \leftarrow O\left(\log\left(\frac{R_0\sqrt{\mu}}{\sigma\sqrt{\kappa\epsilon}}\right)\right)$ for a sufficiently large constant, $t \leftarrow 0$
4: **while** $t < T$ **do**
5: $\quad \theta_{t+1} \leftarrow \mathsf{HalfRadiusAccel}(\theta_t, R_t, \mathcal{O}_{\mathrm{ng}}, \frac{\delta}{T})$
6: $\quad R_{t+1} \leftarrow \frac{1}{2}R_t$
7: $\quad t \leftarrow t + 1$
8: **end while**
9: **return** $\theta_t$

---

**Proposition 7.** RobustAccel correctly returns $\theta$ satisfying (26) with probability $\geq 1 - \delta$. It runs

in $O\left(\sqrt{\kappa}\log\left(\frac{R_0\sqrt{L}}{\sigma\sqrt{\epsilon}}\right)\right)$ calls to $\mathcal{O}_{\mathrm{ng}}$, and $O\left(\sqrt{\kappa}d\log\left(\frac{R_0\sqrt{L}}{\sigma\sqrt{\epsilon}}\right)\right)$ additional time.

*Proof.* Correctness is immediate by iterating the guarantees of Lemma 14 $T$ times, and taking a union bound over all (at most $N$) calls to $\mathcal{O}_{\mathrm{ng}}$, the only source of randomness in the algorithm. The runtime follows from examining HalfRadiusAccel and $\mathcal{O}_{\mathrm{ng}}$, since all operations take $O(d)$ time other than calls to $\mathcal{O}_{\mathrm{ng}}$. $\qquad\square$

By combining Proposition 7 with Corollary 1 and 2, we derive the following conclusions.

**Theorem 6.** *Under Models 1, 4, supposing $\epsilon\kappa^2$ is sufficiently small, given $\theta_0 \in \mathbb{R}^d$ and $\|\theta_0 - \theta^\star\|_{\boldsymbol{\Sigma}^\star} \le R_0$, RobustAccel using the noisy gradient oracle of Corollary 1 returns $\theta$ with $\|\theta - \theta^\star\|_{\boldsymbol{\Sigma}^\star} = O\left(\sigma\kappa\sqrt{\epsilon}\right)$ with probability at least $1 - \delta$. The algorithm runs in*

$$O\left(nd\sqrt{\kappa}\log\left(\frac{R_0}{\sigma\sqrt{\epsilon}}\right)\log^3(n)\log^2\left(\frac{n\log\left(\frac{R_0}{\sigma}\right)}{\delta\epsilon}\right)\right) \ \ time,$$

*where $n$ is the dataset size of Proposition 3. The sample complexity of the method is*

$$O\left(\log\left(\frac{\log\left(\frac{R_0}{\sigma}\right)}{\delta\epsilon}\right) \cdot \left(\frac{d\log(d/\epsilon)}{\epsilon}\right)\right).$$

**Theorem 7.** *Under Models 1, 2, and 5, supposing $\epsilon\kappa^2$ is sufficiently small for $\kappa := \max(1, \frac{L}{\mu})$, given $\theta_0 \in \mathbb{R}^d$ and $\left\|\theta_0 - \theta^\star_{\mathrm{reg}}\right\|_2 \le R_0$, RobustAccel using the noisy gradient oracle of Corollary 2 returns $\theta$ with $\left\|\theta - \theta^\star_{\mathrm{reg}}\right\|_2 = O\left(\sqrt{\frac{\kappa\epsilon}{\mu}}\right)$ with probability at least $1 - \delta$. The algorithm runs in*

$$O\left(nd\sqrt{\kappa}\log\left(\frac{R_0\sqrt{L+\mu}}{\epsilon}\right)\log^3(n)\log^2\left(\frac{n\log\left(R_0\sqrt{L+\mu}\right)}{\delta\epsilon}\right)\right) \ \ time,$$

*where $n$ is the dataset size of Proposition 4. The sample complexity of the method is*

$$O\left(\log\left(\frac{\log\left(R_0\sqrt{L+\mu}\right)}{\delta\epsilon}\right) \cdot \left(\frac{d\log(d/\epsilon)}{\epsilon}\right)\right).$$

We remark that Theorem 6 can afford to reuse the same samples to construct gradient estimates for all $\theta$ we query per Assumption 2: though the set $G_\theta$ may change per $\theta$, our noisy estimator also provides a per-$\theta$ guarantee (and hence does not require the set to be consistent across calls).

## 5   Lipschitz generalized linear models

In this section, we give an algorithm for minimizing the regularized Moreau envelopes of Lipschitz statistical optimization problems under the strong contamination model, following the exposition of Section 2.4. Concretely, we recall we wish to compute an approximation to

$$\theta^\star_{\mathrm{env}} := \mathrm{argmin}_{\theta\in\mathbb{R}^d}\left\{F^\star_\lambda(\theta) + \frac{\mu}{2}\|\theta\|_2^2\right\}, \ \ \text{where } F^\star(\theta) = \underset{f\sim\mathcal{D}_f}{\mathbb{E}}[f(\theta)],$$

$$\text{and } F^\star_\lambda(\theta) := \inf_{\theta'}\left\{F^\star(\theta') + \frac{1}{2\lambda}\left\|\theta - \theta'\right\|_2^2\right\}. \tag{27}$$

Recall that in Corollary 2, we developed a noisy gradient oracle for $F^\star$, as long as our function distribution is captured by Model 6. However, techniques of Section 4 do not immediately apply to this setting, as $F^\star$ is not smooth. On the other hand, we do not have direct access to $F_\lambda^\star$.

We ameliorate this by developing a noisy gradient oracle (Definition 4) for the Moreau envelope $F_\lambda^\star$ in Section 5.1, under only Assumption 3; this will allow us to apply the acceleration techniques of Section 4 to the problem (27), which we complete in Section 5.2. Interestingly, our noisy gradient oracle for $F_\lambda^\star$ will have noise and runtime guarantees *independent* of the envelope parameter $\lambda$, allowing for a range of statistical and runtime tradeoffs for applications.

## 5.1 Noisy gradient oracle for the Moreau envelope

In this section, we give an efficient reduction which enables the construction of a noisy gradient oracle for $F_\lambda^\star$, assuming a radiusless noisy gradient oracle for $F^\star$, and that $F^\star$ is Lipschitz. We note that both of these assumptions hold under Model 6: we showed in Lemma 4 that $F^\star$ is $\sqrt{L}$-Lipschitz, and constructed a radiusless noisy gradient oracle in Corollary 2.

To begin, we recall standard facts about the Moreau envelope $F_\lambda^\star$, which can be found in e.g. [PB14].

**Fact 2.** $F_\lambda^\star$ *is $\lambda^{-1}$-smooth, satisfies $0 \le F^\star(\theta) - F_\lambda^\star(\theta) \le L\lambda$ for all $\theta \in \mathbb{R}^d$, and has gradient*

$$\nabla F_\lambda^\star(\theta) = \frac{1}{\lambda}\left(\theta - \mathrm{prox}_{\lambda,F^\star}(\theta)\right), \ \text{where} \ \mathrm{prox}_{\lambda,F^\star}(\theta) := \mathrm{argmin}_{\theta'}\left\{F^\star(\theta') + \frac{1}{2\lambda}\left\|\theta - \theta'\right\|_2^2\right\}.$$

Fact 2 demonstrates that to construct a noisy gradient oracle for $F_\lambda^\star$, it suffices to approximate the minimizer of the subproblem defining the $\mathrm{prox}_{\lambda,F^\star}$ operator. To this end, we give a simple algorithm which approximates this proximal minimizer, based on noisy projected gradient descent.

---

**Algorithm 6** ApproxMoreauMinimizer$(\bar\theta, \mathcal{O}_{\mathrm{ng}}, \delta)$

1: **Input:** $\mathcal{O}_{\mathrm{ng}}$, a $(L, 1, \frac{\delta}{T})$-radiusless noisy gradient oracle for $\sqrt{L}$-Lipschitz $F^\star : \mathbb{R}^d \to \mathbb{R}$ for $T = O\left(\frac{1}{\epsilon}\log\frac{1}{\epsilon}\right)$, $\bar\theta \in \mathbb{R}^d$, $\delta \in (0,1)$
2: **Output:** With probability $\ge 1 - \delta$, $\hat\theta$ satisfying for a universal constant $C_{\mathrm{env}}$,

$$\left\|\hat\theta - \mathrm{prox}_{\lambda,F^\star}(\bar\theta)\right\|_2 \le C_{\mathrm{env}}\sqrt{L\epsilon}\lambda. \tag{28}$$

3: $T \leftarrow O\left(\frac{1}{\epsilon}\log\frac{1}{\epsilon}\right)$ for a sufficiently large constant, $t \leftarrow 0$, $\theta_0 \leftarrow \bar\theta$, $\eta \leftarrow \frac{4C_{\mathrm{ng}}^2\epsilon\lambda}{5}$
4: **while** $t < T$ **do**
5: $\quad g_t \leftarrow \mathcal{O}_{\mathrm{ng}}(\theta_t, \infty, \frac{\delta}{T}) + \frac{1}{\gamma}(\theta_t - \bar\theta)$
6: $\quad \theta_{t+1} \leftarrow \mathrm{Proj}_{\mathbb{B}}(\theta_t - \eta g_t)$, where Proj is the $\ell_2$ projection and $\mathbb{B} := \left\{\theta \mid \left\|\theta - \bar\theta\right\|_2 \le 2\sqrt{L}\lambda\right\}$
7: $\quad t \leftarrow t + 1$
8: **end while**
9: **return** $\theta_t$

---

We now begin our analysis of ApproxMoreauMinimizer. In the following discussion, for notational simplicity define $\theta_{\bar\theta}^\star := \mathrm{prox}_{\lambda,F^\star}(\bar\theta)$ to be the exact minimizer of the proximal subproblem. We require a simple helper bound showing $\theta_{\bar\theta}^\star$ does not lie too far from $\bar\theta$.

**Lemma 15.** *For $\mathbb{B} := \left\{\theta \mid \left\|\theta - \bar\theta\right\|_2 \le 2\sqrt{L}\lambda\right\}$ and $\theta_{\bar\theta}^\star := \mathrm{prox}_{\lambda,F^\star}(\bar\theta)$, $\theta_{\bar\theta}^\star \in \mathbb{B}$.*

*Proof.* Let $R := \left\| \theta_{\bar{\theta}}^{\star} - \bar{\theta} \right\|_2$. Since $\theta_{\bar{\theta}}^{\star}$ minimizes the proximal subproblem and $F^{\star}$ is convex,

$$F^{\star}\left(\theta_{\bar{\theta}}^{\star}\right) + \frac{R^2}{2\lambda} \le F^{\star}\left(\bar{\theta}\right) \le F^{\star}\left(\theta_{\bar{\theta}}^{\star}\right) + \left\langle \nabla F^{\star}\left(\bar{\theta}\right), \theta_{\bar{\theta}}^{\star} - \bar{\theta} \right\rangle \implies \frac{R^2}{2\lambda} \le \sqrt{L}R.$$

$\square$

We now prove correctness of ApproxMoreauMinimizer.

**Lemma 16.** ApproxMoreauMinimizer *correctly computes* $\hat{\theta}$ *satisfying* (28) *in* $O\left(\frac{1}{\epsilon}\log\frac{1}{\epsilon}\right)$ *calls to* $\mathcal{O}_{\mathrm{ng}}$, *with probability* $\ge 1 - \delta$.

*Proof.* We assume throughout correctness of all calls to $\mathcal{O}_{\mathrm{ng}}$, which follows from a union bound. Consider some iteration $0 \le t < T$, and let $\hat{\theta}_{t+1} := \theta_t - \eta g_t$ be the unprojected iterate. Since Euclidean projections decrease distances to points within a set (see e.g. Lemma 3.1, [Bub15]), letting $g_t = e_t + \nabla F^{\star}(\theta_t) + \frac{1}{\lambda}(\theta_t - \bar{\theta})$, where $\|e_t\|_2 \le C_{\mathrm{ng}}\sqrt{L}\epsilon$,

$$
\begin{aligned}
\frac{1}{2}\left\|\theta_{t+1} - \theta_{\bar{\theta}}^{\star}\right\|_2^2 &\le \frac{1}{2}\left\|\hat{\theta}_{t+1} - \theta_{\bar{\theta}}^{\star}\right\|_2^2 = \frac{1}{2}\left\|\theta_t - \eta g_t - \theta_{\bar{\theta}}^{\star}\right\|_2^2 \\
&= \frac{1}{2}\left\|\theta_t - \theta_{\bar{\theta}}^{\star}\right\|_2^2 - \eta\left\langle e_t + \nabla F^{\star}(\theta_t) + \frac{1}{\lambda}(\theta_t - \bar{\theta}), \theta_t - \theta_{\bar{\theta}}^{\star}\right\rangle + \frac{\eta^2}{2}\|g_t\|_2^2 \\
&\le \frac{1}{2}\left\|\theta_t - \theta_{\bar{\theta}}^{\star}\right\|_2^2 - \frac{\eta}{\lambda}\left\|\theta_t - \theta_{\bar{\theta}}^{\star}\right\|_2^2 + \eta\|e_t\|_2\left\|\theta_t - \theta_{\bar{\theta}}^{\star}\right\|_2 + \frac{\eta^2}{2}\|g_t\|_2^2 \\
&\le \frac{1}{2}\left\|\theta_t - \theta_{\bar{\theta}}^{\star}\right\|_2^2 - \frac{\eta}{\lambda}\left\|\theta_t - \theta_{\bar{\theta}}^{\star}\right\|_2^2 + \eta C_{\mathrm{ng}}\sqrt{L}\epsilon\left\|\theta_t - \theta_{\bar{\theta}}^{\star}\right\|_2 + 5\eta^2 L.
\end{aligned}
\tag{29}
$$

In the third line, we lower bounded $\left\langle \nabla F^{\star}(\theta_t) + \frac{1}{\lambda}(\theta_t - \bar{\theta}), \bar{\theta} - \theta_{\bar{\theta}}^{\star}\right\rangle$ by using strong convexity of $F^{\star} + \frac{1}{\lambda}\left\|\cdot - \bar{\theta}\right\|_2^2$; in the last line, we used the assumed bound on $e_t$ as well as

$$\|g_t\|_2 \le \|\nabla F^{\star}(\theta_t)\|_2 + \|e_t\|_2 + \frac{1}{\lambda}\left\|\theta_t - \bar{\theta}\right\|_2 \le 3\sqrt{L} + C_{\mathrm{ng}}\sqrt{L}\epsilon \le \sqrt{10L},$$

for sufficiently small $\epsilon$. Next, consider some iteration $t$ where iterate $\theta_t$ satisfies

$$\left\|\theta_t - \theta_{\bar{\theta}}^{\star}\right\|_2 \ge 4C_{\mathrm{ng}}\sqrt{L}\epsilon\lambda. \tag{30}$$

On this iteration, we have from the definition of $\eta$ that

$$\eta C_{\mathrm{ng}}\sqrt{L}\epsilon\left\|\theta_t - \theta_{\bar{\theta}}^{\star}\right\|_2 \le \frac{\eta}{4\lambda}\left\|\theta - \theta_{\bar{\theta}}^{\star}\right\|_2^2, \; 5\eta^2 L \le \frac{\eta}{4\lambda}\left\|\theta - \theta_{\bar{\theta}}^{\star}\right\|_2^2.$$

Plugging these bounds back into (29), on any iteration where (30) holds,

$$\left\|\theta_{t+1} - \theta_{\bar{\theta}}^{\star}\right\|_2^2 \le \left(1 - \frac{\eta}{\lambda}\right)\left\|\theta_t - \theta_{\bar{\theta}}^{\star}\right\|_2^2 = \left(1 - \frac{4}{5}C_{\mathrm{ng}}^2\epsilon\right)\left\|\theta_t - \theta_{\bar{\theta}}^{\star}\right\|_2^2.$$

Because the squared distance $\left\|\theta_t - \theta_{\bar{\theta}}^{\star}\right\|_2^2$ is bounded by $16L\lambda^2$ initially and decreases by a factor of $O(\epsilon)$ every iteration until (30) no longer holds, it will reach an iteration where (30) no longer holds within $T$ iterations. Finally, by (29), in every iteration $t$ after the first where (30) is violated,

either the squared distance to $\theta_{\hat{\theta}}^{\star}$ goes down, or it can go up by at most

$$\eta C_{\mathrm{ng}}\sqrt{L\epsilon}\left\|\theta_t - \theta_{\hat{\theta}}^{\star}\right\|_2 + 5\eta^2 L = O\left(\epsilon^2 \lambda^2 L\right).$$

Here we used our earlier claim that the distance can only go up when (30) is false. Thus, the squared distance will never be more than $16C_{\mathrm{ng}}^2 L(\epsilon + O(\epsilon^2))\lambda^2$ within $T$ iterations, as desired. $\qquad\square$

By using the gradient characterization in Fact 2 and the noisy gradient oracle implementation of Corollary 2, we conclude this section with our Moreau envelope noisy gradient oracle claim.

**Corollary 3.** *Consider a robust Lipschitz stochastic optimization instance where we have sample access to datasets $\mathbf{X} = \{X_i\}_{i\in[n]} \in \mathbb{R}^{n\times d}$ and $y = \{y_i\}_{i\in[n]} \in \mathbb{R}^n$ under Models 1, 2, 6 with sample size $n$ corresponding to Proposition 4. For*

$$F(\theta) = F_{\lambda}^{\star}(\theta) + \frac{\mu}{2}\|\theta\|_2^2,$$

*we can construct a $(L, O(1), \delta)$-radiusless noisy gradient oracle in $O(\frac{nd}{\epsilon}\log^3(n)\log^2(\frac{n}{\delta\epsilon})\log(\frac{1}{\epsilon}))$ time. The sample complexity of our noisy gradient oracle is*

$$O\left(\log\left(\frac{1}{\delta\epsilon}\right)\cdot\left(\frac{d\log(d/\epsilon)}{\epsilon}\right)\right).$$

## 5.2 Accelerated optimization of the regularized Moreau envelope

We conclude by combining Proposition 7, the smoothness bound from Fact 2, and the noisy gradient oracle implementation of Corollary 3 to give this section's main result, Theorem 8.

**Theorem 8.** *Under Models 1, 2, and 6, supposing $\epsilon\kappa^2$ is sufficiently small for $\kappa := \max(1, \frac{1}{\lambda\mu})$, given $\theta_0 \in \mathbb{R}^d$ and $\|\theta_0 - \theta_{\mathrm{env}}^{\star}\|_2 \le R_0$, RobustAccel using the noisy gradient oracle of Corollary 3 returns $\theta$ with $\|\theta - \theta_{\mathrm{env}}^{\star}\|_2 = O\left(\sqrt{\frac{\kappa\epsilon}{\mu}}\right)$ with probability at least $1 - \delta$. The algorithm runs in*

$$O\left(\frac{nd\sqrt{\kappa}}{\epsilon}\log\left(\frac{R_0\sqrt{\lambda^{-1}+\mu}}{\epsilon}\right)\log^3(n)\log^2\left(\frac{n\log\left(R_0\sqrt{\lambda^{-1}+\mu}\right)}{\delta\epsilon}\right)\log\left(\frac{1}{\epsilon}\right)\right) \quad time,$$

*where $n$ is the dataset size of Proposition 4. The sample complexity of the method is*

$$O\left(\log\left(\frac{\log\left(R_0\sqrt{\lambda^{-1}+\mu}\right)}{\delta\epsilon}\right)\cdot\left(\frac{d\log(d/\epsilon)}{\epsilon}\right)\right).$$

Combined with Fact 2, Theorem 8 offers a range of tradeoffs by tuning the parameter $\lambda$: the smaller $\lambda$ is, the more the Moreau envelope resembles the original function, but the statistical and runtime guarantees offered by Theorem 8 become correspondingly weaker.

## Acknowledgments

KT is supported by NSF Grant CCF-1955039 and the Alfred P. Sloan Foundation.

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

## A  Deferred proofs from Section 2

### A.1  Proof of Proposition 2

In this section, we prove Proposition 2, restated here for convenience.

**Proposition 2.** Let $\alpha \geq 1$ and let $\epsilon > 0$ be sufficiently small. Let $\{(X_i, y_i)\}_{i \in [n]} \subset \mathbb{R}^d \times \mathbb{R}$ be an $\epsilon$-corrupted set of samples from a distribution $\mathcal{D}_{Xy}$ as in Model 3. Then, if

$$n = O\left(\frac{d\alpha^2 \log d}{\epsilon^4} + \frac{d^2 \alpha^{1.5} \log(d/\epsilon)}{\epsilon^3}\right) ,$$

the set $\{(X_i, y_i)\}_{i \in [n]}$ is $(2\epsilon, \frac{\epsilon^2}{\alpha})$-good for linear regression with probability at least $\frac{9}{10}$.

Before we prove this lemma, we need the following useful technical lemmata. The first shows that given a large enough sample of points from a distribution with bounded second moment, there is a large subset of points with bounded second moment.

**Lemma 17** (Lemma A.20 in [DKK+17])**.** *Let $X_1, \ldots, X_n$ be independent samples from a distribution $\mathcal{D}$ with second moment matrix $\mathbf{\Sigma}^\star$, and let $\epsilon > 0$ be sufficiently small. There exists a universal constant $c > 0$ so that if $n \geq c\frac{d \log d}{\epsilon}$, we have that with probability 0.99, there exists a subset $S$ of size $(1 - \epsilon)n$ satisfying*

$$\frac{1}{|S|} \sum_{i \in S} X_i X_i^\top \preceq \frac{3}{2}\mathbf{\Sigma}^\star .$$

We note that Lemma A.20 is stated for covariance as opposed to second moment, but the same proof immediately implies the same result for second moment.

We also require the following bound.

**Lemma 18** (Lemma 5.1 in [CAT+20])**.** *Let $X_1, \ldots, X_n$ be independent samples from a distribution $\mathcal{D}$ with second moment $\mathbf{\Sigma}^\star$, and let $\epsilon > 0$ be sufficiently small. Assume $\mathcal{D}$ is 2-to-4 hypercontractive with parameter $C = O(1)$. There exists a universal constant $c > 0$ so that if $n \geq \frac{cd \log(d/\epsilon)}{\epsilon}$, then with probability $1 - \frac{1}{d^2}$, we have that for any $S \subset [n]$ of size $(1 - \epsilon)n$,*

$$\frac{1}{|S|} \sum_{i \in S} X_i X_i^\top \succeq (1 - \sqrt{\epsilon})\mathbf{\Sigma}^\star .$$

Finally, we show our main helper lemma, which is used to prove Assumption 1.2 holds.

**Lemma 19.** *Let $\epsilon > 0$ be sufficiently small. Let $X_1, \ldots, X_n$ be $n$ samples from a 2-to-4 hypercontractive distribution $\mathcal{D}$ with parameter $C$ and second moment $\mathbf{\Sigma}^\star$. Then, there exist universal constants $c, C_{\mathrm{est}} > 0$ so that if*

$$n \geq c\left(\frac{d \log d}{\epsilon^4} + \frac{d^2 \log(d/\epsilon)}{\epsilon^3}\right),$$

*then with probability 0.99, for every $u \in \mathbb{R}^d$, there exists an $G \subseteq [n]$ satisfying $|G| \geq (1 - \epsilon^2)n$, and*

$$\left\| \frac{1}{|G|} \sum_{i \in G} \langle X_i, u \rangle^2 X_i X_i^\top \right\|_{\mathrm{op}} \leq C_{\mathrm{est}} L \|u\|_{\mathbf{\Sigma}^\star}^2 .$$

*Proof.* Without loss of generality (by scale invariance), it suffices to prove this for all $u$ with $\|u\|_{\boldsymbol{\Sigma}^\star} = 1$. First, by Markov's inequality with $\mathbb{E}_{X \sim \mathcal{D}}[\|X\|_2^2] = \text{Tr}[\boldsymbol{\Sigma}^\star] \le Ld$, we have that

$$\Pr_{X \sim \mathcal{D}}\left[\|X\|_2^2 \ge \frac{20Ld}{\epsilon^2}\right] \le \frac{\epsilon^2}{20} .$$

Hence, by Bernstein's inequality, we have that with probability 0.999,

$$\frac{|\{i : \|X_i\|_2^2 \ge 20Ld\epsilon^{-2}\}|}{n} \le \frac{\epsilon^2}{10} . \tag{31}$$

Condition on this event holding for the rest of the proof.

For any vector $u \in \mathbb{R}^d$ with $\|u\|_{\boldsymbol{\Sigma}^\star} = 1$, let $H_u \subset \mathbb{R}^d$ be the set given by

$$H_u = \left\{ x \in \mathbb{R}^d : \langle x, u \rangle^2 \ge \frac{10C_{2\to4}^{1/2}}{\epsilon} \right\} .$$

Note that by Chebyshev's inequality, since $\mathbb{E}_{X \sim \mathcal{D}}[\langle x, u \rangle^4] \le C_{2\to4} \|u\|_{\boldsymbol{\Sigma}^\star}^4 \le C_{2\to4}$, $\Pr_{X \sim D}[X \in H_u] \le \frac{\epsilon^2}{100}$. Furthermore, the collection of sets $\{H_u\}_{\|u\|_{\boldsymbol{\Sigma}^\star}=1}$ has VC dimension $O(d)$, as each $H_u$ can be expressed as a restricted intersection of parallel halfspaces, and it is well-known that VC dimension is additive under intersection. Therefore, by the VC inequality, we know that if $n \ge c\frac{d\log d}{\epsilon^4}$ for sufficiently large constant $c$, with probability 0.999, we have that

$$\sup_{H_u} \frac{|\{i : X_i \in H_u\}|}{n} \le \frac{\epsilon^2}{50} . \tag{32}$$

Condition on this event holding for the rest of the proof. All expectations throughout the remainder of the proof are taken with respect to $X \sim \mathcal{D}$ for notational simplicity.

For any fixed $u$, we define the truncated fourth moment (contracted in the direction $u$) by

$$\mathbf{A}_i = \mathbf{A}_i(u) = \langle X_i, u \rangle^2 X_i X_i^\top \mathbb{1}\left[\langle X_i, u \rangle^2 \le \frac{20C_{2\to4}^{1/2}}{\epsilon} \text{ and } \|X_i\|_2^2 \le \frac{20Ld}{\epsilon^2}\right] .$$

Note that

$$\|\mathbb{E}[\mathbf{A}_i]\|_{\text{op}} \le \left\|\mathbb{E}\left[\langle X_i, u \rangle^2 X_i X_i^\top\right]\right\|_{\text{op}} = \sup_{\|v\|_2 = 1} \mathbb{E}\left[\langle X_i, u \rangle^2 \langle X_i, v \rangle^2\right] \le C_{2\to4}L \|u\|_{\boldsymbol{\Sigma}^\star}^2 ,$$

by Cauchy-Schwarz and hypercontractivity. Moreover, by construction, the spectral norm of $\mathbf{A}_i$ is bounded almost surely by $\frac{400LC_{2\to4}^{1/2}d}{\epsilon^3}$. Hence, by a matrix Chernoff bound, we get that if $n \ge c\frac{d^2 \log(d/\epsilon)}{\epsilon^3}$ for a sufficiently large constant $c$, then with probability 0.999, we have that

$$\left\|\frac{1}{n}\sum_{i=1}^n \mathbf{A}_i(u)\right\|_{\text{op}} \le 2C_{2\to4}L \|u\|_{\boldsymbol{\Sigma}^\star}^2 . \tag{33}$$

for all $u$ in a poly$(\frac{\epsilon}{d})$-net of the unit sphere in the $\boldsymbol{\Sigma}^\star$ norm (which has cardinality $(\frac{d}{\epsilon})^{O(d)}$ by Theorem 1.13 of [RH17]). Because we are union bounding over poly$(d, \epsilon^{-1})$ samples, we have with high probability that all $\|X_i\|_{(\boldsymbol{\Sigma}^\star)^{-1}} = \text{poly}(d, \epsilon^{-1})$. Hence, it is straightforward to show that the

bound (33) over our net implies for all $\|u\|_{\Sigma^\star} = 1$,

$$\left\| \frac{1}{n} \sum_{i=1}^{n} \langle X_i, u \rangle^2 X_i X_i^\top \mathbb{1} \left[ X_i \notin H_u \text{ and } \|X_i\|_2^2 \leq \frac{10 C_{2\to4}^{1/2} L d}{\epsilon} \right] \right\|_{\mathrm{op}} \leq 2 C_{2\to4} L , \qquad (34)$$

Combining (31), (32), and (34) implies that for every $u$, the set $G = \{i : X_i \in H_u \text{ and } \|X_i\|_2^2 \leq \frac{20Ld}{\epsilon^2}\}$ satisfies the conditions of the lemma. $\qquad \square$

We are now ready to prove Proposition 2.

*Proof of Proposition 2.* Condition 3 of Assumption 1 follows directly from Markov's inequality, since it is asking about the empirical average over $G$ of $\delta^2 \sim \mathcal{D}_\delta$; the adversary removing points can only affect this upper bound by a constant factor (due to renormalization).

Next, let $[n] = G \cup B$ be the canonical decomposition of the corrupted set of samples. By two applications of Lemma 17, with probability at least 0.99, there exists a set $G' \subset G$ of size $|G'| \geq (1 - \epsilon)|G| \geq (1 - 2\epsilon)n$ so that

$$\frac{1}{|G'|} \sum_{i \in G'} X_i X_i^\top \preceq \frac{3}{2} \Sigma^\star , \qquad (35)$$

$$\frac{1}{|G'|} \sum_{i \in G'} \delta_i^2 X_i X_i^\top \preceq \frac{3}{2} \sigma^2 \Sigma^\star . \qquad (36)$$

Condition on the event that such a $G'$ exists for the remainder of the proof, and also condition on the event that Lemma 19 is satisfied. By a union bound, these events happen together with probability at least 0.9. We will show that this $G'$ will satisfy the conditions of the lemma. The upper bound in Condition 1 of Assumption 1 is immediate, and similarly, the lower bound follows from Lemma 18 and a standard convexity argument (since the vertices of the polytope defining saturated weights are subsets of cardinality $(1 - O(\epsilon))|G|$).

It thus remains to prove Condition 2 of Assumption 1. To do so, we will first prove (8) is satisfied with high probability. By Lemma 19 (adjusting by a factor of $\alpha$ in the definition of $\epsilon^2$), there exists a set $G'' \subseteq G'$ so that $|G''| \geq (1 - \frac{\epsilon^2}{\alpha})|G'|$ so that

$$\left\| \frac{1}{|G''|} \sum_{i \in G''} \langle X_i, \theta - \theta^\star \rangle^2 X_i X_i^\top \right\|_{\mathrm{op}} \leq C_{\mathrm{est}} L \|\theta - \theta^\star\|_{\Sigma^\star}^2 .$$

Hence, for this choice of $G''$, we have

$$
\begin{aligned}
\operatorname{Cov}_{\tilde{w}}\left(\{g_i(\theta)\}_{i \in G''}\right) &= \sum_{i \in G''} \tilde{w}_i \left(\langle X_i, \theta - \theta^\star\rangle + \delta_i\right)^2 X_i X_i^\top \\
&\preceq 2 \sum_{i \in G''} \tilde{w}_i \langle X_i, \theta - \theta^\star\rangle^2 X_i X_i^\top + 2 \sum_{i \in G} \tilde{w}_i \delta_i^2 X_i X_i^\top \\
&\preceq \frac{2(1+2\epsilon)}{|G''|} \sum_{i \in G''} \langle X_i, \theta - \theta^\star\rangle^2 X_i X_i^\top + \frac{2(1+2\epsilon)}{|G'|} \sum_{i \in G'} \delta_i^2 X_i X_i^\top \\
&\preceq \frac{2(1+2\epsilon)}{|G'|} \sum_{i \in G''} \langle X_i, \theta - \theta^\star\rangle^2 X_i X_i^\top + 3\sigma^2 \Sigma^\star \\
&\preceq 2(1+2\epsilon) C_{\mathrm{est}} L \left(\|\theta - \theta^\star\|_{\Sigma^\star}^2 + \sigma^2\right) \mathbf{I}.
\end{aligned}
$$

where the last line follows from (36). By suitably adjusting the choice of $C_{\mathrm{est}}$, this proves that (8) is satisfied for this choice of $G''$. Finally, we claim that (8) implies (7) via standard techniques from the robust mean estimation literature, e.g. in the proof of Lemma 3.2 in [DHL19]. $\qquad \square$

## A.2 Proof of Proposition 5

In this section, we state FastCovFilter and prove Proposition 5, restated for convenience.

**Proposition 5.** *There is an algorithm,* FastCovFilter *(Algorithm 7), taking inputs* $\mathbf{V} := \{v_i\}_{i \in [n]} \in \mathbb{R}^{n \times d}$, *saturated weights* $w \in \Delta^n$ *with respect to bipartition* $[n] = G \cup B$ *with* $|B| = \epsilon n$, $\delta \in (0, 1]$, *and* $R \geq 0$ *with the promise that*

$$
\left\| \sum_{i \in G} \frac{1}{|G|} v_i v_i^\top \right\|_{\mathrm{op}} \leq R.
$$

*Then, with probability at least* $1 - \delta$, FastCovFilter *returns saturated* $w' \in \Delta^n$ *such that*

$$
\left\| \sum_{i \in [n]} w_i' v_i v_i^\top \right\|_{\mathrm{op}} \leq 5R.
$$

The runtime of FastCovFilter is

$$
O\left(nd \log^3(n) \log\left(\frac{n}{\delta}\right)\right).
$$

Before proving Proposition 5, we require three helper facts.

**Fact 3** (Theorem 1, [MM15]). *For any* $\delta \in (0, 1]$ *and* $\mathbf{M} \in \mathbb{S}_{\geq 0}^d$, *there is an algorithm,* Power$(\mathbf{M}, \delta)$, *which returns with probability at least* $1 - \delta$ *a value* $V$ *such that* $\lambda_{\max}(\mathbf{M}) \geq V \geq 0.9\lambda_{\max}(\mathbf{M})$. *The algorithm costs* $O(\log \frac{d}{\delta})$ *matrix-vector products through* $\mathbf{M}$ *plus* $O(d \log \frac{d}{\delta})$ *additional runtime.*

**Fact 4** (Lemma 7, [JLT20]). *Let* $\mathbf{A}, \mathbf{B} \in \mathbb{S}_{\geq 0}^d$ *with* $\mathbf{A} \succeq \mathbf{B}$, *and* $p \in \mathbb{N}$. *Then* $\operatorname{Tr}(\mathbf{A}^{p-1}\mathbf{B}) \geq \operatorname{Tr}(\mathbf{B}^p)$.

**Fact 5.** *For any* $\alpha \geq 0$ *and* $\mathbf{A} \in \mathbb{S}_{\geq 0}^d$,

$$
\alpha \operatorname{Tr}(\mathbf{A}^{2\log d}) \leq \operatorname{Tr}(\mathbf{A}^{2\log d+1}) + d\alpha^{2\log d+1}.
$$

*Proof.* Every eigenvalue $\lambda$ of $\mathbf{A}$ is either at least $\alpha$ (and hence $\lambda^{2\log d+1} \geq \alpha\lambda^{2\log d}$) or not (and hence $\alpha^{2\log d+1} \geq \alpha\lambda^{2\log d}$). Both of these cases are accounted for by the right hand side. $\qquad \square$

---

**Algorithm 7** FastCovFilter($\mathbf{V}, w, \delta, R$)

---

1: **Input:** $\mathbf{V} := \{v_i\}_{i \in [n]} \in \mathbb{R}^{n \times d}$, saturated weights $w \in \Delta^n$ with respect to bipartition $[n] = G \cup B$
   with $|B| = \epsilon n$ for sufficiently small $\epsilon$, $\delta \in (0, 1)$, $R \geq \left\| \sum_{i \in G} \frac{1}{|G|} v_i v_i^\top \right\|_{\mathrm{op}}$

2: **Output:** With probability $\geq 1 - \delta$, saturated $w'$ with respect to bipartition $G \cup B$, with

$$\left\| \sum_{i \in [n]} w_i' v_i v_i^\top \right\|_{\mathrm{op}} \leq 5R.$$

3: Remove all $i \in [n]$ with $\|v_i\|_2^2 \geq nR$, $n \leftarrow$ new dataset size
4: $T \leftarrow O(\log^2 n)$ (for a sufficiently large constant), $t \leftarrow 0$, $w^{(0)} \leftarrow w$
5: **while** $t < T$ and $\mathsf{Power}\left( \sum_{i \in [n]} w_i^{(t)} v_i v_i^\top, \frac{\delta}{2T} \right) > 2R$ **do**
6: $\quad \mathbf{M}_t \leftarrow \sum_{i \in [n]} w_i^{(t)} v_i v_i^\top$, $\mathbf{Y}_t \leftarrow \mathbf{M}_t^{\log d}$
7: $\quad$ Sample $N_{\mathrm{dir}} = O(\log \frac{n}{\delta})$ (for a sufficiently large constant) vectors $\{u_j\}_{j \in [N_{\mathrm{dir}}]} \in \mathbb{R}^d$ each with independent entries $\pm 1$. Let $\tilde{u}_j \leftarrow \mathbf{Y}_t u_j$ for all $j \in [N_{\mathrm{dir}}]$.
8: $\quad$ **for** $j \in [N_{\mathrm{dir}}]$ **do**
9: $\quad\quad \tau_i \leftarrow \langle v_i, \tilde{u}_j \rangle^2$ for all $i \in [n]$
10: $\quad\quad \tau_{\max} \leftarrow \max_{i \in [n] \| w_i \neq 0} \tau_i$
11: $\quad\quad$ **while** $\sum_{i \in [n]} w_i^{(t)} \tau_i \geq 2R \|\tilde{u}_j\|_2^2$ **do**
12: $\quad\quad\quad w_i^{(t)} \leftarrow \left( 1 - \frac{\tau_i}{\tau_{\max}} \right) w_i^{(t)}$ for all $i \in [n]$
13: $\quad\quad$ **end while**
14: $\quad$ **end for**
15: $\quad w^{(t+1)} \leftarrow w^{(t)}$, $t \leftarrow t + 1$
16: **end while**
17: **return** $w^{(t)}$

---

*Proof of Proposition 5.* We discuss correctness, runtime, and the failure probability separately.

*Correctness.* First, Line 3 is correct because these indices cannot belong to $G$ as they would certify a violation to the operator norm bound in the direction of $v$, so this preserves saturation. Next, it is clear by Fact 3 that if the algorithm ever ends because $\mathsf{Power}$ returns too small a number, the output is correct, so it suffices to handle the other case. Define the potential function $\Phi_t := \mathrm{Tr}(\mathbf{Y}_t^2)$. Our main goal is to show that in every iteration the algorithm runs, $\Phi_t$ decreases substantially. To this end, we claim that after all runs of Lines 8-14 of $\mathsf{FastCovFilter}$ have finished, we have in all randomly sampled directions $j \in [N_{\mathrm{dir}}]$ the guarantee

$$\left\langle \tilde{u}_j \tilde{u}_j^\top, \sum_{i \in [n]} w_i^{(t)} v_i v_i^\top \right\rangle \leq 2R \|\tilde{u}_j\|_2^2. \tag{37}$$

This is immediate from the termination condition on Line 11 for each $j \in [N_{\mathrm{dir}}]$, the fact that weights are monotone nonincreasing throughout the whole algorithm, and that the left hand side of (37) is monotone nonincreasing as a function of the weights. Next, by the Johnson-Lindenstrauss

lemma of [Ach03], for a sufficiently large $N_{\mathrm{dir}}$ with probability at least $1 - \frac{\delta}{4T}$,

$$\frac{1}{N_{\mathrm{dir}}} \sum_{j \in [N_{\mathrm{dir}}]} \langle v_i, \tilde{u}_j \rangle^2 = \frac{1}{N_{\mathrm{dir}}} \sum_{j \in [N_{\mathrm{dir}}]} v_i^\top \mathbf{Y}_t \tilde{u}_j \tilde{u}_j^\top \mathbf{Y}_t v_i \in [0.95, 1.05] \left\langle v_i v_i^\top, \mathbf{Y}_t^2 \right\rangle \quad \text{for all } i \in [n].$$

Condition on this event for all runs of Lines 8-14 throughout the algorithm for the remainder of the proof. Combining this guarantee with (37), we have that after Lines 8-14 terminate,

$$\left\langle \mathbf{Y}_t^2, \sum_{i \in [n]} w_i^{(t)} v_i v_i^\top \right\rangle = \sum_{i \in [n]} w_i^{(t)} \left\langle v_i v_i^\top, \mathbf{Y}_t^2 \right\rangle$$

$$\leq \frac{1.1}{N_{\mathrm{dir}}} \sum_{j \in [N_{\mathrm{dir}}]} \sum_{i \in [n]} w_i^{(t)} \left\langle v_i, \tilde{u}_j^2 \right\rangle$$

$$= \frac{1.1}{N_{\mathrm{dir}}} \sum_{j \in [N_{\mathrm{dir}}]} \left\langle \tilde{u}_j \tilde{u}_j^\top, \sum_{i \in [n]} w_i^{(t)} v_i v_i^\top \right\rangle \leq \frac{2.2R}{N_{\mathrm{dir}}} \sum_{j \in [N_{\mathrm{dir}}]} \|\tilde{u}_j\|_2^2.$$

Next, by the Johnson-Lindenstrauss lemma of [Ach03], since all $\{u_j\}_{j \in [N_{\mathrm{dir}}]}$ were sampled independently of $\mathbf{M}_t$, we have with probability at least $1 - \frac{\delta}{4T}$ that

$$\frac{1}{N_{\mathrm{dir}}} \sum_{j \in [N_{\mathrm{dir}}]} \|\tilde{u}_j\|_2^2 = \frac{1}{N_{\mathrm{dir}}} \sum_{j \in [N_{\mathrm{dir}}]} u_j^\top \mathbf{Y}_t^2 u_j \leq 1.1 \mathrm{Tr}(\mathbf{Y}_t^2).$$

Conditioning on this event in every iteration, at the start of the next iteration, we will have

$$\left\langle \mathbf{Y}_t^2, \mathbf{M}_{t+1} \right\rangle \leq 2.5R \mathrm{Tr}(\mathbf{Y}_t^2). \tag{38}$$

We now show how (38) implies a rapid potential decrease:

$$\Phi_{t+1} = \mathrm{Tr}\left(\mathbf{M}_{t+1}^{2\log d}\right) \leq \frac{1}{2.8R} \mathrm{Tr}\left(\mathbf{M}_{t+1}^{2\log d+1}\right) + d(2.8R)^{2\log d}$$

$$\leq \frac{1}{2.8R} \mathrm{Tr}\left(\mathbf{Y}_t^2 \mathbf{M}_{t+1}\right) + d(2.8R)^{2\log d} \leq 0.9 \Phi_t + d(2.8R)^{2\log d}.$$

In the first inequality, we used Lemma 5 with $\alpha = 2.8R$; in the second, we used Fact 4 with $\mathbf{A} = \mathbf{M}_t$, $\mathbf{B} = \mathbf{M}_{t+1}$, and $p = 2\log d + 1$. The last inequality applied (38). The above display implies that until $\Phi_t \leq 20d(2.8R)^{2\log d}$, the potential is decreasing by a constant factor every iteration, and $\Phi_0 \leq d\lambda_{\max}(\mathbf{M}_0)^{2\log d} \leq d(nR)^{2\log d}$, so within $O(\log^2 n)$ iterations we will have

$$\Phi_t = \mathrm{Tr}(\mathbf{M}_t^{2\log d}) \leq 20d(2.8R)^{2\log d}.$$

At this point, it is clear the operator norm of $\mathbf{M}_t$ achieves the desired bound of $5R$. It remains to show that all weight removals in Lines 11-13 were safe throughout the algorithm. Here we use Lemma 1: it suffices to show that throughout the algorithm,

$$\sum_{i \in G} w_i^{(t)} \tau_i \leq R \|\tilde{u}_j\|_2^2, \tag{39}$$

because then whenever Line 11 fails, the scores are safe with respect to the weights and Lemma 1

applies. However, (39) follows from the assumption on $\left\| \sum_{i \in G} \frac{1}{|G|} v_i v_i^\top \right\|_{\mathrm{op}}$, yielding

$$\sum_{i \in G} w_i^{(t)} \tau_i \leq \sum_{i \in G} \frac{1}{|G|} \tau_i = \left\langle \tilde{u}_j \tilde{u}_j^\top, \sum_{i \in G} \frac{1}{|G|} v_i v_i^\top \right\rangle \leq R \left\| \tilde{u}_j \right\|_2^2.$$

*Runtime.* The cost of all lines other than Lines 6-7 and the repeated loops of Lines 11-13 clearly fall within the budget. To implement Lines 6-7, we never need to form the matrices $\mathbf{M}_t$ or $\mathbf{Y}_t$, and instead form all $\{\tilde{u}_j\}_{j \in [N_{\mathrm{dir}}]}$ in time

$$O \left( nd \log d \log \frac{n}{\delta} \right)$$

implicitly through matrix-vector multiplications with $\mathbf{M}_t$, each of which take time $O(nd)$. To implement Lines 11-13, let $\bar{w}$ denote the value of $w^{(t)}$ right after an execution of Line 10. We wish to determine the smallest value $K$ such that

$$\sum_{i \in [n]} \left( 1 - \frac{\tau_i}{\tau_{\max}} \right)^K \bar{w}_i \tau_i \leq 2R \left\| \tilde{u}_j \right\|_2^2.$$

Checking if the above display holds for a particular guess of $K$ clearly takes $O(n)$ time, and we can upper bound $K$ by the following inequality:

$$\sum_{i \in [n]} \left( 1 - \frac{\tau_i}{\tau_{\max}} \right)^K \bar{w}_i \tau_i \leq \sum_{i \in [n]} \exp \left( -\frac{K \tau_i}{\tau_{\max}} \right) \bar{w}_i \tau_i \leq \frac{1}{eK} \sum_{i \in [n]} \bar{w}_i \tau_{\max} \leq \frac{\tau_{\max}}{K}.$$

Here the second inequality used $x \exp(-Cx) \leq \frac{1}{eC}$ for all nonnegative $x, C$, where we chose $C = \frac{K}{\tau_{\max}}$ and $x = \tau_i$. Now since $\tau_{\max} \leq \left\| \tilde{u}_j \right\|_2^2 \left\| v_i \right\|_2^2 \leq nR \left\| \tilde{u}_j \right\|_2^2$ by Cauchy-Schwarz, we have that $K = O(n)$ as desired. At this point, a binary search on $K$ suffices, so all loops take time $O(n \log n)$.

*Failure probability.* The only randomness used in the algorithm appears in the guarantees of Power and the guarantees of the Johnson-Lindenstrauss projections. Taking a union bound over $T$ iterations shows these all succeed with probability at least $1 - \delta$. $\square$