# OpenReview forum: "Robust Regression Revisited: Acceleration and Improved Estimation Rates"
_NeurIPS.cc/2021/Conference — NeurIPS 2021 Poster_

### Official Review · Reviewer_jG4X · 2021-07-12

**Rating:** 6
**Confidence:** 2

**Summary:**

The proposed new algorithms for robust regression problems with improved runtime and estimation guarantees.

**Limitations And Societal Impact:**

This is mainly a theoretical paper while I would prefer some simulations or experiments to show the improvements of the new algorithm do exist.

**Main Review:**

I think it is important to have fast algorithms that optimize a generalized linear model (GLM) given corrupted samples. The new algorithm in the paper has improved the running time and estimation error comparing to prior work. However,  I am not familiar with the related work and it is unclear to me how significant the contribution is. One suggestion is that the authors can present a table that summarizes the error estimation bound and running time of prior works and the work in this paper so that we have a better understanding of how much improvements have been made. In addition, as a paper aiming to improve running time, it would be great to have some experiments to show that the improvements do exist.

**Time Spent Reviewing:**

6

---

> ### Author Response · Authors · 2021-08-10
> **Response to Reviewer jG4X**
>
> Thank you for your review and helpful comments. Due to the scope of the paper and the range of theoretical questions we aimed to tackle through our frameworks, we chose to focus on developing the theory of robust regression in our present submission. However, we agree that empirical evaluations of our theoretical improvements are important, and defer such a study to future work.
>
> We agree with your suggestion of including a table. We summarized the improvements of our paper compared to the existing literature in the meta-comment. We hope this clarifies the position of our paper’s results with respect to state-of-the-art (namely, the nature of both our runtime and statistical improvements), and that our explanation elevates your view of our paper.

---

### Official Review · Reviewer_Xq7Q · 2021-07-15

**Rating:** 7
**Confidence:** 4

**Summary:**

This paper resolves an open question of the recent work [PSBR 20], which uses robust gradient descent for robust estimation tasks, to achieve faster accelerated convergence rate and better statistical accuracy. This is done via a clever modification to the accelerated proximal point framework of [MS13] with the gradient step replaced by noisy gradient.

**Ethical Concerns:**

-

**Limitations And Societal Impact:**

-

**Main Review:**

This paper extends the recent work [PSBR 20], which uses robust gradient descent for robust estimation tasks, to achieve faster accelerated convergence rate and better statistical accuracy. This is done via a clever modification to the accelerated proximal point framework of [MS13] with the gradient step replaced by noisy gradient.

The authors proposed a clean formulation of robust gradient estimators as an oracle, and showed how to implement this oracle in near linear time which improves from prior works. Next, the author showed that the algorithm of [MS13], is "stable" to the noise in the gradient oracle and can achieve a smaller statistical error via a iterating halving argument. Finally, the author showed how the proposed methodology can be applied to robust linear regression with strong statistical guarantee and practical computational complexity, without resorting to the costly Sum-of-square estimators.

Overall, this paper is well-written and it solved an important open question elegantly. I recommend a clear accept.

The only minor drawback of this approach, is that it requires a larger number of samples comparing to the SoS-based estimators (and other slightly stronger assumptions) - there is a d^2 factor in the O(1/eps) term. I am curious about the necessity about this - is there some fundamental limit of the robust GD-based approach that strictly separates it from SoS?

**Time Spent Reviewing:**

5

---

> ### Author Response · Authors · 2021-08-10
> **Response to Reviewer Xq7Q**
>
> Thank you for your helpful and encouraging comments, as well as your clarifying questions. We are glad that you found our paper well-written, and our approach elegant.
>
> With regards to the paper summary you provide, we also would like to point out that our paper contains two types of algorithms: one is the improvement of the method of [PSBR20] as you mentioned, building upon the analysis of [MS13]. The other type of algorithm we develop, which is incomparable to our acceleration framework, is detailed in Section 4 of the main submission. It obtains guarantees which are incomparable to the accelerated linear regression algorithm of Section 3, with improved estimation rates and robustness tolerance (weaker assumption on relationship between $\epsilon$ and $\kappa$) at the cost of a worse sample size.
>
> We agree that it is an interesting open problem whether the worse sample complexity of Theorem 4 (compared to Theorem 3) is truly necessary, and it is currently unclear to us. This bottleneck arises because the analysis of Theorem 4 requires all filtering steps to be with respect to the same set (up to small additive perturbations). This requires a stronger statistical assumption (Assumption 1 vs. Assumption 2 in the supplement).  To satisfy the more stringent Assumption 1, in order to apply a matrix Chernoff bound, the sample complexity is inflated by a factor of $d$ (this comes from the bound in the first displayed equation in Lemma 20’s proof, see the supplement). However, as you say it may be a difficulty in the analysis rather than a fundamental limit, and we will make this distinction clear in the exposition.

---

### Official Review · Reviewer_56zo · 2021-07-17

**Rating:** 7
**Confidence:** 4

**Summary:**

The paper develops fast algorithms for robust stochastic optimization in the presence of strong contamination model, where the objective function belongs to one of the following: linear regression, (regularized) generalized linear models, and (regularized) Moureau Envelope. The main contributions of the paper are as follows:
1. Building on the gradient descent framework of [PSBR20], the paper develops accelerated gradient descent method using a noisy gradient oracle. This allows them to get better dependence on the condition number in the all of the mentioned models (with similar statistical rate as existing works).
2. Inspired by the identifiability proof of [BP21], the paper gives a practical algorithm for linear regression that has better statistical rates than existing works (at the cost of additional sample complexity). This is the first work that achieves this rate without certifiable hypercontractivity.




**Limitations And Societal Impact:**

Yes

**Main Review:**

I believe the paper is a good contribution to the field of robust stochastic optimization, showing that proximal accelerated methods are amenable to noisy gradient oracles. Thus I recommend "accept" for the paper with the following changes.

**Main Comments**

1. As $\kappa$, the condition number, is an important parameter in this project, the dependence on $\kappa$ must be mentioned in the sample complexities in main theorems. (For example, value of $\alpha$ from Line 316 + Proposition 1 suggests that there is a quadratic dependence on $\kappa$ in Theorem 4).

2. Independence of noise: It must be mentioned in the main paper that the noise distribution is independent of the covariates  in Theorem 3 and 4 (Model 3 and Model 4 in the supplementary material). This is a critical modeling assumption.

3. The paper should clearly state the contributions of their work. In particular, while comparing with [CAT+20], the focus should be on the number of calls to robust mean estimation. Removing $O(\epsilon^{-6})$ in runtime from [CAT+20] is a blackbox benefit from [DHL19] or [DL19]. It is not obvious from reading the paper.

4. Paper claims multiple times that there are only two approaches for robust linear regression: (i) sum-of-squares based approaches and (ii) Gradient based approaches. The paper is missing the "covariate filtering" approach of [PJL20].

5. Theorem 3 and 4 have different upper bounds on $\epsilon$ in terms of $\kappa$. What is the main reason behind this?

**Other Comments**

1. Line 144: Dependence on $d$ is improved but is worse on $\epsilon$.

2. Line 36: What is "highly robust regression"?

3. Line 99: [Dep20] also provides a near linear time algorithm.

4. Line 64: Typo in defining the domain and range of $\gamma$.

5. Line 165: In [BDLS17], apart from Gaussianity, the identity covariance assumption is critical too.

6. Line 206: what is $c$ here?

7. Footnote 7: What depends polylogarithmically on failure probability? Time complexity.

8. Line 363: The last expression should be $2\alpha/\epsilon^2$ instead of $\epsilon^2/2 \alpha$.

**References**

[Dep20] J. Depersin. A spectral algorithm for robust regression with subgaussian rates. CoRR abs/2007.06072. (2020)

[DL19] J. Depersin & G. Lecué. Robust subgaussian estimation of a mean vector in nearly linear time. CoRR abs/1906.03058, (2019)

[PJL20] A. Pensia, V. Jog, & P. Loh. Robust regression with covariate filtering: Heavy tails and adversarial contamination. CoRR abs/2009.12976, (2020)

===============UPDATE After Author Response====================

I thank the authors for their thoughtful response, and answering my questions. I maintain my score and recommend 'acceptance'.

**Time Spent Reviewing:**

5

---

> ### Author Response · Authors · 2021-08-10
> **Response to Reviewer 56zo**
>
> Thank you for your thorough feedback, and very helpful pointers to references.
>
> *Main comments.*
>
> Regarding the statement of dependence on condition number, we made the simplification of eliminating $\kappa$ terms when they were not dominant, since $\kappa^2 = O(1/\epsilon)$ by assumption. We will carefully clarify this in a revision whenever relevant.
>
> Regarding independence of noise, thank you for pointing this out; we agree this is important, and will clearly state this in the main body in a revision. The assumption of independent noise was also used by the most directly prior work [CAT+20]. We believe it is an interesting problem to relax this assumption.
>
> We agree that the removal of $\epsilon$ factors from the gradient oracle construction runtime should not be substantially emphasized as a technical contribution, as it is implicit in prior work. We will clarify this in a revision when stating our results (e.g. Theorem 3).
>
> Thank you for the pointer to the work [PJL20]. We will make sure to discuss this work in the revision. At a high level, the main difference is that our paper focuses primarily on the ill-conditioned setting, i.e. improving the statistical dependence on $\kappa$ in the final error guarantee, whereas [PJL20] focuses primarily on the well-conditioned setting, when the true covariance is either identity, or at the very least, has constant condition number.
>
> The improved range of $\epsilon$ in Theorem 4 compared to Theorem 3 is a novel contribution of the algorithm we develop in Section 4. By using the identifiability proof from the SoS algorithm directly, we are able to avoid lossy norm conversions between the Mahalanobis norm and the $\ell_2$ norm, and improve the dependence of our analysis on $\kappa$, yielding this improvement.
>
> *Other comments.*
>
> Regarding comment 2, we will take care to not use unnecessary jargon in the revision, and be more precise. By this terminology, we meant to say robust regression with weak distributional or corruption assumptions.
>
> Regarding comment 6, there was a typo; $c$ is meant to be an additive relaxation. We will correct this in a revision; see Definition 1 in the supplement.
>
> Regarding comment 7, we will clarify the dependence. Along with the runtime, the sample complexity in some of our applications (namely, Theorems 1-3) also depends polylogarithmically on the failure probability, due to the construction of our gradient oracle (see Corollaries 1 and 2 in the supplement).
>
> We agree with comments 1, 3, 4, 5, and 8, and will make sure to address these in a revision. Thank you once again for your careful reading.

---

### Official Review · Reviewer_KSqf · 2021-08-02

**Rating:** 7
**Confidence:** 3

**Summary:**

This paper consider the robust regression problem with adversarial corruption, including linear regression, generalized linear models (GLMs) and Moreau envelopes of Lipschitz GLMs. Leveraging recent advances in robust mean estimation and proximal point methods, they provide improved algorithms in terms of optimization convergence rates and statistical error rates.

**Limitations And Societal Impact:**

This work is theoretical oriented.

**Main Review:**

This paper consider the robust regression problem with adversarial corruption, including linear regression, generalized linear models (GLMs) and Moreau envelopes of Lipschitz GLMs. Leveraging recent advances in robust mean estimation and proximal point methods, they provide improved algorithms in terms of optimization convergence rates and statistical error rates.

Specifically, their contributions include:

1) Following the noisy gradient oracle framework in [PSBR20], they use tools from [DHL19] to show that such an oracle can be implemented in near-linear time. Furthermore, they show that the accelerated proximal point algorithm in [MS13] is robust to the noise in the above oracle. Combining these results allows them to develop accelerated algorithms whose convergence rates feature better dependence on the condition number without sacrificing the statistical error rates.

2) In the context of robust linear regression, they leverage the identifiability proof in [BP21] to develop a non-SOS algorithm that achieves better statistical error rates.

This paper is a solid contribution to the problem of robust estimation/regression, improving upon existing work in several aspects and in particular answering the question left open in [PSBR20] on whether robust gradient descent can be accelerated.

Other comments:

- In the statement of all theoretical results, the authors should clearly speak out the dependence on all relevant problem parameters including the condition number $\kappa$, the dimension $d$ and the error $\epsilon$.
- The authors may consider providing a summary (say as a table) of their quantitative results and those from prior work for ease of comparison.
- Line 64: should be $\gamma: \mathbb{R}^2 \times \mathbb{R} \to \mathbb{R}$


===================

I acknowledge that I have read the rebuttal and other comments. I maintain my score.

**Time Spent Reviewing:**

6

---

> ### Author Response · Authors · 2021-08-10
> **Response to Reviewer KSqf**
>
> Thank you for your careful reading of our paper and helpful feedback. We are glad that you felt our contributions to the fields of robust estimation and regression were significant.
>
> Regarding the statement of dependencies on $\kappa$, $d$, and $\epsilon$, each of our main results in the primary submission (Theorems 1-4) have precise statements of the algorithm runtimes, accuracies, and sample complexities in terms of these parameters, up to logarithmic factors (for notational convenience, with complete statements in the supplement). We note there was an erratum to the sample complexity of Theorem 4, as stated in the first page of the supplement.
>
> Regarding the comment in Line 64, the correct statement is $\gamma: \mathbb{R}^2 \to \mathbb{R}$: we will fix this, and thank you for this observation.

---

### Author Response · Authors · 2021-08-10
**Meta-comment**

Reviewers 1 and 4 suggested including a table comparing our results to the prior work, for convenience to the reader. We agree with this comment and will make sure to include one in the revision. For convenience to the reviewers, we summarize the comparisons and improvements in this comment.

Theorem 1 is most directly comparable to the work of [PSBR20], maintaining the estimation rate and sample complexity while accelerating the dependence on kappa in the runtime. The problem formulations we consider are slightly different (we directly regularize the objective to make it strongly convex, whereas [PSBR20] typically constrains the domain to maintain conditioning), but in terms of the black-box formulation of a robust gradient oracle (i.e. Equation 2), this speedup is representative of our improvement. The work [DKK+19] also obtains results for this problem, but the number of iterations scales with the dimension in the worst case, which can be much larger than $\kappa$ or $\sqrt{\kappa}$ in many settings.

Theorem 2 is a novel result to the best of our knowledge, but follows mostly straightforwardly from the techniques of Theorem 1. We do not know of comparable results in the literature in our setting.

Theorem 3 is most directly comparable to the main algorithmic result of [CAT+20], and similarly to Theorem 1, it maintains the sample complexity and estimation error while accelerating the runtime (by a $\sqrt{\kappa}$ factor).

Theorem 4 is not strictly comparable to similar results in [DKK+19, CAT+20]. It has a larger sample complexity than [CAT+20], but has a higher robustness tolerance ($\epsilon < 1/\kappa$ rather than $\epsilon < 1/\kappa^2$) and an improved estimation error (scaling as $\sqrt{\kappa}$ rather than $\kappa$). Both of these latter qualities are the first of their kind in the literature amongst non-SoS algorithms (notably, our algorithm also does not require “certifiable hypercontractivity”).

Theorem 4 retains a nearly-linear runtime in the dataset size (using roughly $1/\epsilon$ calls to a stationary point finder), improving upon the estimation rates of [DKK+19] (which uses roughly $d$ calls, albeit with smaller dataset size) in many interesting regimes without substantially sacrificing in overall runtime. The runtime is not directly comparable to [CAT+20], but will typically be worse because of the larger dataset size.

---

### Decision · Program_Chairs · 2021-09-27

**Decision:**

Accept (Poster)

**Comment:**

The reviewers all agree that this paper makes solid contributions to the robust estimation problem and improves upon existing work in several aspects. The authors should incorporate the reviewers' suggestions into the final version of the paper.